# Neural Collapse under Gradient Flow on Shallow ReLU Networks for Orthogonally Separable Data

**Hancheng Min**[*]    **Zhihui Zhu**[†]    **René Vidal**[‡]

[*]Institute of Natural Sciences & School of Mathematical Sciences, Shanghai Jiao Tong University
[†]Computer Science and Engineering, Ohio State University
[‡]Electrical and Systems Engineering & Department of Radiology, University of Pennsylvania
`hanchmin@sjtu.edu.cn, zhu.3400@osu.edu, vidalr@upenn.edu`

## Abstract

Among many mysteries behind the success of deep networks lies the exceptional discriminative power of their learned representations as manifested by the intriguing Neural Collapse (NC) phenomenon, where simple feature structures emerge at the last layer of a trained neural network. Prior works on the theoretical understandings of NC have focused on analyzing the optimization landscape of matrix-factorization-like problems by considering the last-layer features as unconstrained free optimization variables and showing that their global minima exhibit NC. In this paper, we show that gradient flow on a two-layer ReLU network for classifying orthogonally separable data provably exhibits NC, thereby advancing prior results in two ways: First, we relax the assumption of unconstrained features, showing the effect of data structure and nonlinear activations on NC characterizations. Second, we reveal the role of the implicit bias of the training dynamics in facilitating the emergence of NC.

## 1   Introduction

Among many mysteries behind the success of deep learning lies the exceptional discriminative power of neural networks as manifested by the intriguing *Neural Collapse* (NC) phenomenon [1], where simple feature structures emerge in the last layer of a trained network. The NC phenomenon is typically characterized by the following three properties (see the top plot in Figure 1):

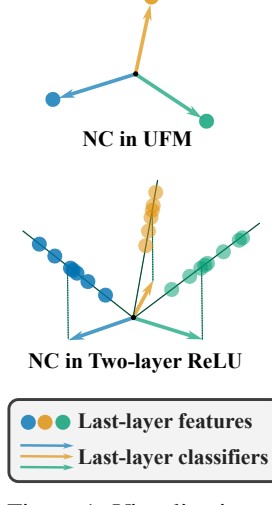

**NC in UFM**

**NC in Two-layer ReLU**

**●●● Last-layer features**
**→→ Last-layer classifiers**

Figure 1: Visualization of the NC phenomenon.

1. *Intra-class variability collapse*: The last-layer feature vectors of the data from the same class collapse into a singleton;
2. *Maximal separation of the class means*: The *class means*, i.e., the mean feature vectors for each class, are maximally separated;
3. *Self-duality*: The classifier weights align with the class means.

Prior works on the theoretical understandings of NC have focused on analyzing the optimization landscape of matrix factorization-like problems by considering the last-layer features as unconstrained free optimization variables [2–9], showing their global minima exhibit NC. Extensions of this so-called *Unconstrained Feature Model* (UFM) include adding nonlinearity and additional hidden layers [7, 8, 10, 11], studying local convergence of gradient-based optimization algorithms around global

---

[*]Corrrespondance to Hancheng Min

39th Conference on Neural Information Processing Systems (NeurIPS 2025).

minima [2, 12, 13]. Until recently, Jacot et al. [14] show convergence towards NC in wide networks. Therefore, the question of how training a neural network leads to NC has remained underexplored.

At the center of this problem lies the fact that practical neural networks are typically overparameterized, i.e., their number of parameters is several orders of magnitude larger than the number of available training examples. As a consequence, there are infinitely many parameter choices that can perfectly fit the data, and most critically, they do not necessarily correspond to a trained network that exhibits NC. However, NC is observed even for networks trained without explicit regularization [3], such as weight decay (often associated with the emergence of NC). This suggests a close relationship between the NC phenomenon and the *implicit bias* [15] of training algorithms.

**Paper contributions**. In this paper, we investigate this relationship between NC and implicit bias by analyzing the dynamics of gradient flow (GF) on a two-layer ReLU network for classification problems, focusing on orthogonally separable data; i.e., any pair of input data with the same or different labels is positively or negatively correlated, respectively. We make the following contributions:

1. In Section 3 we present Theorem 1, which shows that GF with small initialization provably converges to solutions that exhibit NC and provides precise NC characterizations for the trained network, as illustrated in the bottom plot of Figure 1. Compared to prior works for the unconstrained feature model, our results highlight the role of input data and ReLU nonlinearity in determining the NC characteristics. The former causes *intra-class directional collapse* instead of collapse to a singleton, i.e., the feature vectors of the data from the same class collapse into a one-dimensional subspace. The latter leads to *orthogonal class means*, instead of maximally separated class means, and a *projected self-duality* where the classifier weights align with the projected class means (the projection is obtained by subtracting the global mean of all features).
2. In Section 4.1, through our proof of Theorem 1 for the case of binary classification, we explain how the implicit bias of GF facilitates the emergence of NC as training proceeds. Using results on the implicit bias of GF [16–21], we show that in the early phase of training the neurons' directional alignment [16–20] with the training data makes inter-class features mutually orthogonal. Then, during the late phase of training, such inter-class separation subsequently promotes intra-class directional collapse due to the asymptotic max-margin bias [21] of GF.
3. In Section 4.2, in our proof sketch of Theorem 1 for the case of multi-class classification, we extend the aforementioned results on implicit bias for binary problems to multi-class problems. We make technical contributions in addressing new challenges that arise in the dynamic analysis due to the multi-dimensional network output and the cross-entropy loss.

In summary, our work bridges the theoretical analysis of NC and implicit bias of GF by drawing explicit connections between the two. Moreover, we further advance the theoretical understandings for both by highlighting the role of input data and nonlinear activations in NC characterization and by addressing the challenges in analyzing the implicit bias of GF in multi-class problems.

**Notations**. We denote the Euclidean norm of a vector $\boldsymbol{x}$ by $\|\boldsymbol{x}\|$ and its $i$-th entry by $[\boldsymbol{x}]_i$, denote the inner product between vectors $\boldsymbol{x}$ and $\boldsymbol{y}$ by $\langle \boldsymbol{x}, \boldsymbol{y} \rangle = \boldsymbol{x}^\top \boldsymbol{y}$ and write $\boldsymbol{x} \geq \boldsymbol{0}$ or $\boldsymbol{x} > \boldsymbol{0}$ if all the entries of $\boldsymbol{x}$ are non-negative or positive, respectively. For an $n \times m$ matrix $\boldsymbol{A}$, we let $\|\boldsymbol{A}\|_F$ denote the Frobenius norm of $\boldsymbol{A}$. For a scalar-valued or matrix-valued function of time, $\boldsymbol{F}(t)$, we let $\frac{d}{dt}\boldsymbol{F}(t)$ denote its time derivative. We define $\mathbb{1}$ to be the vector of all ones, whose dimension will be clear from the context. We let $\boldsymbol{I}_n$ be the identity matrix of order $n$ and sometimes drop the subscript if its order is clear from the context. We let $[N] := \{1, \cdots, N\}$ and let $\mathbb{S}^{D-1}$ be the unit-sphere in $\mathbb{R}^D$.

## 2 Preliminaries

**Orthogonally separable data**. We consider a classification problem on a dataset $\{\boldsymbol{x}_i, \boldsymbol{y}_i\}_{i=1}^n$ of size $n$, where each *data point* $\boldsymbol{x}_i \in \mathbb{R}^D$ is associated with its *label* $\boldsymbol{y}_i \in \mathbb{R}^{d_y}$, and the number of unique elements in $\{\boldsymbol{y}_i\}_{i=1}^n$ determines the number of classes $K$. Throughout this paper, we assume:

**Assumption 1** (Orthogonal separability)**.** *Any pair of data with the same (different) label(s) are positively (negatively) correlated, i.e., $\exists\, 0 < \mu_s \leq 1$ and $0 < \mu_d \leq \frac{1}{\sqrt{K-1}}$[2] such that $\forall 1 \leq i, j \leq n$,*

$$\left\langle \frac{\boldsymbol{x}_i}{\|\boldsymbol{x}_i\|}, \frac{\boldsymbol{x}_j}{\|\boldsymbol{x}_j\|} \right\rangle \geq \mu_s, \text{ if } \boldsymbol{y}_i = \boldsymbol{y}_j, \qquad \left\langle \frac{\boldsymbol{x}_i}{\|\boldsymbol{x}_i\|}, \frac{\boldsymbol{x}_j}{\|\boldsymbol{x}_j\|} \right\rangle \leq -\mu_d, \text{ if } \boldsymbol{y}_i \neq \boldsymbol{y}_j. \tag{1}$$

[2]No dataset can satisfy orthogonal separability with $\mu_d > \frac{1}{\sqrt{K-1}}$.

**Two-layer ReLU network**. We are interested in solving this classification problem by training a width-$h$ two-layer ReLU network $f(\,\cdot\,;\boldsymbol{\theta}) : \mathbb{R}^D \to \mathbb{R}^{d_y}$, parametrized by $\boldsymbol{\theta} := \{\boldsymbol{W}, \boldsymbol{V}\} \in \mathbb{R}^{D \times h} \times \mathbb{R}^{d_y \times h}$ with $\boldsymbol{W} = [\boldsymbol{w}_1, \cdots, \boldsymbol{w}_h]$ and $\boldsymbol{V} = [\boldsymbol{v}_1, \cdots, \boldsymbol{v}_h]$, and defined as

$$f(\boldsymbol{x}; \boldsymbol{\theta}) = \boldsymbol{V}\sigma\left(\boldsymbol{W}^\top \boldsymbol{x}\right) = \sum_{j=1}^{h} \boldsymbol{v}_j \sigma(\langle \boldsymbol{w}_j, \boldsymbol{x}\rangle), \tag{2}$$

where $\sigma(\cdot) = \max\{\,\cdot\,, 0\}$ is the element-wise ReLU activation function. We consider networks with width $h \geq K$; we call $(\boldsymbol{w}_j, \boldsymbol{v}_j)$ the $j$-th neuron pair in the network, $\boldsymbol{w}_j$ its *input neuron weight* and $\boldsymbol{v}_j$ its *output neuron weight*. Moreover, we let $\boldsymbol{\phi}_{\boldsymbol{\theta}}(\boldsymbol{x}) = [\sigma(\langle \boldsymbol{w}_1, \boldsymbol{x}\rangle), \sigma(\langle \boldsymbol{w}_2, \boldsymbol{x}\rangle), \cdots, \sigma(\langle \boldsymbol{w}_h, \boldsymbol{x}\rangle)]^\top \in \mathbb{R}^h$ be the *last-layer feature* of $\boldsymbol{x}$, and $\boldsymbol{V} = [\boldsymbol{v}_1, \boldsymbol{v}_2, \cdots, \boldsymbol{v}_h] \in \mathbb{R}^{d_y \times h}$ denote the *last-layer classifier*. Note that we have considered bias-free ReLU networks; see the remark in the Appendix A for extending the results to networks with biases.

**Gradient flow with small initialization**. Given some $\ell : \mathbb{R}^{d_y} \times \mathbb{R}^{d_y} \to \mathbb{R}_{\geq 0}$ such that $\ell(\boldsymbol{y}_i, \hat{\boldsymbol{y}}_i)$ (expressions shown in later sections) measures the discrepancy between the actual label $\boldsymbol{y}_i$ and a predicted label $\hat{\boldsymbol{y}}_i$, we let $\mathcal{L}(\boldsymbol{\theta}) = \sum_{i=1}^{n} \ell(\boldsymbol{y}_i, f(\boldsymbol{x}_i; \boldsymbol{\theta}))$ be the *loss function*. We train the network via *gradient flow* (GF), which can be viewed as gradient descent with infinitesimal step size:

$$\frac{\mathrm{d}}{\mathrm{d}t}\boldsymbol{\theta} \in -\partial_{\boldsymbol{\theta}}\mathcal{L}(\boldsymbol{\theta}), \tag{3}$$

where $\partial_{\boldsymbol{\theta}}$ denotes the Clarke sub-differential [22] operator. We study *solutions* (or *trajectories*) $\boldsymbol{\theta}(t)$, $t \geq 0$ that satisfies (3) almost everywhere. We assume that the initialization $\boldsymbol{\theta}(0)$ satisfies

**Assumption 2** ($\epsilon$-small and balanced initialization)**.** *The initialization* $\boldsymbol{\theta}(0) = \{\boldsymbol{w}_j(0), \boldsymbol{v}_j(0)\}_{j=1}^{h}$ *satisfies the following: there exists an* **initialization shape** $\{\boldsymbol{w}_{j0}, \boldsymbol{v}_{j0}\}_{j=1}^{h}$ *with* $\boldsymbol{w}_{j0}, \boldsymbol{v}_{j0} \neq \boldsymbol{0}, \forall j$ *and an* **initialization scale** $\epsilon > 0$ *such that* $\forall j$, $\boldsymbol{w}_j(0) = \epsilon \boldsymbol{w}_{j0}$, $\boldsymbol{v}_j(0) = \epsilon \boldsymbol{v}_{j0}$, $\|\boldsymbol{w}_{j0}\| = \|\boldsymbol{v}_{j0}\|$.

Aside from the two assumptions, we will introduce additional assumptions in different training scenarios when their respective settings become clear. For now, we shall remark on these two.

**Remark 1.** *While the data assumption is strong, there are two main reasons for considering it: First, we investigate NC in shallow networks, whose single hidden layer has limited expressive power of collapsing features, thus one shall study more structured data, as also noted by Hong and Ling [23]. Second, as it will become clearer in Section 4, the emergence of NC is closely related to the asymptotic convergence of the network weights, whose precise characterization is limited to cases with structurally simple data [18, 20, 24]. Nonetheless, as we show in Section 5, simple real data satisfies orthogonal separability approximately, leading to NC characters that match our theorem.*

**Remark 2.** *Under the assumption of balanced initialization, we have* $\|\boldsymbol{w}_j(0)\| = \|\boldsymbol{v}_j(0)\|, \forall j$, *and this balance is maintained throughout the GF trajectories, i.e.,* $\|\boldsymbol{w}_j(t)\| = \|\boldsymbol{v}_j(t)\|, \forall t, \forall j$ *[25]. This assumption of balanced initialization has been common in prior works of this type [18, 20, 24] for the sake of tractable analysis. Our experiments in Section 5 do not require balanced initialization.*

## 3 Main result: Neural Collapse under GF on Two-layer ReLU Networks

Our main result shows that under small initialization, with some additional assumptions on the initialization shape, GF provably converges to neural collapse solutions on orthogonally separable data. Our results are presented for both binary classification and multi-class classification problems:

- **Case one: Binary classification:** We consider binary ($K = 2$) data with scalar $\pm 1$ labels, i.e. $d_y = 1$ and $y_i \in \{-1, +1\}, \forall i$. Accordingly, the two-layer ReLU network $f(\boldsymbol{x}; \boldsymbol{\theta})$ has a scalar output $\hat{y}$. The loss function can be either the exponential loss $\ell(y, \hat{y}) = \exp(-y\hat{y})$ or the logistic loss $\ell(y, \hat{y}) = 2\log(1 + \exp(-y\hat{y}))$. For this case, we use plain font to suggest that label $y$ and network output $\hat{y} = f(\boldsymbol{x}; \boldsymbol{\theta})$ are scalars. Moreover, we define the index sets $\mathcal{I}_+ := \{i : y_i = +1\}$ and $\mathcal{I}_- := \{i : y_i = -1\}$ for $\pm 1$-class data respectively.
- **Case two: Multi-class classification:** We consider multi-class ($K > 2$) data with one-hot labels, i.e., $d_y = K$, and $\boldsymbol{y}_i \in \{\boldsymbol{e}_1, \cdots, \boldsymbol{e}_K\}$, where $\boldsymbol{e}_k$ is the $k$-th column of the identity matrix $\boldsymbol{I}_K$. Accordingly, the two-layer ReLU network has its output $\hat{\boldsymbol{y}} = f(\boldsymbol{x}; \boldsymbol{\theta}) \in \mathbb{R}^K$. The loss function is the Cross-Entropy (CE) loss $\ell(\boldsymbol{y}, \hat{\boldsymbol{y}}) = -\sum_{k=1}^{K}[\boldsymbol{y}]_k \log \frac{\exp([\hat{\boldsymbol{y}}]_k)[\boldsymbol{y}]_k}{\sum_{l=1}^{K} \exp([\hat{\boldsymbol{y}}]_l)}$. Moreover, we define the index sets $\mathcal{I}_k := \{i : \boldsymbol{y}_i = \boldsymbol{e}_k\}, \forall k \in [K]$ for data from each class.

**Main result**. Our main theorem follows. Note that our theorem *requires additional assumptions on the data and initialization shape that vary depending on the case*, thus we feel it is better to introduce and explain them alongside technical discussions on the convergence analysis in later sections.

**Theorem 1** (NC of GF on Two-layer ReLU Networks). *Given orthogonally separable data (Assumption 1), $\epsilon$-small and balanced initialization (Assumption 2) for a sufficiently small $\epsilon$, and some additional case-dependent assumptions on the data and initialization shape, the limit $\bar{\boldsymbol{\theta}} := \lim_{t\to\infty} \frac{\boldsymbol{\theta}(t)}{\|\boldsymbol{\theta}(t)\|_F}$ exists for any solution $\boldsymbol{\theta}(t), t \geq 0$ to (3). Moreover, for the limit $\bar{\boldsymbol{\theta}} = \{\bar{\boldsymbol{W}}, \bar{\boldsymbol{V}}\}$, we have: $\exists \bar{\boldsymbol{\phi}}_k \in \mathbb{S}^{h-1}, k \in \mathcal{K}$, where $\mathcal{K} := \{+, -\}$ (case one) or $\mathcal{K} := [K]$ (case two), such that*

1. *(**Intra-class directional collapse**) The last-layer features of the training data satisfy that*

$$\boldsymbol{\phi}_{\bar{\boldsymbol{\theta}}}(\boldsymbol{x}_i) = \langle s_k \boldsymbol{u}_k, \boldsymbol{x}_i \rangle \cdot \bar{\boldsymbol{\phi}}_k, \ \forall i \in \mathcal{I}_k, k \in \mathcal{K}, \ s_k = \sqrt{\gamma_k^{-1}/(2\textstyle\sum_{k\in\mathcal{K}}\gamma_k^{-1})}; \qquad (4)$$

   *where $\gamma_k = \max_{\boldsymbol{u}\in\mathbb{S}^{D-1}} \min_{i\in\mathcal{I}_k} \langle \boldsymbol{x}_i, \boldsymbol{u} \rangle, k \in \mathcal{K}$ is the maximum margin achieved exclusively for class-$k$ data and $\boldsymbol{u}_k$ is the corresponding max-margin direction;*

2. *(**Orthogonal class means**) The class means directions $\bar{\boldsymbol{\phi}}_k, k \in \mathcal{K}$ satisfy that*

$$\bar{\boldsymbol{\phi}}_k \geq \mathbf{0}, \quad \langle \bar{\boldsymbol{\phi}}_k, \bar{\boldsymbol{\phi}}_{k'} \rangle = 0, \quad \forall k, k' \in \mathcal{K}, k \neq k' ; \qquad (5)$$

3. *(**Projected self-duality**) The last-layer classifier satisfies that*

$$\bar{\boldsymbol{V}} = s_+ \bar{\boldsymbol{\phi}}_+^\top - s_- \bar{\boldsymbol{\phi}}_-^\top (\textbf{case one}) \text{ or } \bar{\boldsymbol{V}} = \sqrt{\tfrac{K}{K-1}} \left(\boldsymbol{I} - \tfrac{1}{K}\mathbb{1}\mathbb{1}^\top\right)\left[s_1\bar{\boldsymbol{\phi}}_1, \cdots, s_K\bar{\boldsymbol{\phi}}_K\right]^\top (\textbf{case two}). \quad (6)$$

Note that GF on positively homogeneous networks with classification losses drives the network weights to diverge to infinity [26, 27]. It suffices to study the properties of the asymptotic weight direction $\bar{\boldsymbol{\theta}}$ since we have $f(\cdot; \boldsymbol{\theta}) = \|\boldsymbol{\theta}\|_F^2 f(\cdot; \frac{\boldsymbol{\theta}}{\|\boldsymbol{\theta}\|_F})$ due to the positive homogeneity of $f$ w.r.t. $\boldsymbol{\theta}$. The following remarks compare the NC characterizations in Theorem 1 with those in prior works.

**NC in two-layer ReLU**. Theorem 1 shows the following NC characters at the late stage of training:

- (*Intra-class directional collapse*) Unlike previous works [2–8], which study unconstrained feature models and show that features collapse to class means with equal length, our work addresses a more realistic and challenging setting involving input data. In this setting, the limited expressiveness of two-layer ReLU networks may prevent exact collapse to a singleton. Nevertheless, the result in (4) shows a *direction collapse* in the sense that all data points in the $k$-th class $\mathcal{I}_k$ have their last-layer features $\phi_{\bar{\boldsymbol{\theta}}}(\boldsymbol{x}_i)$ collapse into a one-dimensional subspace spanned by $\bar{\boldsymbol{\phi}}_k$, though the features may have varying lengths. Consequently, the intra-class variability at the last layer is determined by the variability of projections $\{\langle s_k \boldsymbol{u}_k, \boldsymbol{x}_i \rangle\}_i$, a significant reduction compared to the variability of the original data $\{\boldsymbol{x}_i\}_i$. Moreover, if the features are normalized to unit norm (e.g., by applying RMSnorm [28]), they collapse exactly to their corresponding class means, shedding light on the role of the normalization layer in the neural collapse phenomenon.

- (*Orthogonal class means*) The result (5) suggests that the class-mean features are orthogonal to each other, forming a non-negative orthogonal frame when normalized. The orthogonal structure, rather than a simplex Equiangular Tight Frame (simplex ETF)[3], arises because the features are always non-negative due to ReLU—but any orthogonal frame can be transformed into a simplex ETF by removing its global mean. This finding aligns with results from the unconstrained features model using ReLU as the activation [7].

- (*Projected self-duality*) In the case of binary classification, the classifier $\bar{\boldsymbol{V}}$ converges to $s_+ \bar{\boldsymbol{\phi}}_+ - s_- \bar{\boldsymbol{\phi}}_-$, which yields a maximum margin, as we will show in Section 4.1. For the case of multi-class classification, (6) implies that $\bar{\boldsymbol{V}}\bar{\boldsymbol{V}}^\top = \frac{K}{K-1}\left(\boldsymbol{I} - \tfrac{1}{K}\mathbb{1}\mathbb{1}^\top\right)\bar{\boldsymbol{\Phi}}\bar{\boldsymbol{\Phi}}^\top\left(\boldsymbol{I} - \tfrac{1}{K}\mathbb{1}\mathbb{1}^\top\right)$, where $\bar{\boldsymbol{\Phi}} = \left[s_1\bar{\boldsymbol{\phi}}_1, \cdots, s_K\bar{\boldsymbol{\phi}}_K\right]$. Since $\bar{\boldsymbol{\Phi}}\bar{\boldsymbol{\Phi}}^\top$ is a diagonal matrix with positive diagonals, $\bar{\boldsymbol{V}}$ forms a (scaled) simplex ETF, thereby achieving maximum margin. In particular, when the diagonal scales $s_k, k \in \mathcal{K}$ are all equal, $\bar{\boldsymbol{V}}$ becomes an exact simplex ETF, and each classifier converges to the corresponding *projected class mean* (projection is obtained by subtracting the global mean), up to a scaling factor—achieving self-duality between features and classifiers weights[1].

---

[3]A $K$-simplex ETF in $\mathbb{R}^h$ is a collection of points specified by the columns of $\tilde{\boldsymbol{E}} = \sqrt{\frac{K}{K-1}}\boldsymbol{P}(\boldsymbol{I} - \tfrac{1}{k}\mathbb{1}\mathbb{1}^\top)$, where $\boldsymbol{P} \in \mathbb{R}^{h\times K}$ and $\boldsymbol{P}^\top \boldsymbol{P} = \boldsymbol{I}$.

**Convergence of GF/GD to NC**. Prior works on the convergence of gradient-based methods towards NC consider the mean squared loss [2, 12–14]; Besides, additional conditions such as initialization close to a global optimum [13], weight decay regularization [2, 13, 14], or large width [14] are needed. Compared with these works, we study the convergence under the cross-entropy loss without explicit regularization or width-overparametrization, showing that NC happens under a broader class of problems. Moreover, our results highlight the role of implicit bias of the training algorithm, which we shall discuss next.

# 4 Detailed Discussions: Connecting Neural Collapse with Implicit Bias of GF

## 4.1 Proof of Neural Collapse in Binary Classification

In this section, we provide a proof of Theorem 1 for binary classification of orthogonally separable data. Recall that $\mathcal{I}_+$ and $\mathcal{I}_-$ denote the index sets for data with positive and negative labels, respectively. Let $\mathcal{N}_+(t) := \{j \in [h] : \text{sign}(v_j(t)) = +1\}$ and $\mathcal{N}_-(t) := \{j \in [h] : \text{sign}(v_j(t)) = -1\}$ denote the index set of neurons whose last-layer weights $v_j(t)$ at time $t$ has positive and negative signs, respectively. Under Assumption 2, we have that $\text{sign}(v_j(t)) = \text{sign}(v_j(0))$ [18, 20], thus $\mathcal{N}_+(t) \equiv \mathcal{N}_+(0), \mathcal{N}_-(t) \equiv \mathcal{N}_-(0), \forall t$, and we conveniently let $\mathcal{N}_+ := \mathcal{N}_+(0)$ and $\mathcal{N}_- := \mathcal{N}_-(0)$.

**Alignment phase of the GF**. During the early phase of training, often referred to as *alignment phase*, several works [16–18, 20, 24, 29, 11, 30] have shown that the norm of the neuron weights, which is initially of scale $\mathcal{O}(\epsilon)$, remains small (of scale $\mathcal{O}(\epsilon^{1/2})$) for an extended period of time of length $\Theta(\log \frac{1}{\epsilon})$. As a result, one focuses on understanding the directional dynamics of the input neuron weights during this phase, which can be approximated as follows $\forall j \in [h]$:

$$\frac{\mathrm{d}}{\mathrm{d}t} \frac{\boldsymbol{w}_j}{\|\boldsymbol{w}_j\|} = \text{sign}(v_j)\Pi_{\boldsymbol{w}_j^\perp}\left(\textstyle\sum_{i=1}^n \xi_{ij}\boldsymbol{x}_i y_i + \mathcal{O}(\epsilon)\right), \text{ for some } \xi_{ij} \in \partial_z\sigma(z)|_{z=\langle\boldsymbol{x}_i,\boldsymbol{w}_j\rangle}, \quad (7)$$

where $\Pi_{\boldsymbol{w}_j^\perp} = \left(\boldsymbol{I} - \boldsymbol{w}_j\boldsymbol{w}_j^\top/\|\boldsymbol{w}_j\|^2\right)$ defines the projection onto the subspace orthogonal to $\boldsymbol{w}_j$. If one neglects the $\mathcal{O}(\epsilon)$ term, the dynamics $\frac{\mathrm{d}}{\mathrm{d}t} \frac{\boldsymbol{w}_j}{\|\boldsymbol{w}_j\|}, j \in [h]$ are decoupled. The dynamic behavior of $\frac{\boldsymbol{w}_j}{\|\boldsymbol{w}_j\|}$ critically depends on the stationary points of (7), which we shall address next.

The following discussions assume the $\mathcal{O}(\epsilon)$ error term is zero (only for the sake of the explanation here, the error terms are appropriately handled in the analyses). First of all, the directions $\frac{\boldsymbol{w}_j}{\|\boldsymbol{w}_j\|}$ that render $\xi_{ij} = 0, \forall i \in [n]$ are trivial stationary points of (7), and they form the "dead region" for the neuron as all the activations to the data are zero (as its name suggest, neuron weights within dead region have zero gradient thus receive no update along GF). Next, the stationary points with some $\xi_{ij} \neq 0$ are often called *extremal vectors* [16, 17, 29], and the analyses in the work of Phuong and Lampert [17], Min et al. [20] have suggested that for binary orthogonally separable data the only extremal vectors are the class mean directions: $\bar{\boldsymbol{x}}_+$ and $\bar{\boldsymbol{x}}_-$. More importantly, for neurons with $j \in \mathcal{N}_+$, $\bar{\boldsymbol{x}}_+$ is an attractor, and $\bar{\boldsymbol{x}}_-$ is a repeller[4] (the opposite for $j \in \mathcal{N}_-$). Therefore, by following (7), the neurons weights with $j \in \mathcal{N}_+$ either fall into the dead region, or converge in direction to the average direction $\bar{\boldsymbol{x}}_+$ of the positive class; while those with $j \in \mathcal{N}_-$ either fall into the dead region or converge to $\bar{\boldsymbol{x}}_-$.

**Transient analysis: inter-class separation via alignment dynamics of neurons**. Based on the discussions above, the convergence analysis requires a non-degenerate initialization shape $\{\boldsymbol{w}_{j0}\}_{j=1}^h$ such that 1) *no input neuron weight is initialized to align with the repeller* for (7), since moving away from the repeller can take a long time that cannot be quantified; and 2) *there must exist at least one neuron weight per class that is guaranteed to converge to the average direction of that class*, avoiding the uninteresting case of all neuron weights entering the dead region.

**Assumption 3** (Non-degenerate initialization). *Let $\bar{\boldsymbol{x}}_+ = \frac{\boldsymbol{x}_+}{\|\boldsymbol{x}_+\|}$ and $\bar{\boldsymbol{x}}_- = \frac{\boldsymbol{x}_-}{\|\boldsymbol{x}_-\|}$, where $\boldsymbol{x}_+ := \sum_{i \in \mathcal{I}_+} \boldsymbol{x}_i$ and $\boldsymbol{x}_- := \sum_{i \in \mathcal{I}_-} \boldsymbol{x}_i$. The initialization shape $\{\boldsymbol{w}_{j0}\}_{j=1}^h$ satisfies that $\mathcal{N}_+, \mathcal{N}_- \neq \emptyset$ and*

$$\max_{i \in \mathcal{I}_+, j \in \mathcal{N}_+} \langle\boldsymbol{x}_i, \boldsymbol{w}_{j0}\rangle > 0, \quad \max_{i \in \mathcal{I}_-, j \in \mathcal{N}_-} \langle\boldsymbol{x}_i, \boldsymbol{w}_{j0}\rangle > 0; \quad (8)$$

$$\max_{j \in \mathcal{N}_+} \langle\bar{\boldsymbol{x}}_-, \frac{\boldsymbol{w}_{j0}}{\|\boldsymbol{w}_{j0}\|}\rangle < 1, \quad \max_{j \in \mathcal{N}_-} \langle\bar{\boldsymbol{x}}_+, \frac{\boldsymbol{w}_{j0}}{\|\boldsymbol{w}_{j0}\|}\rangle < 1. \quad (9)$$

---

[4]Roughly speaking, an attractor or a repeller is a stationary point that has the flow around its neighborhood pointing towards or against it, respectively.

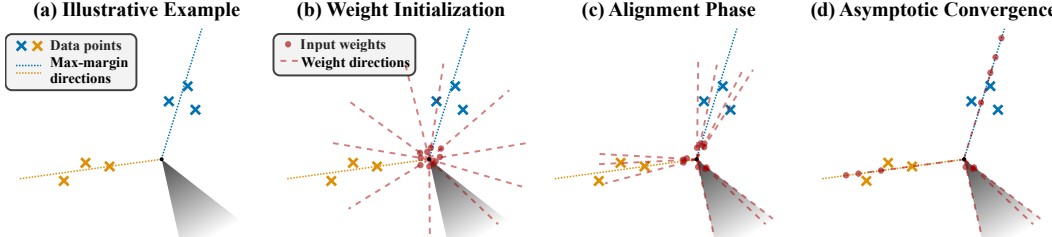

Figure 2: Convergence and implicit bias of GF in two-layer ReLU networks: (a) An example of orthogonally separable data (Assumption 1), gray region indicates the "dead region" for neuron weights: $\{\boldsymbol{w} : \langle \boldsymbol{w}, x_i \rangle \le 0, \forall i \in [n]\}$; (b) Weight initialization following Assumption 2, input neuron weights have small norm and random directions; (c) During alignment phase, inter-class separation is achieved through directional alignment between input neuron weights and data points, as described in (10); (d) Asymptotically, neuron weights diverge to infinity while their directions align with the class-wise max-margin directions, as described in (12).

Given a non-degenerate initialization[5], whenever the neuron weights converge to the vicinity of their respective attracting class averages, they stay close to the class averages for the rest of the training, leading to the following inter-class separation, due to the orthogonally separability of the data:

**Claim** (Inter-class separation via alignment, based the analyses from Phuong and Lampert [17], Min et al. [20]). *Given orthogonally separable data (Assumption 1), $\epsilon$-small, balanced and non-degenerate initialization (Assumptions 2, 3) for a sufficiently small $\epsilon$, for any solution $\boldsymbol{\theta}(t), t \ge 0$ to (3), $\exists T^*$ and $\emptyset \ne \tilde{\mathcal{N}}_+ \subseteq \mathcal{N}_+$ and $\emptyset \ne \tilde{\mathcal{N}}_- \subseteq \mathcal{N}_-$ such that $\forall t \ge T^*$, we have*

$$\boldsymbol{W}_+^\top(t)\boldsymbol{x}_i \begin{cases} > \boldsymbol{0}, & \forall i \in \mathcal{I}_+ \\ \le \boldsymbol{0}, & \forall i \in \mathcal{I}_- \end{cases}, \quad \boldsymbol{W}_-(t)^\top \boldsymbol{x}_i \begin{cases} \le \boldsymbol{0}, & \forall i \in \mathcal{I}_- \\ > \boldsymbol{0}, & \forall i \in \mathcal{I}_+ \end{cases}, \text{ and } \boldsymbol{W}_c(t)^\top \boldsymbol{x}_i \le \boldsymbol{0}, \forall i \in [n], \quad (10)$$

*where $\boldsymbol{W}_+(t) := [\boldsymbol{w}_j(t)]_{j \in \tilde{\mathcal{N}}_+}, \boldsymbol{W}_-(t) := [\boldsymbol{w}_j(t)]_{j \in \tilde{\mathcal{N}}_-}$, and $\boldsymbol{W}_c(t) := [\boldsymbol{w}_j(t)]_{j \in [h] - (\tilde{\mathcal{N}}_+ \cup \tilde{\mathcal{N}}_-)}$.*

*As a result, the **inter-class separation** $\langle \phi_{\boldsymbol{\theta}(t)}(\boldsymbol{x}_i), \phi_{\boldsymbol{\theta}(t)}(\boldsymbol{x}_{i'}) \rangle = 0, \forall i \in \mathcal{I}_+, i' \in \mathcal{I}_-$ holds $\forall t \ge T^*$.*

We refer to Figure 2c for an illustration of the weight-data alignment that achieves inter-class separation. As shown in this claim, all neuron weights with index $j$ outside $\tilde{\mathcal{N}}_+ \cup \tilde{\mathcal{N}}_-$ will stay within the dead region after $T^*$ and can be disregarded in the subsequent analysis. Therefore, without loss of generality, we assume until the end of Section 4.1 that $\tilde{\mathcal{N}}_+ \cup \tilde{\mathcal{N}}_- = [h]$ and reorder the indices such that $\tilde{\mathcal{N}}_+ = [|\tilde{\mathcal{N}}_+|]$ and $\tilde{\mathcal{N}}_- = [h] - [|\tilde{\mathcal{N}}_+|]$. To see that (10) indeed suggests inter-class separation characterized by last-layer features being mutually orthogonal, notice that $\forall t \ge T^*$, we have

$$\phi_{\boldsymbol{\theta}(t)}(\boldsymbol{x}_i) \overset{(\forall i \in \mathcal{I}_+)}{=} \begin{bmatrix} \sigma(\boldsymbol{W}_+^\top(t)\boldsymbol{x}_i) \\ \sigma(\boldsymbol{W}_-^\top(t)\boldsymbol{x}_i) \end{bmatrix} = \begin{bmatrix} \boldsymbol{W}_+^\top(t)\boldsymbol{x}_i \\ \boldsymbol{0} \end{bmatrix}, \quad \phi_{\boldsymbol{\theta}(t)}(\boldsymbol{x}_i) \overset{(\forall i \in \mathcal{I}_+)}{=} \begin{bmatrix} \sigma(\boldsymbol{W}_+^\top(t)\boldsymbol{x}_i) \\ \sigma(\boldsymbol{W}_-^\top(t)\boldsymbol{x}_i) \end{bmatrix} = \begin{bmatrix} \boldsymbol{0} \\ \boldsymbol{W}_-^\top(t)\boldsymbol{x}_i \end{bmatrix}.$$

**Asymptotic analysis: intra-class directional collapse and self-duality via max-margin bias**. Now with the inter-class separation described in (10), we are ready to study the asymptotic convergence of the weights through the lens of max-margin bias. In particular, notice that given inter-class separation (10), we rewrite the loss as (where we define $\boldsymbol{V}_+ := [\boldsymbol{v}_j(t)]_{j \in \mathcal{N}_+}$ and $\boldsymbol{V}_- := [\boldsymbol{v}_j(t)]_{j \in \mathcal{N}_-}$):

$$\mathcal{L}(\boldsymbol{\theta}) = \sum_{i=1}^n \ell(\boldsymbol{y}_i, \boldsymbol{f}(\boldsymbol{x}_i; \boldsymbol{\theta})) = \sum_{i \in \mathcal{I}_+} \ell(\boldsymbol{y}_i, \boldsymbol{V}_+ \boldsymbol{W}_+^\top \boldsymbol{x}_i) + \sum_{i \in \mathcal{I}_-} \ell(\boldsymbol{y}_i, \boldsymbol{V}_- \boldsymbol{W}_-^\top \boldsymbol{x}_i). \quad (11)$$

This suggests that after $T^*$, the GF on $\{\boldsymbol{W}_+, \boldsymbol{V}_+\}$ is fully decoupled from that on $\{\boldsymbol{W}_-, \boldsymbol{V}_-\}$, allowing one to study them separately. Moreover, each of the flows corresponds to training a *two-layer linear network* on positively correlated data with the same label, whose asymptotic convergence of the weight directions has been characterized in the work of Phuong and Lampert [17], mainly based on the analysis from Ji and Telgarsky [21] on the max-margin bias of GF in linear networks.

To be precise, for the GF on losses of the form $\sum_{i \in \mathcal{I}_+} \ell(\boldsymbol{y}_i, \boldsymbol{V}\boldsymbol{W}^\top \boldsymbol{x}_i\}$, Ji and Telgarsky [21] show that as time $t \to \infty$, both $\boldsymbol{V}$ and $\boldsymbol{W}^\top$ diverge to infinity and their limiting directions exist. In

---

[5] Min et al. [20] has shown that the non-degeneracy is satisfied with high probability when the input weight shapes are randomly initialized.

the case of (11), this means $\boldsymbol{\theta} = \{\boldsymbol{W}_+, \boldsymbol{V}_+, \boldsymbol{W}_-, \boldsymbol{V}_-\}$ diverge to infinity, and $\lim_{t\to\infty} \frac{\boldsymbol{\theta}(t)}{\|\boldsymbol{\theta}(t)\|} :=$ $\bar{\boldsymbol{\theta}} = \{\bar{\boldsymbol{W}}_+, \bar{\boldsymbol{V}}_+, \bar{\boldsymbol{W}}_-, \bar{\boldsymbol{V}}_-\}$. Moreover, they show that the limiting directions satisfies the following alignment condition $\bar{\boldsymbol{V}}_+ \bar{\boldsymbol{W}}_+^\top \propto \boldsymbol{u}_+$, the class-wise max-margin direction we have defined in Theorem 1 and balancedness condition $\bar{\boldsymbol{V}}_+^\top \bar{\boldsymbol{V}}_+ = \bar{\boldsymbol{W}}_+^\top \bar{\boldsymbol{W}}_+$ (similar for $\bar{\boldsymbol{W}}_-, \bar{\boldsymbol{V}}_-$, with $\boldsymbol{u}_+$ replaced by $\boldsymbol{u}_-$). It was first found in the work of Phuong and Lampert [17] that the only time these two conditions are satisfied is when $\bar{\boldsymbol{W}}_+^\top$ has rank 1 and the top left and right singular vectors align with $\bar{\boldsymbol{V}}_+$ and $\boldsymbol{u}_+$, respectively. We show that this necessarily implies NC. The formal results are:

**Claim** (Directional collapse and self-duality via max-margin bias, based on the analyses from Phuong and Lampert [17], Ji and Telgarsky [21]). *Given Assumptions 1,2,&3, the limit $\bar{\boldsymbol{\theta}} := \lim_{t\to\infty} \frac{\boldsymbol{\theta}(t)}{\|\boldsymbol{\theta}(t)\|_F}$ exists for any solution $\boldsymbol{\theta}(t), t \geq 0$ to (3). For the limiting direction $\bar{\boldsymbol{\theta}} = \{\bar{\boldsymbol{W}}_+, \bar{\boldsymbol{V}}_+, \bar{\boldsymbol{W}}_-, \bar{\boldsymbol{V}}_-\}$, $\exists \boldsymbol{g}_+ \in \mathbb{S}^{|\mathcal{N}_+|-1}, \boldsymbol{g}_- \in \mathbb{S}^{|\mathcal{N}_-|-1}$, such that*

$$\bar{\boldsymbol{W}}_+ = s_+ \boldsymbol{u}_+ \boldsymbol{g}_+^\top, \ \bar{\boldsymbol{V}}_+ = s_+ \boldsymbol{g}_+^\top, \ \bar{\boldsymbol{W}}_- = s_- \boldsymbol{u}_- \boldsymbol{g}_-^\top, \ \bar{\boldsymbol{V}}_- = -s_- \boldsymbol{g}_-^\top, \quad (12)$$

*where $s_+$ and $s_-$ are defined in Theorem 1. As a result, we have the **intra-class directional collapse***

$$\phi_{\bar{\boldsymbol{\theta}}}(\boldsymbol{x}_i) = \begin{bmatrix} \langle s_+ \boldsymbol{u}_+, \boldsymbol{x}_i \rangle \boldsymbol{g}_+ \\ \boldsymbol{0} \end{bmatrix} = \langle s_+ \boldsymbol{u}_+, \boldsymbol{x}_i \rangle \begin{bmatrix} \boldsymbol{g}_+ \\ \boldsymbol{0} \end{bmatrix} := \langle s_+ \boldsymbol{u}_+, \boldsymbol{x}_i \rangle \cdot \bar{\boldsymbol{\phi}}_+, \forall i \in \mathcal{I}_+, \quad (13)$$

$$\phi_{\bar{\boldsymbol{\theta}}}(\boldsymbol{x}_i) = \begin{bmatrix} \boldsymbol{0} \\ \langle s_- \boldsymbol{u}_-, \boldsymbol{x}_i \rangle \boldsymbol{g}_- \end{bmatrix} = \langle s_- \boldsymbol{u}_-, \boldsymbol{x}_i \rangle \begin{bmatrix} \boldsymbol{0} \\ \boldsymbol{g}_- \end{bmatrix} := \langle s_- \boldsymbol{u}_-, \boldsymbol{x}_i \rangle \cdot \bar{\boldsymbol{\phi}}_-, \forall i \in \mathcal{I}_-, \quad (14)$$

*and the **self-duality** between the last-layer classifier weight and the last-layer feature*

$$\bar{\boldsymbol{V}} = \begin{bmatrix} \bar{\boldsymbol{V}}_+ & \boldsymbol{0} \end{bmatrix} + \begin{bmatrix} \boldsymbol{0} & \bar{\boldsymbol{V}}_- \end{bmatrix} = s_+ \begin{bmatrix} \boldsymbol{g}_+^\top & \boldsymbol{0} \end{bmatrix} - s_- \begin{bmatrix} \boldsymbol{0} & \boldsymbol{g}_-^\top \end{bmatrix} = s_+ \bar{\boldsymbol{\phi}}_+ - s_- \bar{\boldsymbol{\phi}}_-. \quad (15)$$

Note that the scaling factors $s_+, s_-$ are determined based on the results in the work of Lyu and Li [27] that the limiting $\bar{\boldsymbol{\theta}}$ must satisfy another max-margin problem defined on the entire dataset.

**Connecting NC with implicit bias of GF**. In summary, we bridge the theoretical analysis of NC and the implicit bias of GF closer by showing how the latter facilitates the emergence of NC along GF. Notably, the inter-class separation is achieved by the directional alignment of neuron weights thanks to the small initialization scale. This resonates with a large amount of work [31–41] that identifies the small initialization as the active learning or feature learning regime, allowing simple weight and hidden feature structures to arise during the early phase of GF, which otherwise cannot be achieved if initialized in the so-called lazy regime [42–44]. Then, we have shown how the asymptotic max-margin bias promotes the intra-class directional collapse and self-duality after inter-class separation, where the key observation is that the max-margin bias often makes the weights asymptotically converge in direction to low-rank matrices, leading to low-dimensional projections that significantly reduce the variability within the input data. We note that prior work [3] studies the max-margin bias in UFM, while ours considers such bias in ReLU networks.

## 4.2 Proof Sketch of Neural Collapse in Multi-class Classification

As shown in the last section, the proof of the NC characterization for binary classification in Theorem 1 follows from existing results on the implicit bias of GF [17, 20, 21]. However, to understand similar NC characterizations in the case of multi-class classification, one needs to extend the implicit bias analyses to multi-class problems, which prior work rarely does. In this section, we provide a proof sketch of Theorem 1 for multi-class classification, emphasizing the additional challenges it brings to the convergence analysis by considering a multi-dimensional network output and the cross-entropy loss, and discussing our contributions in addressing these challenges.

**Weight alignment in multi-class problems**. Recall that in binary problems, the directional dynamics are studied only for the input weights $\boldsymbol{w}_j, j \in [h]$, because the output weights $v_j$ are scalars whose sign (i.e. the "direction" of the scalar) remains the same as its initialization. However, for multi-class problems, each $\boldsymbol{v}_j$ becomes a $K$-dimensional vector, whose directional dynamics are non-trivial. Indeed, we show that (See Appendix C.2) during the early phase of GF, we have

$$\frac{\mathrm{d}}{\mathrm{d}t} \frac{\boldsymbol{w}_j}{\|\boldsymbol{w}_j\|} = \sqrt{\frac{K-1}{K}} \Pi_{\boldsymbol{w}_j^\perp} \left( \sum_{i=1}^n \xi_{ij} \left\langle \tilde{\boldsymbol{E}} \boldsymbol{y}_i, \frac{\boldsymbol{v}_j}{\|\boldsymbol{v}_j\|} \right\rangle \boldsymbol{x}_i + \mathcal{O}(\epsilon) \right), \quad (16)$$

$$\frac{\mathrm{d}}{\mathrm{d}t} \frac{\boldsymbol{v}_j}{\|\boldsymbol{v}_j\|} = \sqrt{\frac{K-1}{K}} \Pi_{\boldsymbol{v}_j}^\perp \left( \sum_{i=1}^n \xi_{ij} \left\langle \boldsymbol{x}_i, \frac{\boldsymbol{w}_j}{\|\boldsymbol{w}_j\|} \right\rangle \tilde{\boldsymbol{E}} \boldsymbol{y}_i + \mathcal{O}(\epsilon) \right), \quad (17)$$

where $\Pi_{\boldsymbol{w}_j^\perp}, \Pi_{\boldsymbol{v}_j}^\perp$ and $\xi_{ij}$ are defined similarly as in (7), and $\tilde{\boldsymbol{E}} = \sqrt{\frac{K}{K-1}}\left(\boldsymbol{I} - \frac{1}{K}\mathbb{1}\mathbb{1}^\top\right)$. We let $\tilde{\boldsymbol{e}}_k$ be the $k$-th column of $\tilde{\boldsymbol{E}}$ and call it the *pseudo-label* of class $k$, as opposed to the one-hot label $\boldsymbol{e}_k$.

From (16)(17) (still, we exclude the $\mathcal{O}(\epsilon)$ error terms for discussions), we see that although the dynamics $\{\frac{\mathrm{d}}{\mathrm{d}t}\frac{\boldsymbol{w}_j}{\|\boldsymbol{w}_j\|}, \frac{\mathrm{d}}{\mathrm{d}t}\frac{\boldsymbol{v}_j}{\|\boldsymbol{v}_j\|}\}, j \in [h]$ are decoupled among neuron pairs, the directional dynamics of each neuron pair, now concerning both input and output weights, are described by a Riemannian flow on $\mathbb{S}^{D-1} \times \mathbb{S}^{K-1}$ that is highly nonlinear (due to the $\xi_{ij}$) and has non-trivial interactions between the input and output weights. This is the major challenge to the convergence analysis of GF under small initialization for multi-class problems. For our purpose, we discuss the alignment of the neuron weights through the lens of stationary points.

Aside from trivial stationary points that correspond to the dead regions for the input neuron weight, $\{\bar{\boldsymbol{x}}_k, \tilde{\boldsymbol{e}}_k\}, k \in [K]$, where $\bar{\boldsymbol{x}}_k = \sum_{i \in \mathcal{I}_k} \boldsymbol{x}_i / \|\sum_{i \in \mathcal{I}_k} \boldsymbol{x}_i\|$, are attractors of (16)(17). To give a rough explanation, assume $\frac{\boldsymbol{v}_j}{\|\boldsymbol{v}_j\|} = \tilde{\boldsymbol{e}}_k$, i.e. perfect alignment between output weights and the pseudo-label always holds, then one can write (16) as

$$\frac{\mathrm{d}}{\mathrm{d}t}\frac{\boldsymbol{w}_j}{\|\boldsymbol{w}_j\|} = \sqrt{\frac{K-1}{K}}\Pi_{\boldsymbol{w}_j^\perp}\left(\sum_{i \in \mathcal{I}_k} \xi_{ij} \underbrace{\langle \tilde{\boldsymbol{e}}_k, \tilde{\boldsymbol{e}}_k \rangle}_{=1} \boldsymbol{x}_i\right) + \sqrt{\frac{K-1}{K}}\Pi_{\boldsymbol{w}_j^\perp}\left(\sum_{k' \neq k}\sum_{i \in \mathcal{I}_{k'}} \xi_{ij} \underbrace{\langle \tilde{\boldsymbol{e}}_{k'}, \tilde{\boldsymbol{e}}_k \rangle}_{<0} \boldsymbol{x}_i\right),$$

resulting on a flow that pushes $\frac{\boldsymbol{w}_j}{\|\boldsymbol{w}_j\|}$ towards (against) the directions of data points in the $k$-th class (other classes), and eventually towards $\bar{\boldsymbol{x}}_k$ when sufficient alignment with the $k$-th class has led to $\xi_{ij} = \mathbb{1}_{i \in \mathcal{I}_k}$. Similarly, by assuming $\frac{\boldsymbol{w}_j}{\|\boldsymbol{w}_j\|} = \bar{\boldsymbol{x}}_k$, we can write (17) as $\frac{\mathrm{d}}{\mathrm{d}t}\frac{\boldsymbol{v}_j}{\|\boldsymbol{v}_j\|} = \sqrt{\frac{K-1}{K}}\Pi_{\boldsymbol{v}_j}^\perp\left(\sum_{i \in \mathcal{I}_k}\langle \boldsymbol{x}_i, \bar{\boldsymbol{x}}_k \rangle \tilde{\boldsymbol{e}}_k\right)$, pushing the output weight toward the pseudo-label $\tilde{\boldsymbol{e}}_k$.

**Transient analysis: inter-class separation via alignment dynamics of neurons**. Based on the discussion above, there is the *region of attraction* (ROA) for each attractor $\{\bar{\boldsymbol{x}}_k, \tilde{\boldsymbol{e}}_k\}$ such that all the neuron weights initialized within the ROA are guaranteed to converge to $\{\bar{\boldsymbol{x}}_k, \tilde{\boldsymbol{e}}_k\}$ via (16)(17). Moreover, the boundaries between two ROAs of different (class average)-(pseudo-label) pairs together form an invariant set that does not converge to any of the attractors $\{\bar{\boldsymbol{x}}_k, \tilde{\boldsymbol{e}}_k\}$, where there exist saddle points of (16)(17). The exact generalization of Assumption 3 to the multi-class case should assume that no weight direction falls on the aforementioned boundaries and each ROA contains at least one neuron pair. However, finding an analytic expression for the boundaries is a challenging problem by itself and far beyond the scope of this paper. Instead, our analysis identifies an invariant subset for each ROA, thus by initializing within those invariant subsets, we guarantee the directional convergence of the neuron weights to $\{\bar{\boldsymbol{x}}_k, \tilde{\boldsymbol{e}}_k\}, k \in [K]$, which implies inter-class separation.

Formally, we let $\mathcal{I}_k^{\boldsymbol{w}} = \{i \in \mathcal{I}_k : \langle \boldsymbol{x}_i, \frac{\boldsymbol{w}}{\|\boldsymbol{w}\|}\rangle > 0\}$ denote the index set for class-$k$ data points that activates the input neuron weight $\boldsymbol{w}$, let $\mathcal{A}_k^{\boldsymbol{w}} = \sum_{i \in \mathcal{I}_k^{\boldsymbol{w}}}\langle \boldsymbol{x}_i, \frac{\boldsymbol{w}}{\|\boldsymbol{w}\|}\rangle$ be the *aggregate alignment with the $k$-th class* of the input neuron weight $\boldsymbol{w}$, and let $\mathcal{B}_k^{\boldsymbol{v}} = \langle \tilde{\boldsymbol{e}}_k, \frac{\boldsymbol{v}}{\|\boldsymbol{v}\|}\rangle$ be the *alignment with the $k$-th pesudo-label* of the output neuron $\boldsymbol{v}$. Then we define the following:

**Assumption 4** (Semi-local initialization). *The initialization shape $\{\boldsymbol{w}_{j0}, \boldsymbol{v}_{j0}\}_{j=1}^h$ satisfies that $\exists$ a partition $\{\mathcal{N}_k, k \in [K]\}$ of $[h]$, such that $\forall k \in [K]$, we have*

$$|\mathcal{I}_k^{\boldsymbol{w}_{j0}}|^2 > \sum_{k' \neq k}|\mathcal{I}_{k'}^{\boldsymbol{w}_{j0}}|^2, \quad \mathcal{A}_k^{\boldsymbol{w}_{j0}} > 2\sum_{k' \neq k}\mathcal{A}_{k'}^{\boldsymbol{w}_{j0}}, \quad \mathcal{B}_k^{\boldsymbol{v}_{j0}} \geq 1 - \frac{1}{2(K-1)}, \ \forall j \in \mathcal{N}_k. \quad (18)$$

Given some additional assumption on the orthogonal separability of the data (see below), the condition in (18) defines an invariant subset of the ROA of the attractor $\{\bar{\boldsymbol{x}}_k, \tilde{\boldsymbol{e}}_k\}$: any neuron weights initialized to satisfy (18) remains to do so during the alignment phase of GF, while getting attracted by $\{\bar{\boldsymbol{x}}_k, \tilde{\boldsymbol{e}}_k\}$, leading to the desired inter-class separation:

**Proposition 1** (Inter-class separation in multi-class problems). *Let $K > 2$. Given orthogonally separable data (Assumption 1) with $\frac{X_{\max}^2}{X_{\min}^2 \mu_d \mu_s^2} < 2K - 3$, where $X_{\max} := \max_{i \in [n]} \|\boldsymbol{x}_i\|$ and $X_{\min} := \min_{i \in [n]} \|\boldsymbol{x}_i\|$, $\epsilon$-small, balanced and semi-local initialization (Assumptions 2, 4) for a sufficiently small $\epsilon$, for any solution $\boldsymbol{\theta}(t), t \geq 0$ to (3), $\exists T^*$ such that $\forall t \geq T^*$, we have*

$$\boldsymbol{W}_k(t)^\top \boldsymbol{x}_i \begin{cases} > \boldsymbol{0}, & \forall i \in \mathcal{I}_k \\ \leq \boldsymbol{0}, & \forall i \notin \mathcal{I}_k \end{cases}, \forall i \in [n], k \in [K], \text{ where } \boldsymbol{W}_k(t) := [\boldsymbol{w}_j(t)]_{j \in \mathcal{N}_k}. \text{ As a result, the}$$

*inter-class separation $\langle \boldsymbol{\phi}_{\boldsymbol{\theta}(t)}(\boldsymbol{x}_i), \boldsymbol{\phi}_{\boldsymbol{\theta}(t)}(\boldsymbol{x}_{i'})\rangle = 0, \forall i \in \mathcal{I}_k, i' \in \mathcal{I}_{k'}, k \neq k'$ holds $\forall t \geq T^*$.*

**Asymptotic analysis: intra-class directional collapse and projected self-duality via max-margin bias**. Once the inter-class separation is achieved, we can again decompose the loss function as $\mathcal{L}(\boldsymbol{\theta}) = \sum_{k=1}^{K} \sum_{i \in \mathcal{I}_k} \ell_{\mathrm{CE}}(\boldsymbol{y}_i, \boldsymbol{V}_k \boldsymbol{W}_k^\top \boldsymbol{x}_i)$, where $\boldsymbol{V}_k = [\boldsymbol{v}_j]_{j \in \mathcal{N}_k}$, thus it suffices to study the GF on $\{\boldsymbol{W}_k, \boldsymbol{V}_k\}$ for training a two-layer linear network on exclusively on class-$k$ data for each $k \in [K]$ with cross-entropy loss. Another major technical contribution of our analysis is to extend the max-margin results in the work of Phuong and Lampert [17], Ji and Telgarsky [21] to the multi-class problems (albeit under a special case that all data have the same label and are positively correlated), leading to the following asymptotic characterization of GF:

**Proposition 2** (Variability collapse and self-duality in multi-class problems)**.** *Let $K > 2$. Given Assumptions 1,2,&4, the limit $\bar{\boldsymbol{\theta}} := \lim_{t \to \infty} \frac{\boldsymbol{\theta}(t)}{\|\boldsymbol{\theta}(t)\|_F}$ exists for any solution $\boldsymbol{\theta}(t), t \geq 0$ to (3). For the limiting direction $\bar{\boldsymbol{\theta}} = \{\bar{\boldsymbol{W}}_k, \bar{\boldsymbol{V}}_k\}_{k=1}^{K}, \exists \boldsymbol{g}_k \in \mathbb{S}^{|\mathcal{N}_k|-1}$, such that*

$$\bar{\boldsymbol{W}}_k = s_k \boldsymbol{u}_k \boldsymbol{g}_k^\top, \ \bar{\boldsymbol{V}}_k = s_k \tilde{\boldsymbol{e}}_k \boldsymbol{g}_k^\top, \tag{19}$$

*where $s_k, k \in [K]$ are defined in Theorem 1. As a result, the **intra-class directional collapse** (4) and **projected self-duality** (6) hold.*

**Limitations of current analysis**. Aside from the orthogonal separability of the data, for which we have made remarks in Section 2, the convergence analysis for multi-class problems requires a stricter separability condition ($\frac{X_{\max}^2}{X_{\min}^2 \mu_d \mu_s^2} < 2K - 3$), as shown in Proposition 1. This assumption is required to show that the subsets of the parameter space defined in Assumption 4 are invariant under GF. We believe such an assumption is not needed in practice, but our limited understanding of the ROAs and their invariant subsets has led to this additional technical condition to ensure directional convergence. Future research on better characterizations of the ROAs and their invariant sets will naturally relax or even potentially remove this requirement. The additional assumption (Assumption 4) on the initialization shape for multi-class problems is another limitation of our analysis for the early phase of GF, requiring all neuron weights to have decent alignment with one of the (class-average)-(pseudo-label) pair. Relaxing such an assumption necessitates a careful in-depth analysis of the neuron weight alignment dynamics shown in (16)(17), for which we have discussed the underlying challenges, and we leave it as an important future research direction. Nonetheless, we would like point out that: First, our result on asymptotic convergence of the weights is applicable whenever one can show that inter-class separation happens at sometime during the GF, and our transient analysis simply provides one condition under which the separation is guaranteed to happen; Moreover, the semi-local initialization can be satisfied if, instead of random initialization, one initializes all the neuron pair shapes $\{\boldsymbol{w}_{j0}, \boldsymbol{v}_{j0}\}_{j=1}^{h}$ by drawing uniformly from the (data)-(pesudo-label) pairs $\{\boldsymbol{x}_i, \tilde{\boldsymbol{E}} \boldsymbol{y}_i\}_{i=1}^{n}$, which is a practically possible initialization scheme.

# 5    Numerical Experiments

We conduct experiments primarily for the purpose of validating our theoretical results. We first train a two-layer ReLU network for classifying three MNIST [45] digits and visualize the neuron weights alignment at the end of the training, thereby showing the NC characterizations in Theorem 1. Next, based on our remarks on intra-class directional collapse, our Theorem 1 suggests that proper normalization layers such as RMSNorm [28] can potentially lead to a more significant level of NC, and we conduct some preliminary experiments with ResNet [46] to verify this conjecture.

**Validating Theorem 1 in MNIST digits classification**. We train a two-layer ReLU network to classify three MNIST digits $\{0, 1, 2\}$. The experimental details are in Appendix B.1. Figure 3 visualizes the training results: First, we show that the dataset of MNIST digits, centered by the mean digit of the entire dataset, approximately satisfies the orthogonally separable assumption. Then, we visualized the neuron pairs, showing their respective alignment with the data and the pseudo-labels. Moreover, we visualize the top 3 principal components of the last-layer feature of the digits, together with the classifiers, whose structure matches the NC characterizations in Theorem 1.

**Experiments on the role of normalization layers on NC**. Next, we train a modified ResNet18 (by replacing the final linear classifier by a two-layer ReLU classifier) on MNIST and CIFAR10 [47] datasets. In addition, we add a normalization layer (Identity/None, LayerNorm [48], or RMSNorm [28]) before the ReLU classifier and vary the methods for normalization. The experimental details are in Appendix B.2. Figure 4 reports (repeated for 5 runs; mean(line) and std(shade) are

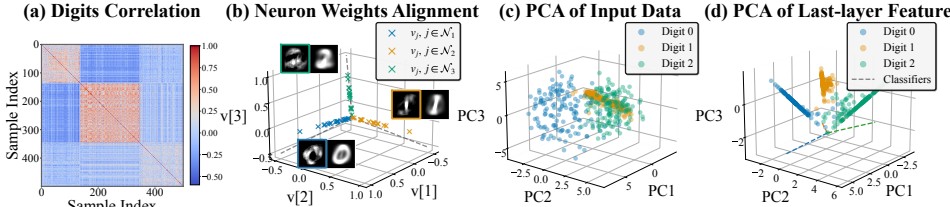

Figure 3: Validating Theorem 1 in classifying MNIST digits $\{0, 1, 2\}$. (a) Normalized correlation matrix of subsampled 500 MNIST digits; (b) (For the trained network) visualization of output neuron weights (as crosses, gray dashed line represents $\tilde{e}_k$ directions for references), the average input neuron weights (as grayscale image, surrounded by colored box), and the average of the digits (for comparison, next to the neuron weights); (c) PCA of raw digits data $X$, keeping the top 3 principal components (1000 points visualized) results in a $\sim 61\%$ relative approximation error for $X$; (d) (For the trained network) PCA of last-layer feature $\phi_\theta(X)$ and classifiers (rows of $V$), keeping the top 3 components (1000 points visualized) results in a $\sim 0.2\%$ relative approximation error for $\phi_\theta(X)$.

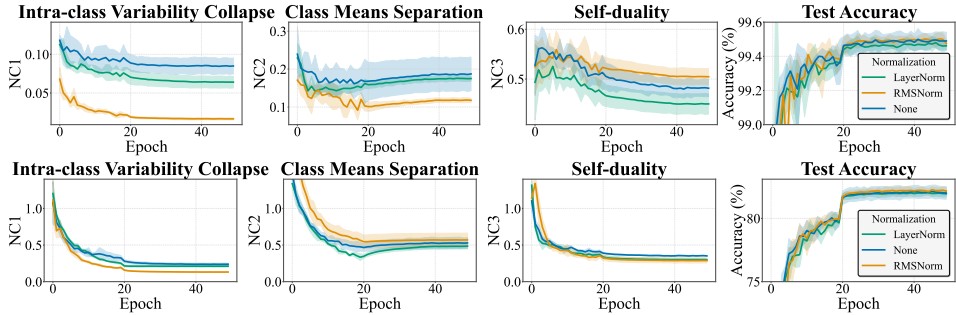

Figure 4: Measuring NC in trained modified ResNet18 on MNIST (top) and CIFAR10 (bottom)

reported) the evolutions over training 50 epochs of the metrics NC1, NC2 and NC3 that measure the NC characteristics in Theorem 1 (lower value implies more prominent NC; definitions in Appendix B.2) at the last-layer of the ReLU classifier. Notably, using the RMSNorm layer significantly improves the intra-class directional collapse, as we conjectured in the remark for Theorem 1, suggesting potential practical value in using RMSNorm layers for promoting NC.

# 6 Conclusion

In this paper, we investigated the connection between NC and the implicit bias of GF through a convergence analysis of GF on two-layer ReLU networks for orthogonally separable data and showed that the implicit bias of GF facilitates the emergence of NC along the GF trajectory. Future work includes relaxing the assumptions on the data and initialization, and extending the convergence analysis to understand the emergence of NC in deeper networks; For example, similar early weight directional alignment and asymptotic max-margin bias have been studied in prior works [30, 26, 27], following the same high-level proof and utilizing these existing results on alignment and max-margin for deep networks might extend the current work to a more practical setting.

## Acknowledgments and Disclosure of Funding

This work is primarily done when H. Min was a postdoc at the University of Pennsylvania. The authors acknowledge the support of the NSF under grants 2031985 and IIS-2312840, the Simons Foundation under grant 814201, the ONR MURI Program under grant 503405-78051, and the University of Pennsylvania Startup Funds.

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

# A Additional Remarks and Related Works

We make the following remark on ReLU networks with biases

**Extension to ReLU nets with biases** When considering two-layer ReLU networks with biases, since $\sigma(\langle \boldsymbol{w}, \boldsymbol{x} \rangle + b) = \sigma(\langle [\boldsymbol{w}; b], [\boldsymbol{x}; 1] \rangle)$, adding a bias term effectively adds one homogenous coordinate to the entire dataset. Therefore, our results still hold if the augmented dataset satisfies the orthogonal separability condition. Notably, homogenous coordinate increases data correlation: $\langle [\boldsymbol{x}_i; 1], [\boldsymbol{x}_j; 1] \rangle = \langle \boldsymbol{x}_i, \boldsymbol{x}_j \rangle + 1$, thus it mainly affects the negative correlation between data points with different labels. In the case when $\min_i \|\boldsymbol{x}_i\| \gg 1$, the orthogonal separability (among augmented data $\{[\boldsymbol{x}_i; 1]\}_{i=1}^n$) still holds with the bias term.

## A.1 Additional related works

**Matrix factorization in deep learning theory** A major part of the research efforts in the theoretical understandings of deep learning is through tractable mathematical problems, where similar phenomena can manifest to those observed in deep learning practice. Among these problems, matrix factorization [49] is an important one, which has been studied for understanding the convergence rate of gradient descent on neural networks [50–54], the implicit bias of neural network training algorithms [32, 55, 21, 31, 33, 37, 36, 56–59], and, as we already mentioned in the introduction, the NC phenomena [2–9]. More recently, the prevalence of LoRA [60] in practice has motivated many works on its theoretical properties [39, 61–63] through analyzing matrix factorization problems. While matrix factorization problems offer many valuable insights into various aspects of deep learning, they generally neglect the role of training data in these problems. For example, the convergence analysis often assumes input data with isotropic covariance [50, 64], and as we have discussed, the analysis of NC often assumes the last-layer features as an unconstrained optimization variable [2–4]. Given that NC is characterizing the ability of neural networks to map input data to structured latent features, it is also important to consider the role of input data. Indeed, our work highlights how the input data structure induces a directional collapse rather than a singleton collapse.

**Gradient descent on shallow neural networks** Theoretical properties of the gradient descent algorithms on shallow networks have been studied in different learning regimes and from various perspectives. Earlier works [65, 66] concern the convergence of gradient descent in the so-called kernel regime, where under specific settings (large network width, random weight initialization with large variance, etc.) the linearization around network weight initialization holds valid throughout training [42, 67]. However, limitations of such "lazy regime" training [43] are identified through some specific student-teacher learning settings [68, 69]. This motivates the study of convergence in small initialization settings, often called *active* or *feature learning regime* [34]. From a dynamics perspective, in the infinite width limit with proper weight initialization scaling, the weight evolution during training can be characterized by some mean-field dynamics [70, 71]; In the finite width with vanishing weight initialization, the early training phase can be characterized by the directional alignment between the weights and the input data [16, 29, 19]. From a generalization perspective, learning in the feature learning regime can enjoy many advantages, such as sample efficiency [72, 73] or benign overfitting [74]. Our work studies the convergence of ReLU networks in the feature learning regime and contributes to this line of work in the following regards: First, prior works primarily concern the convergence of the weights, while our result discusses its implications on the learned last-layer feature. Second, from a technical perspective, our result addresses several challenges emerging from considering a multi-class problem with the cross-entropy loss.

# B Experimental Details

## B.1 Experimental details on classifying MNIST digits

**Preprocessing data**. We first preprocess the training data, i.e. digits $\{0, 1, 2\}$ by centering: $\boldsymbol{x}_i \leftarrow \boldsymbol{x}_i - \bar{\boldsymbol{x}}$, where $\bar{\boldsymbol{x}} = \sum_{i \in [n]} \boldsymbol{x}_i / n$ is the global mean image of the entire training data. Then we have plotted the normalized correlation matrix $\left[ \left\langle \frac{\boldsymbol{x}_i}{\|\boldsymbol{x}_i\|}, \frac{\boldsymbol{x}_{i'}}{\|\boldsymbol{x}_{i'}\|} \right\rangle \right]_{i,i' \in [n]}$ of the centered data, showing in Figure 3(a) that two data points of the same digit are likely to have a positive correlation and those different digits are likely to have negative correlations. This suggests that the orthogonality separability assumption in Assumption 1 is approximately satisfied.

**Training**. Given the centered data, for a two-layer ReLU network (2) of width-50, we initialize all entries of the network weights with i.i.d. Gaussians with variance $10^{-6}$. Then we run SGD of batch size 1000 with learning rate 0.1 for 50 epochs. For the trained network, we visualize the output neuron weights $\boldsymbol{v}_j, j \in [h]$ and determine the $\mathcal{N}_k$ by letting $\mathcal{N}_k = \{j \in [h] : k = \arg\max_{k'} \langle \tilde{\boldsymbol{e}}_{k'}, \boldsymbol{v}_j \rangle\}$, then also visualize the average direction of the input neuron weights $\frac{\sum_{j \in \mathcal{N}_k} \boldsymbol{w}_j}{\| \sum_{j \in \mathcal{N}_k} \boldsymbol{w}_j \|}$ for each group $\mathcal{N}_k$, as shown in Figure 3(b).

## B.2 Experimental details on normalization layers

**Modified ResNet**. We take the ResNet18 and ResNet50 implementations (The first conv layer is modified to accommodate MNIST and CIFAR10 input sizes) in Pytorch and replace the final linear classifier with a two-layer ReLU network of width-1000, and also add a normalization layer (Identity/None, LayerNorm, or RMSNorm) between the classifier and the feature extractor. The initialization follows the Pytorch default.

**Training**. For each choice of (model: ResNet18, ResNet50)-(Dataset: MNIST, CIFAR10), we repeat 5 runs (with different random seeds) of SGD of batch size 128 and learning rate 0.1 (for ResNet18) and 0.02 (for ResNet50) with momentum 0.95 for 50 epochs; and for every 20 epochs, we reduce the learning rate to 0.1 of its current value. We plot the NC metrics and test accuracy against training epochs in Figure 4.

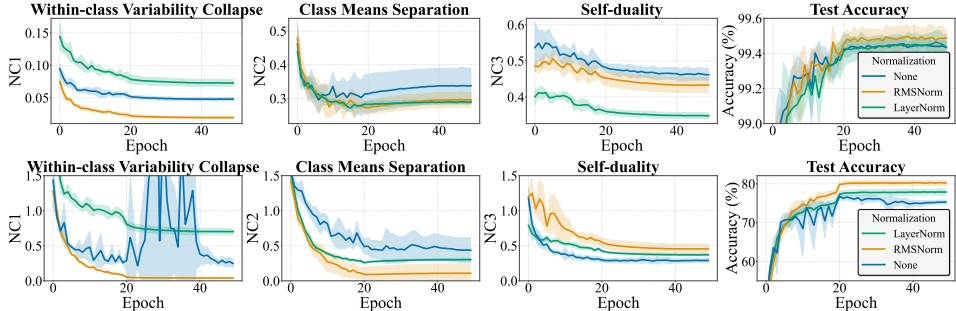

Figure 5: Measuring NC in trained modified ResNet50 on MNIST (top) and CIFAR10 (bottom)

**NC metrics**. The NC metrics follow those used in prior works except for the projected self-duality. Given the class means $\phi_k = \frac{\sum_{i \in \mathcal{I}_k} \phi_{\boldsymbol{\theta}}(\boldsymbol{x}_i)}{|\mathcal{I}_k|}$ and global mean $\bar{\phi} = \sum_{k=1}^{K} \phi_k / K$, NC1 is defined to be the ratio between intra-class variance and the inter-class variance $\frac{\sum_{k=1}^{K} \sum_{i \in \mathcal{I}_k} \|\phi_{\boldsymbol{\theta}}(\boldsymbol{x}_i) - \phi_k\|^2 / |\mathcal{I}_k|}{\sum_{k=1}^{K} \|\phi_k - \bar{\phi}\|^2}$. NC2 is defined to be the proximity of the gram matrix of the class mean directions to the identity matrix $\|\frac{\boldsymbol{G}}{\|\boldsymbol{G}\|_F} - \frac{1}{\sqrt{K}} \boldsymbol{I}\|_F$, where $\boldsymbol{G} = \begin{bmatrix} \bar{\phi}_1 & \cdots & \bar{\phi}_K \end{bmatrix}^\top \begin{bmatrix} \bar{\phi}_1 & \cdots & \bar{\phi}_K \end{bmatrix}$. NC3 is defined to be the proximity of $\boldsymbol{V} \bar{\boldsymbol{\Phi}}^\dagger$ to an identity matrix $\|\frac{\boldsymbol{V} \bar{\boldsymbol{\Phi}}^\dagger}{\|\boldsymbol{V} \bar{\boldsymbol{\Phi}}^\dagger\|_F} - \frac{1}{\sqrt{K-1}} \tilde{\boldsymbol{E}}\|_F$.

In the main paper, we have only provided the plot for ResNet18. We show the plot for ResNet50 in Figure 5.

# C  Neural Alignment under Multi-class Orthogonally Separable Data

## C.1  Basics on neuron dynamics under multi-class problems

The differential inclusion $\dot{\boldsymbol{\theta}} \in -\nabla_{\boldsymbol{\theta}}\mathcal{L}(\boldsymbol{\theta})$ gives rise to the following characterization of the time derivatives of neuron weights $\forall j \in [h]$:

$$\frac{\mathrm{d}}{\mathrm{d}t}\boldsymbol{w}_j = \sum_{i=1}^n \xi_{ij} \langle \boldsymbol{y}_i - \hat{\boldsymbol{y}}_i, \boldsymbol{v}_j \rangle \boldsymbol{x}_i\,, \tag{20}$$

$$\frac{\mathrm{d}}{\mathrm{d}t}\boldsymbol{v}_j = \sum_{i=1}^n \xi_{ij}(\boldsymbol{y}_i - \hat{\boldsymbol{y}}_i)\langle \boldsymbol{x}_i, \boldsymbol{w}_j \rangle\,, \tag{21}$$

$$\text{where } \xi_{ij} \begin{cases} = 1, & \langle \boldsymbol{x}_i, \boldsymbol{w}_j \rangle > 0 \\ \in [0,1], & \langle \boldsymbol{x}_i, \boldsymbol{w}_j \rangle = 0 \\ = 0, & \langle \boldsymbol{x}_i, \boldsymbol{w}_j \rangle < 0 \end{cases}, \ \hat{\boldsymbol{y}}_i = \mathrm{Softmax}(\boldsymbol{f}(\boldsymbol{x}_i; \boldsymbol{\theta}))$$

It will become clear soon that it is convenient to decompose the weight dynamics into those of the weight norm and of the weight direction, for which we use the balancedness that $\|\boldsymbol{w}_j\| \equiv \|\boldsymbol{v}_j\|, \forall j$:

**(weight norm dynamics)**

$$\frac{\mathrm{d}}{\mathrm{d}t}\|\boldsymbol{w}_j\|^2 \left(\text{also } \frac{\mathrm{d}}{\mathrm{d}t}\|\boldsymbol{v}_j\|^2\right) = 2\sum_{i=1}^n \xi_{ij} \langle \boldsymbol{y}_i - \hat{\boldsymbol{y}}_i, \boldsymbol{v}_j \rangle \langle \boldsymbol{x}_i, \boldsymbol{w}_j \rangle$$

$$= 2\sum_{i=1}^n \xi_{ij} \left\langle \boldsymbol{y}_i - \hat{\boldsymbol{y}}_i, \frac{\boldsymbol{v}_j}{\|\boldsymbol{v}_j\|} \right\rangle \left\langle \boldsymbol{x}_i, \frac{\boldsymbol{w}_j}{\|\boldsymbol{w}_j\|} \right\rangle \|\boldsymbol{w}_j\|^2\,; \tag{22}$$

**(input neuron angular dynamics)**

$$\frac{\mathrm{d}}{\mathrm{d}t}\frac{\boldsymbol{w}_j}{\|\boldsymbol{w}_j\|} = \Pi_{\boldsymbol{w}_j}^{\perp} \frac{1}{\|\boldsymbol{w}_j\|} \sum_{i=1}^n \xi_{ij} \langle \boldsymbol{y}_i - \hat{\boldsymbol{y}}_i, \boldsymbol{v}_j \rangle \boldsymbol{x}_i$$

$$= \Pi_{\boldsymbol{w}_j}^{\perp} \sum_{i=1}^n \xi_{ij} \left\langle \boldsymbol{y}_i - \hat{\boldsymbol{y}}_i, \frac{\boldsymbol{v}_j}{\|\boldsymbol{v}_j\|} \right\rangle \boldsymbol{x}_i\,; \tag{23}$$

**(output neuron angular dynamics)**

$$\frac{\mathrm{d}}{\mathrm{d}t}\frac{\boldsymbol{v}_j}{\|\boldsymbol{v}_j\|} = \Pi_{\boldsymbol{v}_j}^{\perp} \frac{1}{\|\boldsymbol{v}_j\|} \sum_{i=1}^n \xi_{ij}(\boldsymbol{y}_i - \hat{\boldsymbol{y}}_i)\langle \boldsymbol{x}_i, \boldsymbol{w}_j \rangle$$

$$= \Pi_{\boldsymbol{v}_j}^{\perp} \sum_{i=1}^n \xi_{ij} \left\langle \boldsymbol{x}_i, \frac{\boldsymbol{w}_j}{\|\boldsymbol{w}_j\|} \right\rangle (\boldsymbol{y}_i - \hat{\boldsymbol{y}}_i)\,, \tag{24}$$

where $\Pi_{\boldsymbol{u}}^{\perp} := \left(\boldsymbol{I} - \frac{\boldsymbol{u}\boldsymbol{u}^{\top}}{\|\boldsymbol{u}\|^2}\right)$ denote the project matrix onto the orthogonal complement of $\boldsymbol{u}$.

By inspecting (22)(23)(24), we note that the dynamic of each neuron pair $(\boldsymbol{w}_j, \boldsymbol{v}_j)$ is almost decoupled from each other except for the interaction through $\hat{\boldsymbol{y}}_i, i \in [n]$. Interestingly, at the early phase of the GF, (we will show that) the norm of the weights remains close to zero, resulting in $\hat{\boldsymbol{y}}_i \simeq \frac{1}{K}\mathbb{1}, \forall i \in [n]$ thus fully decouples the neuron pair dynamics, the precise statement on such an approximation $\hat{\boldsymbol{y}}_i \simeq \frac{1}{K}\mathbb{1}$ is as follow:

**Lemma 1.** $\|\hat{\boldsymbol{y}}_i - \frac{1}{K}\mathbb{1}\| \leq \frac{8}{\sqrt{K}}\|\boldsymbol{f}(\boldsymbol{x}_i; \boldsymbol{\theta})\|$ whenever $\|\boldsymbol{f}(\boldsymbol{x}_i; \boldsymbol{\theta})\| \leq \frac{1}{4}$.

*Proof.* First of all, we have

$$|\exp(z) - 1| = \max\{\exp(z) - 1, 1 - \exp(z)\} = \begin{cases} \exp(|z|) - 1, & z \geq 0 \\ 1 - \exp(-|z|), & z < 0 \end{cases}.$$

We always have $1 - \exp(-|z|) \leq |z|$. Moreover, whenever $|z| \leq 1$, we have $\exp(|z|) - 1 \leq 2|z|$. Therefore, we conclude that

$$|\exp(z) - 1| \leq 2|z|, \ \forall |z| \leq 1\,. \tag{25}$$

With (25), whenever $\|\boldsymbol{f}(\boldsymbol{x}_i;\boldsymbol{\theta})\| \leq 1$, we have

$$\max_k |\exp([\boldsymbol{f}(\boldsymbol{x}_i;\boldsymbol{\theta})]_k) - 1| \leq 2 \max_k |[\boldsymbol{f}(\boldsymbol{x}_i;\boldsymbol{\theta})]_k| \leq 2\|\boldsymbol{f}(\boldsymbol{x}_i;\boldsymbol{\theta})\|. \tag{26}$$

Now we bound $\|\hat{\boldsymbol{y}}_i - \frac{1}{K}\mathbb{1}\|$ using entrywise bound. Notice that

$$
\begin{aligned}
\left|[\hat{\boldsymbol{y}}_i]_k - \tfrac{1}{K}\right| &= \left| \frac{\exp([\boldsymbol{f}(\boldsymbol{x}_i;\boldsymbol{\theta})]_k)}{\sum_{k'=1}^K \exp([\boldsymbol{f}(\boldsymbol{x}_i;\boldsymbol{\theta})]_{k'})} - \tfrac{1}{K}\right| \\
&= \left| \frac{1+(\exp([\boldsymbol{f}(\boldsymbol{x}_i;\boldsymbol{\theta})]_k)-1)}{K+\sum_{k'=1}^K(\exp([\boldsymbol{f}(\boldsymbol{x}_i;\boldsymbol{\theta})]_{k'})-1)} - \tfrac{1}{K}\right| \\
&= \left| \frac{K(\exp([\boldsymbol{f}(\boldsymbol{x}_i;\boldsymbol{\theta})]_k)-1)-\sum_{k'=1}^K(\exp([\boldsymbol{f}(\boldsymbol{x}_i;\boldsymbol{\theta})]_{k'})-1)}{K\left(K+\sum_{k'=1}^K(\exp([\boldsymbol{f}(\boldsymbol{x}_i;\boldsymbol{\theta})]_{k'})-1)\right)}\right| \\
&\leq \frac{K|\exp([\boldsymbol{f}(\boldsymbol{x}_i;\boldsymbol{\theta})]_k)-1|+\sum_{k'=1}^K|\exp([\boldsymbol{f}(\boldsymbol{x}_i;\boldsymbol{\theta})]_{k'})-1|}{K\left(K-\sum_{k'=1}^K|\exp([\boldsymbol{f}(\boldsymbol{x}_i;\boldsymbol{\theta})]_{k'})-1|\right)} \\
&\overset{(26)}{\leq} \frac{4K\|\boldsymbol{f}(\boldsymbol{x}_i;\boldsymbol{\theta})\|}{K(K-2K\|\boldsymbol{f}(\boldsymbol{x}_i;\boldsymbol{\theta})\|)} \overset{(\|\boldsymbol{f}\|\leq\frac14)}{\leq} \frac{8}{K}\|\boldsymbol{f}(\boldsymbol{x}_i;\boldsymbol{\theta})\|.
\end{aligned}
\tag{27}
$$

Finally, we have $\|\hat{\boldsymbol{y}}_i - \frac{1}{K}\mathbb{1}\| \leq \sqrt{K} \max_k \left|[\hat{\boldsymbol{y}}_i]_k - \frac{1}{K}\right| \leq \frac{8}{\sqrt{K}}\|\boldsymbol{f}(\boldsymbol{x}_i;\boldsymbol{\theta})\|.$ $\qquad\square$

### C.2 Analyzing neuron dynamics during alignment phase

In this section, we show the formal statements for the alignment dynamics we have introduced in (16)(17). During the early phase of the GF training, the norms of the weights remain small (Lemma 2), leading to an approximate alignment dynamics in Lemma 3, which will be crucial for subsequent analysis.

**Lemma 2.** *Given some balanced, $\epsilon$-small initialization $\boldsymbol{\theta}(0)$ with $\epsilon \leq \frac{\sqrt{K}}{16X_{\max}\sqrt{h}}$, any solution $\boldsymbol{\theta}(t)$ to the GF dynamics (3) satisfies that $\forall t \leq \frac{1}{4nX_{\max}}\log\frac{1}{\sqrt{h}\epsilon} := T$,*

$$\|\boldsymbol{w}_j(t)\|^2 = \|\boldsymbol{v}_j(t)\|^2 \leq \frac{\epsilon}{\sqrt{h}}, \quad \forall j \in [h], \quad \|\boldsymbol{f}(\boldsymbol{x}_i;\boldsymbol{\theta}(t))\| \leq 2\epsilon X_{\max}\sqrt{h}. \tag{28}$$

The alignment phase refers to the training phase until $T = \frac{1}{4nX_{\max}}\log\frac{1}{\sqrt{h}\epsilon}$. With Lemma (2), we can approximate the angular dynamics $\frac{\mathrm{d}}{\mathrm{d}t}\frac{\boldsymbol{w}_j}{\|\boldsymbol{w}_j\|}$ and $\frac{\mathrm{d}}{\mathrm{d}t}\frac{\boldsymbol{v}_j}{\|\boldsymbol{v}_j\|}$ throughout the alignment phase as follow:

**Lemma 3.** *Given some balanced, $\epsilon$-small initialization $\boldsymbol{\theta}(0)$ with $\epsilon \leq \frac{1}{16X_{\max}\sqrt{h}}$, any solution $\boldsymbol{\theta}(t)$ to the GF dynamics (3) satisfies that $\forall t \leq \frac{1}{4nX_{\max}}\log\frac{1}{\sqrt{h}\epsilon} := T$,*

$$\left\| \frac{\mathrm{d}}{\mathrm{d}t}\frac{\boldsymbol{w}_j}{\|\boldsymbol{w}_j\|} - \Pi^{\perp}_{\boldsymbol{w}_j}\left( \sqrt{\frac{K-1}{K}} \sum_{i=1}^n \xi_{ij} \left\langle \tilde{\boldsymbol{E}}\boldsymbol{y}_i, \frac{\boldsymbol{v}_j}{\|\boldsymbol{v}_j\|}\right\rangle \boldsymbol{x}_i \right)\right\| \leq \frac{16}{\sqrt{K}}\epsilon n X_{\max}\sqrt{h}, \; \forall j \in [h],$$

$$\left\| \frac{\mathrm{d}}{\mathrm{d}t}\frac{\boldsymbol{v}_j}{\|\boldsymbol{v}_j\|} - \Pi^{\perp}_{\boldsymbol{v}_j}\left( \sqrt{\frac{K-1}{K}} \sum_{i=1}^n \xi_{ij} \left\langle \boldsymbol{x}_i, \frac{\boldsymbol{w}_j}{\|\boldsymbol{w}_j\|}\right\rangle \tilde{\boldsymbol{E}}\boldsymbol{y}_i \right)\right\| \leq \frac{16}{\sqrt{K}}\epsilon n X_{\max}\sqrt{h}, \; \forall j \in [h]$$

*Proof of Lemma 2.* From Section C.1, we have

$$\frac{\mathrm{d}}{\mathrm{d}t}\|\boldsymbol{w}_j\|^2 = 2\sum_{i=1}^n \xi_{ij} \left\langle \boldsymbol{y}_i - \hat{\boldsymbol{y}}_i, \frac{\boldsymbol{v}_j}{\|\boldsymbol{v}_j\|}\right\rangle \left\langle \boldsymbol{x}_i, \frac{\boldsymbol{w}_j}{\|\boldsymbol{w}_j\|}\right\rangle \|\boldsymbol{w}_j\|^2 \tag{29}$$

Let $T := \inf\{t: \max_i |\boldsymbol{f}(\boldsymbol{x}_i;\boldsymbol{\theta}(t))| > 2\epsilon X_{\max}\sqrt{h}\}$, then $\forall t \leq T, j \in [h]$, we have

$$
\begin{aligned}
\frac{\mathrm{d}}{\mathrm{d}t}\|\boldsymbol{w}_j\|^2 &= 2\sum_{i=1}^n \xi_{ij} \left\langle \boldsymbol{y}_i - \hat{\boldsymbol{y}}_i, \frac{\boldsymbol{v}_j}{\|\boldsymbol{v}_j\|}\right\rangle \left\langle \boldsymbol{x}_i, \frac{\boldsymbol{w}_j}{\|\boldsymbol{w}_j\|}\right\rangle \|\boldsymbol{w}_j\|^2 \\
&\leq 2\sum_{i=1}^n |\xi_{ij}| \left|\left\langle \boldsymbol{y}_i - \hat{\boldsymbol{y}}_i, \frac{\boldsymbol{v}_j}{\|\boldsymbol{v}_j\|}\right\rangle\right| \left|\left\langle \boldsymbol{x}_i, \frac{\boldsymbol{w}_j}{\|\boldsymbol{w}_j\|}\right\rangle\right| \|\boldsymbol{w}_j\|^2
\end{aligned}
$$

$$\le 2 \sum_{i=1}^{n} \|\boldsymbol{y}_i - \hat{\boldsymbol{y}}_i\| \|\boldsymbol{x}_i\| \|\boldsymbol{w}_j\|^2$$

$$\le 2 \sum_{i=1}^{n} \left( \left\| \boldsymbol{y}_i - \tfrac{1}{K}\mathbb{1} \right\| + \left\| \tfrac{1}{K}\mathbb{1} - \hat{\boldsymbol{y}}_i \right\| \right) X_{\max} \|\boldsymbol{w}_j\|^2$$

$$\le 2 \sum_{i=1}^{n} \left( \sqrt{\tfrac{K-1}{K}} + \tfrac{8}{\sqrt{K}} \|\boldsymbol{f}(\boldsymbol{x}_i; \boldsymbol{\theta})\| \right) X_{\max} \|\boldsymbol{w}_j\|^2$$

$$\le 2 \sum_{i=1}^{n} \left( X_{\max} + \tfrac{16\epsilon X_{\max}^2 \sqrt{h}}{\sqrt{K}} \right) \|\boldsymbol{w}_j\|^2 \le 2n \left( X_{\max} + \tfrac{16\epsilon X_{\max}^2 \sqrt{h}}{\sqrt{K}} \right) \|\boldsymbol{w}_j\|^2 \qquad (30)$$

Let $\tau_j := \inf\{t : \|\boldsymbol{w}_j(t)\|^2 > \tfrac{\epsilon}{\sqrt{h}}\}$, and let $j^* := \arg\min_j \tau_j$. Then $\tau_{j^*} = \min_j \tau_j \le T$, which can be shown by contradiction:

Suppose $\tau_{j^*} > T$, then at $t = T < \tau_{j^*}$, by the definition of $\tau_{j^*}$, we have $\max_j \|\boldsymbol{w}_j\|^2 \le \tfrac{\epsilon}{\sqrt{h}}$, and by the definition of $T$ and the continuity of $\boldsymbol{\theta}(t)$ w.r.t. $t$, $\exists i^* \in [n]$ such that $|\boldsymbol{f}(\boldsymbol{x}_{i^*}; \boldsymbol{\theta}(T))| = 2\epsilon X_{\max} \sqrt{h}$, therefore,

$$2\epsilon X_{\max} \sqrt{h} = |\boldsymbol{f}(\boldsymbol{x}_{i^*}; \boldsymbol{\theta}(T))| = \left| \sum_{j \in [h]} \xi_{i^* j} \boldsymbol{v}_j \langle \boldsymbol{w}_j, \boldsymbol{x}_{i^*} \rangle \right|$$

$$\le \sum_{j \in [h]} |\xi_{i^* j}| \|\boldsymbol{v}_j\| \|\boldsymbol{w}_j\| \|\boldsymbol{x}_{i^*}\|$$

$$\le \sum_{j \in [h]} X_{\max} \|\boldsymbol{w}_j\|^2 \le h X_{\max} \max_{j \in [h]} \|\boldsymbol{w}_j\|^2, \qquad (31)$$

which suggests that $\max_{j \in [h]} \|\boldsymbol{w}_j\|^2 \ge \tfrac{2\epsilon}{\sqrt{h}}$, a contradiction.

Now for $t \le \tau_{j^*} \le T$, we have

$$\frac{\mathrm{d}}{\mathrm{d}t} \|\boldsymbol{w}_{j^*}\|^2 \le 2n \left( X_{\max} + \tfrac{16\epsilon X_{\max}^2 \sqrt{h}}{\sqrt{K}} \right) \|\boldsymbol{w}_{j^*}\|^2. \qquad (32)$$

By Grönwall's inequality, we have $\forall t \le \tau_{j^*}$

$$\|\boldsymbol{w}_{j^*}(t)\|^2 \le \exp\left( 2n \left( X_{\max} + \tfrac{16\epsilon X_{\max}^2 \sqrt{h}}{\sqrt{K}} \right) t \right) \|\boldsymbol{w}_{j^*}(0)\|^2 \le \exp\left( 2n \left( X_{\max} + \tfrac{16\epsilon X_{\max}^2 \sqrt{h}}{\sqrt{K}} \right) t \right) \epsilon^2.$$

Suppose $\tau_{j^*} < \tfrac{1}{4nX_{\max}} \log \tfrac{1}{\sqrt{h}\epsilon}$, then by the continuity of $\|\boldsymbol{w}_{j^*}(t)\|^2$, we have

$$\frac{\epsilon}{\sqrt{h}} = \|\boldsymbol{w}_{j^*}(\tau_{j^*})\|^2 \le \exp\left( 2n \left( X_{\max} + \tfrac{16\epsilon X_{\max}^2 \sqrt{h}}{\sqrt{K}} \right) \tau_{j^*} \right) \epsilon^2$$

$$< \exp\left( 2n \left( X_{\max} + \tfrac{16\epsilon X_{\max}^2 \sqrt{h}}{\sqrt{K}} \right) \tfrac{1}{4nX_{\max}} \log \tfrac{1}{\sqrt{h}\epsilon} \right) \epsilon^2$$

$$\le \exp\left( \left( \tfrac{1}{2} + \tfrac{8\epsilon X_{\max} \sqrt{h}}{\sqrt{K}} \right) \log \tfrac{1}{\sqrt{h}\epsilon} \right) \epsilon^2$$

$$\le \exp\left( \log \tfrac{1}{\sqrt{h}\epsilon} \right) \epsilon^2 = \tfrac{\epsilon}{\sqrt{h}},$$

where the last inequality is due to $\epsilon \le \tfrac{\sqrt{K}}{16 X_{\max} \sqrt{h}}$. This leads to a contradiction. Therefore, one must have $T \ge \tau_{j^*} \ge \tfrac{1}{4nX_{\max}} \log \left( \tfrac{1}{\sqrt{h}\epsilon} \right)$. This finishes the proof. $\qquad \square$

*Proof of Lemma 3.* We have shown in Section C.1 that

$$\frac{\mathrm{d}}{\mathrm{d}t} \frac{\boldsymbol{w}_j}{\|\boldsymbol{w}_j\|} = \Pi_{\boldsymbol{w}_j}^{\perp} \left( \sum_{i=1}^{n} \xi_{ij} \left\langle \boldsymbol{y}_i - \hat{\boldsymbol{y}}_i, \frac{\boldsymbol{v}_j}{\|\boldsymbol{v}_j\|} \right\rangle \boldsymbol{x}_i \right). \qquad (33)$$

Therefore, $\forall \le T$,

$\frac{\mathrm{d}}{\mathrm{d}t} \frac{\boldsymbol{w}_j}{\|\boldsymbol{w}_j\|} = \Pi_{\boldsymbol{w}_j}^{\perp} \left( \sum_{i=1}^{n} \xi_{ij} \left\langle \boldsymbol{y}_i - \hat{\boldsymbol{y}}_i, \frac{\boldsymbol{v}_j}{\|\boldsymbol{v}_j\|} \right\rangle \boldsymbol{x}_i \right)$

$$= \Pi^{\perp}_{\boldsymbol{w}_j} \left( \sum_{i=1}^n \xi_{ij} \left\langle \underbrace{(\boldsymbol{y}_i - \tfrac{1}{K}\mathbb{1})}_{=\sqrt{\frac{K-1}{K}}\tilde{\boldsymbol{E}}\boldsymbol{y}_i} + (\tfrac{1}{K}\mathbb{1} - \hat{\boldsymbol{y}}_i), \tfrac{\boldsymbol{v}_j}{\|\boldsymbol{v}_j\|} \right\rangle \boldsymbol{x}_i \right)$$

$$= \Pi^{\perp}_{\boldsymbol{w}_j} \left( \sqrt{\tfrac{K-1}{K}} \sum_{i=1}^n \xi_{ij} \left\langle \tilde{\boldsymbol{E}}\boldsymbol{y}_i, \tfrac{\boldsymbol{v}_j}{\|\boldsymbol{v}_j\|} \right\rangle \boldsymbol{x}_i \right) + \Pi^{\perp}_{\boldsymbol{w}_j} \left( \sum_{i=1}^n \xi_{ij} \left\langle (\tfrac{1}{K}\mathbb{1} - \hat{\boldsymbol{y}}_i), \tfrac{\boldsymbol{v}_j}{\|\boldsymbol{v}_j\|} \right\rangle \boldsymbol{x}_i \right).$$

Finally, we have

$$\left\| \tfrac{\mathrm{d}}{\mathrm{d}t} \tfrac{\boldsymbol{w}_j}{\|\boldsymbol{w}_j\|} - \Pi^{\perp}_{\boldsymbol{w}_j} \left( \sqrt{\tfrac{K-1}{K}} \sum_{i=1}^n \xi_{ij} \left\langle \tilde{\boldsymbol{E}}\boldsymbol{y}_i, \tfrac{\boldsymbol{v}_j}{\|\boldsymbol{v}_j\|} \right\rangle \boldsymbol{x}_i \right) \right\|$$

$$= \left\| \Pi^{\perp}_{\boldsymbol{w}_j} \left( \sum_{i=1}^n \xi_{ij} \left\langle (\tfrac{1}{K}\mathbb{1} - \hat{\boldsymbol{y}}_i), \tfrac{\boldsymbol{v}_j}{\|\boldsymbol{v}_j\|} \right\rangle \boldsymbol{x}_i \right) \right\|$$

$$\leq \sum_{i=1}^n |\xi_{ij}| \|\boldsymbol{x}_i\| \|\tfrac{1}{K}\mathbb{1} - \hat{\boldsymbol{y}}_i\| \overset{\text{(Lemma 1)}}{\leq} \tfrac{8}{\sqrt{K}} n X_{\max} \|\boldsymbol{f}(\boldsymbol{x}_i; \boldsymbol{\theta})\| \overset{\text{(Lemma 2)}}{\leq} \tfrac{16}{\sqrt{K}} \epsilon n X_{\max}^2 \sqrt{h}, \quad (34)$$

where we note that applying Lemma 1 requires $\|\boldsymbol{f}(\boldsymbol{x}_i; \boldsymbol{\theta})\| \leq \tfrac{1}{4}$, which is guaranteed by Lemma 2 and our choice $\epsilon \leq \tfrac{1}{16 X_{\max}\sqrt{h}}$. We have shown the approximation error bound for $\tfrac{\mathrm{d}}{\mathrm{d}t} \tfrac{\boldsymbol{w}_j}{\|\boldsymbol{w}_j\|}$. A similar bound can be derived for $\tfrac{\mathrm{d}}{\mathrm{d}t} \tfrac{\boldsymbol{v}_j}{\|\boldsymbol{v}_j\|}$. $\qquad\square$

### C.3 Neural alignment under multi-class orthogonally separable data

**Sufficient statement for Proposition 1**. It is easy to check that the following proposition is sufficient for Proposition 1 to hold.

**Proposition 3** (Sufficient statement for Proposition 1). *Let $K > 2$. Given orthogonally separable data (Assumption 1) with $\tfrac{X_{\max}^2}{X_{\min}^2 \mu_d \mu_s^2} < 2K - 3$, where $X_{\max} := \max_{i \in [n]} \|\boldsymbol{x}_i\|$ and $X_{\min} := \min_{i \in [n]} \|\boldsymbol{x}_i\|$, $\epsilon$-small, balanced and semi-local initialization (Assumptions 2, 4) for a sufficiently small $\epsilon$, for any solution $\boldsymbol{\theta}(t), t \geq 0$ to (3) and any $j \in \mathcal{N}_k, k \in [K]$, define*

$$T_{j,k}^* = \inf\{t \geq 0 : |\mathcal{I}_k^{\boldsymbol{w}_j}| = |\mathcal{I}_k|, |\mathcal{I}_{k'}^{\boldsymbol{w}_j}| = 0, \forall k' \neq k\}, \quad (35)$$

*then $\forall t \geq T_{j,k}^*$, we have $|\mathcal{I}_k^{\boldsymbol{w}_j}| = |\mathcal{I}_k|, |\mathcal{I}_{k'}^{\boldsymbol{w}_j}| = 0, \forall k' \neq k$, thus $\boldsymbol{w}_k(t)^\top \boldsymbol{x}_i \begin{cases} > 0, & \forall i \in \mathcal{I}_k \\ < 0, & \forall i \notin \mathcal{I}_k \end{cases}, \forall i \in [n]$.*

**Therefore, we can study the dynamic behavior of each neuron pair individually, for convenience, let $j \in \mathcal{N}_k$, and we drop the index $j$.**

For a neuron pair $(\boldsymbol{w}, \boldsymbol{v})$, we have defined the following:

$$\alpha_i := \left\langle \boldsymbol{x}_i, \tfrac{\boldsymbol{w}}{\|\boldsymbol{w}\|} \right\rangle, \qquad \text{(alignment between input neuron and $i$-th data)}$$

$$\beta_i := \left\langle \tilde{\boldsymbol{E}}\boldsymbol{y}_i, \tfrac{\boldsymbol{v}}{\|\boldsymbol{v}\|} \right\rangle, \qquad \text{(alignment between output neuron and $i$-th label)}$$

$$\mathcal{I}_k^{\boldsymbol{w}} := \{i \in \mathcal{I}_k : \alpha_i > 0\}, \qquad \text{(number of active data points in $k$-th class)}$$

and

$$\mathcal{A}_k := \sum_{i \in \mathcal{I}_k^{\boldsymbol{w}}} \alpha_i = \underbrace{\sum_{i \in \mathcal{I}_k : \alpha_i > 0} \alpha_i}_{\text{we mostly use this notation for clarity}} , \quad \text{(alignment between input neuron and $k$-th class)}$$

$$\mathcal{B}_k := \left\langle \tilde{\boldsymbol{e}}_k, \tfrac{\boldsymbol{v}}{\|\boldsymbol{v}\|} \right\rangle. \qquad \text{(alignment between output neuron and $k$-th class)}$$

**Overview of the proof of Proposition 1**. First, we utilize the alignment dynamics in Lemma 3, to show that (Recall that $T = \tfrac{1}{4n X_{\max}} \log \tfrac{1}{\sqrt{h}\epsilon}$)

**Lemma 4.** *Given a neuron $(\boldsymbol{w}_j, \boldsymbol{v}_j), j \in \mathcal{N}_k$, during the alignment phase $t \leq \min\{T_{j,k}^*, T\}$, the following holds:*

1. $\frac{\boldsymbol{v}_j}{\|\boldsymbol{v}_j\|}$ remains close to its target pseudo-label: $\mathcal{B}_k \geq 1 - \frac{1}{2(K-1)}$;

2. $\frac{\boldsymbol{w}_j}{\|\boldsymbol{w}_j\|}$ remains close to its target class: $\mathcal{A}_k - 2\sum_{k' \neq k} \mathcal{A}_{k'} \geq \mathcal{A}_k(0) - 2\sum_{k' \neq k} \mathcal{A}_{k'}(0)$;

3. Neuron $\frac{\boldsymbol{w}_j}{\|\boldsymbol{w}_j\|}$ does not deactivate data from target class, nor activate data from non-target class:
$|\mathcal{I}_k^{\boldsymbol{w}}| \geq |\mathcal{I}_k^{\boldsymbol{w}(0)}|$ and $|\mathcal{I}_{k'}^{\boldsymbol{w}}| \leq |\mathcal{I}_{k'}^{\boldsymbol{w}(0)}|, \forall k' \neq k$.

The characterizations in 4 suggest that the neuron weight directions $\{\frac{\boldsymbol{w}_j}{\|\boldsymbol{w}_j\|}, \frac{\boldsymbol{v}_j}{\|\boldsymbol{v}_j\|}\}$ remains close to the attractor $\{\bar{\boldsymbol{x}}_k, \tilde{\boldsymbol{e}}_k\}$, and as the weights move closer to the attractor, $|\mathcal{I}_k^{\boldsymbol{w}_j}|$ increases to $|\mathcal{I}_k|$, and $|\mathcal{I}_{k'}^{\boldsymbol{w}_j}|$ decreases to 0. When the initialization scale $\epsilon$ is sufficiently small so that $T$ is large, this Lemma will show that $T_{j,k}^*$ is finite, and we will provide an upper bound.

Then the following lemma shows that the desired property for neuron $(\boldsymbol{w}_j, \boldsymbol{v}_j)$ still holds after $T_{j,k}^*$.

**Lemma 5.** *for any neuron* $(\boldsymbol{v}_j, \boldsymbol{w}_j), j \in \mathcal{N}_k$, *we have* $\forall t > T_{j,k}^*$:

1. $\frac{\boldsymbol{v}_j}{\|\boldsymbol{v}_j\|}$ *remains close to its target pseudo-label:* $\mathcal{B}_k^{\boldsymbol{v}_j} \geq \frac{\sqrt{2}}{2}$;

2. $\frac{\boldsymbol{w}_j}{\|\boldsymbol{w}_j\|}$ *is exclusively activated by data from its target class:* $|\mathcal{I}_k^{\boldsymbol{w}_j}| = N_k, |\mathcal{I}_{k'}^{\boldsymbol{w}_j}| = 0, \forall k' \neq k$.

The remaining parts of this section are dedicated to proving these two Lemmas. The next section will formally prove Proposition 3, thereby proving Proposition 1.

### C.3.1 Proof of Lemma 4

**Basic dynamics**. The main proof concerns the time derivatives of the alignment to classes $\frac{\mathrm{d}}{\mathrm{d}t}\mathcal{A}_k, \frac{\mathrm{d}}{\mathrm{d}t}\mathcal{B}_k$. With Lemma 3, we have their approximations during the alignment phase:

$$
\begin{aligned}
\frac{\mathrm{d}}{\mathrm{d}t}\mathcal{A}_k &= \sum_{i \in \mathcal{I}_k : \alpha_i > 0} \frac{\mathrm{d}}{\mathrm{d}t}\alpha_i \\
&= \sum_{i \in \mathcal{I}_k : \alpha_i > 0} \left\langle \boldsymbol{x}_i, \frac{\mathrm{d}}{\mathrm{d}t}\frac{\boldsymbol{w}}{\|\boldsymbol{w}\|} \right\rangle \\
&= \sum_{i \in \mathcal{I}_k : \alpha_i > 0} \left\langle \boldsymbol{x}_i, \Pi_{\boldsymbol{w}}^{\perp} \left( \sqrt{\tfrac{K-1}{K}} \sum_{i'=1}^{n} \xi_{i'} \left\langle \tilde{\boldsymbol{E}}\boldsymbol{y}_{i'}, \frac{\boldsymbol{v}}{\|\boldsymbol{v}\|} \right\rangle \boldsymbol{x}_{i'} \right) \right\rangle + \mathcal{O}(\epsilon) && (36) \\
&= \sqrt{\tfrac{K-1}{K}} \sum_{i \in \mathcal{I}_k : \alpha_i > 0} \left\langle \boldsymbol{x}_i, \Pi_{\boldsymbol{w}}^{\perp} \left( \sum_{i'=1}^{n} \xi_{i'} \beta_{i'} \boldsymbol{x}_{i'} \right) \right\rangle + \mathcal{O}(\epsilon) \\
&= \sqrt{\tfrac{K-1}{K}} \sum_{i \in \mathcal{I}_k : \alpha_i > 0} \sum_{i' : \alpha_{i'} \geq 0} \left( \langle \boldsymbol{x}_i, \xi_{i'}\boldsymbol{x}_{i'} \rangle - \alpha_i \xi_{i'}\alpha_{i'} \right) \beta_{i'} + \mathcal{O}(\epsilon) && (37) \\
&= \sqrt{\tfrac{K-1}{K}} \sum_{i \in \mathcal{I}_k : \alpha_i > 0} \sum_{1 \leq k' \leq K} \sum_{i' \in \mathcal{I}_{k'} : \alpha_{i'} \geq 0} \left( \langle \boldsymbol{x}_i, \xi_{i'}\boldsymbol{x}_{i'} \rangle - \alpha_i \xi_{i'}\alpha_{i'} \right) \beta_{i'} + \mathcal{O}(\epsilon) \\
&= \sqrt{\tfrac{K-1}{K}} \sum_{1 \leq k' \leq K} \sum_{i \in \mathcal{I}_k : \alpha_i > 0} \sum_{i' \in \mathcal{I}_{k'} : \alpha_{i'} \geq 0} \left( \langle \boldsymbol{x}_i, \xi_{i'}\boldsymbol{x}_{i'} \rangle - \alpha_i \xi_{i'}\alpha_{i'} \right) \mathcal{B}_{k'} + \mathcal{O}(\epsilon) && (38) \\
&= \sqrt{\tfrac{K-1}{K}} \sum_{1 \leq k' \leq K} \mathcal{B}_{k'} \left( \sum_{i \in \mathcal{I}_k : \alpha_i > 0} \sum_{i' \in \mathcal{I}_{k'} : \alpha_{i'} \geq 0} \left( \langle \boldsymbol{x}_i, \xi_{i'}\boldsymbol{x}_{i'} \rangle - \alpha_i \xi_{i'}\alpha_{i'} \right) \right) + \mathcal{O}(\epsilon) \\
&= \sqrt{\tfrac{K-1}{K}} \sum_{1 \leq k' \leq K} \mathcal{B}_{k'} \left( \left\langle \sum_{i \in \mathcal{I}_k : \alpha_i > 0} \boldsymbol{x}_i, \sum_{i' \in \mathcal{I}_{k'} : \alpha_{i'} \geq 0} \xi_{i'}\boldsymbol{x}_{i'} \right\rangle \right. \\
&\qquad\qquad \left. - \left( \sum_{i \in \mathcal{I}_k : \alpha_i > 0} \alpha_i \right) \left( \sum_{i' \in \mathcal{I}_{k'} : \alpha_{i'} \geq 0} \xi_{i'}\alpha_{i'} \right) \right) + \mathcal{O}(\epsilon) \\
&= \sqrt{\tfrac{K-1}{K}} \sum_{1 \leq k' \leq K} \mathcal{B}_{k'} \left( \left\langle \sum_{i \in \mathcal{I}_k : \alpha_i > 0} \boldsymbol{x}_i, \sum_{i' \in \mathcal{I}_{k'} : \alpha_{i'} \geq 0} \xi_{i'}\boldsymbol{x}_{i'} \right\rangle - \mathcal{A}_k \mathcal{A}_{k'} \right) + \mathcal{O}(\epsilon),
\end{aligned}
$$
$$(39)$$

where (39) uses the simple fact that $\sum_{i' \in \mathcal{I}_{k'} : \alpha_{i'} \geq 0} \xi_{i'}\alpha_{i'} = \sum_{i' \in \mathcal{I}_{k'} : \alpha_{i'} > 0} \alpha_{i'} = \mathcal{A}_{k'}$, (38) uses the fact that $\beta_{i'} = \left\langle \tilde{\boldsymbol{E}}\boldsymbol{y}_{i'}, \frac{\boldsymbol{v}}{\|\boldsymbol{v}\|} \right\rangle = \left\langle \tilde{\boldsymbol{e}}_{k'}, \frac{\boldsymbol{v}}{\|\boldsymbol{v}\|} \right\rangle$ if $i' \in \mathcal{I}_{k'}$, (37) uses the fact that $\xi_{i'} = 0$ if $\alpha_{i'} < 0$, and (36) uses Lemma 3, with the $\mathcal{O}(\epsilon)$ term being

$$
\sum_{i \in \mathcal{I}_k : \alpha_i > 0} \left\langle \boldsymbol{x}_i, \frac{\mathrm{d}}{\mathrm{d}t}\frac{\boldsymbol{w}}{\|\boldsymbol{w}\|} - \Pi_{\boldsymbol{w}}^{\perp} \left( \sqrt{\tfrac{K-1}{K}} \sum_{i'=1}^{n} \xi_{i'} \left\langle \tilde{\boldsymbol{E}}\boldsymbol{y}_{i'}, \frac{\boldsymbol{v}}{\|\boldsymbol{v}\|} \right\rangle \boldsymbol{x}_{i'} \right) \right\rangle,
$$

whose norm can be upper bounded as follows:

$$\left\| \sum_{i \in \mathcal{I}_k : \alpha_i > 0} \left\langle \boldsymbol{x}_i, \frac{\mathrm{d}}{\mathrm{d}t} \frac{\boldsymbol{w}}{\|\boldsymbol{w}\|} - \Pi_{\boldsymbol{w}}^{\perp} \left( \sqrt{\frac{K-1}{K}} \sum_{i'=1}^{n} \xi_{i'} \left\langle \tilde{\boldsymbol{E}} \boldsymbol{y}_{i'}, \frac{\boldsymbol{v}}{\|\boldsymbol{v}\|} \right\rangle \boldsymbol{x}_{i'} \right) \right\rangle \right\|$$

$$\leq \sum_{i=1}^{n} \|\boldsymbol{x}_i\| \left\| \frac{\mathrm{d}}{\mathrm{d}t} \frac{\boldsymbol{w}}{\|\boldsymbol{w}\|} - \Pi_{\boldsymbol{w}}^{\perp} \left( \sqrt{\frac{K-1}{K}} \sum_{i'=1}^{n} \xi_{i'} \left\langle \tilde{\boldsymbol{E}} \boldsymbol{y}_{i'}, \frac{\boldsymbol{v}}{\|\boldsymbol{v}\|} \right\rangle \boldsymbol{x}_{i'} \right) \right\| \leq \frac{16}{\sqrt{K}} \epsilon n^2 X_{\max}^3 \sqrt{h} \,.$$

and

$$\begin{aligned}
\frac{\mathrm{d}}{\mathrm{d}t} \mathcal{B}_k &= \left\langle \tilde{\boldsymbol{e}}_k, \frac{\mathrm{d}}{\mathrm{d}t} \frac{\boldsymbol{v}}{\|\boldsymbol{v}\|} \right\rangle \\
&= \left\langle \tilde{\boldsymbol{e}}_k, \Pi_{\boldsymbol{v}}^{\perp} \left( \sqrt{\frac{K-1}{K}} \sum_{i=1}^{n} \xi_i \left\langle \boldsymbol{x}_i, \frac{\boldsymbol{w}}{\|\boldsymbol{w}\|} \right\rangle \tilde{\boldsymbol{E}} \boldsymbol{y}_i \right) \right\rangle + \mathcal{O}(\epsilon) \\
&= \sqrt{\frac{K-1}{K}} \left\langle \tilde{\boldsymbol{e}}_k, \Pi_{\boldsymbol{v}}^{\perp} \left( \sum_{1 \leq k' \leq K} \sum_{i \in \mathcal{I}_{k'} : \alpha_i > 0} \alpha_i \tilde{\boldsymbol{E}} \boldsymbol{y}_i \right) \right\rangle + \mathcal{O}(\epsilon) \\
&= \sqrt{\frac{K-1}{K}} \left\langle \tilde{\boldsymbol{e}}_k, \Pi_{\boldsymbol{v}}^{\perp} \left( \sum_{1 \leq k' \leq K} \sum_{i \in \mathcal{I}_{k'} : \alpha_i > 0} \alpha_i \tilde{\boldsymbol{e}}_{k'} \right) \right\rangle + \mathcal{O}(\epsilon) \\
&= \sqrt{\frac{K-1}{K}} \sum_{1 \leq k' \leq K} \sum_{i \in \mathcal{I}_{k'} : \alpha_i > 0} \left\langle \tilde{\boldsymbol{e}}_k, \Pi_{\boldsymbol{v}}^{\perp} \tilde{\boldsymbol{e}}_{k'} \right\rangle \alpha_i + \mathcal{O}(\epsilon) \\
&= \sqrt{\frac{K-1}{K}} \sum_{1 \leq k' \leq K} \left( \langle \tilde{\boldsymbol{e}}_k, \tilde{\boldsymbol{e}}_{k'} \rangle - \mathcal{B}_k \mathcal{B}_{k'} \right) \sum_{i \in \mathcal{I}_{k'} : \alpha_i > 0} \alpha_i + \mathcal{O}(\epsilon) \\
&= \sqrt{\frac{K-1}{K}} \sum_{1 \leq k' \leq K} \mathcal{A}_{k'} \left( \langle \tilde{\boldsymbol{e}}_k, \tilde{\boldsymbol{e}}_{k'} \rangle - \mathcal{B}_k \mathcal{B}_{k'} \right) + \mathcal{O}(\epsilon) \\
&= \sqrt{\frac{K-1}{K}} \left( \mathcal{A}_k (1 - \mathcal{B}_k^2) + \sum_{k' \neq k} \mathcal{A}_{k'} \left( -\frac{1}{K-1} - \mathcal{B}_k \mathcal{B}_{k'} \right) \right) + \mathcal{O}(\epsilon) \,, \qquad (40)
\end{aligned}$$

where multiple facts used to derive (39) are also used here, and in (40), the $\mathcal{O}(\epsilon)$ has its norm upper bounded by $\frac{16}{\sqrt{K}} \epsilon n X_{\max}^2 \sqrt{h}$.

**Axuillary Lemmas.** The following lemmas will be needed.

**Lemma 6.** *Given $\mathcal{B}_k, k = 1, \cdots, K$ defined for a single neuron pair $(\boldsymbol{w}, \boldsymbol{v})$, we have*
$$-2(1 - \mathcal{B}_k) - \tfrac{1}{K-1} \leq \mathcal{B}_{k'} \leq 2(1 - \mathcal{B}_k) - \tfrac{1}{K-1}, \forall k, k' \text{with } k' \neq k \,.$$

*Proof.* With the following basic derivation
$$\mathcal{B}_{k'} = \left\langle \tilde{\boldsymbol{e}}_{k'}, \frac{\boldsymbol{v}}{\|\boldsymbol{v}\|} \right\rangle \left\langle \tilde{\boldsymbol{e}}_{k'}, \frac{\boldsymbol{v}}{\|\boldsymbol{v}\|} - \tilde{\boldsymbol{e}}_k + \tilde{\boldsymbol{e}}_k \right\rangle \left\langle \tilde{\boldsymbol{e}}_{k'}, \frac{\boldsymbol{v}}{\|\boldsymbol{v}\|} - \tilde{\boldsymbol{e}}_k + \tilde{\boldsymbol{e}}_k \right\rangle = \left\langle \tilde{\boldsymbol{e}}_{k'}, \frac{\boldsymbol{v}}{\|\boldsymbol{v}\|} - \tilde{\boldsymbol{e}}_k \right\rangle - \frac{1}{K-1} \,,$$

the desired result comes from the fact that $\left| \left\langle \tilde{\boldsymbol{e}}_{k'}, \frac{\boldsymbol{v}}{\|\boldsymbol{v}\|} - \tilde{\boldsymbol{e}}_k \right\rangle \right| \leq \left\| \frac{\boldsymbol{v}}{\|\boldsymbol{v}\|} - \tilde{\boldsymbol{e}}_k \right\| = 2(1 - \mathcal{B}_k)$. $\qquad \square$

**Lemma 7.** *Given a dataset that satisfies Assumption 1, then the following is true:*

- $\forall k$ and some $a_i, \forall i \in \mathcal{I}_k$, we have
$$\left\| \sum_{i \in \mathcal{I}_k} a_i \boldsymbol{x}_i \right\| \geq \sqrt{\mu_s} \sum_{i \in \mathcal{I}_k} a_i X_{\min} \,; \qquad (41)$$

- $\forall k \neq k'$ and some $a_i, b_{i'} \geq 0, \forall i \in \mathcal{I}_k, i' \in \mathcal{I}_{i'}$, we have
$$\left\langle \sum_{i \in \mathcal{I}_k} a_i \boldsymbol{x}_i, \sum_{i' \in \mathcal{I}_{k'}} b_{i'} \boldsymbol{x}_{i'} \right\rangle \leq -\mu_d \left\| \sum_{i \in \mathcal{I}_k} a_i \boldsymbol{x}_i \right\| \left\| \sum_{i' \in \mathcal{I}_{k'}} b_{i'} \boldsymbol{x}_{i'} \right\| \,. \qquad (42)$$

*Proof.* For the first inequality,
$$\begin{aligned}
\left\| \sum_{i \in \mathcal{I}_k} a_i \boldsymbol{x}_i \right\| &= \sqrt{\sum_{i \in \mathcal{I}_k} a_i^2 \|\boldsymbol{x}_i\|^2 + \sum_{i \neq j} a_i a_j \langle \boldsymbol{x}_i, \boldsymbol{x}_j \rangle} \\
&\geq \sqrt{\sum_{i \in \mathcal{I}_k} a_i^2 \mu_s \|\boldsymbol{x}_i\|^2 + \sum_{i \neq j} a_i a_j \mu_s \|\boldsymbol{x}_i\| \|\boldsymbol{x}_j\|} \\
&\geq \sqrt{\mu_s} \sqrt{\sum_{i \in \mathcal{I}_k} a_i^2 + \sum_{i \neq j} a_i a_j} X_{\min} = \sqrt{\mu_s} \sum_{i \in \mathcal{I}_k} a_i X_{\min} \,.
\end{aligned}$$

For the second inequality,
$$\begin{aligned}
\left\langle \sum_{i \in \mathcal{I}_k} a_i \boldsymbol{x}_i, \sum_{i' \in \mathcal{I}_{k'}} b_{i'} \boldsymbol{x}_{i'} \right\rangle &= \sum_{i \in \mathcal{I}_k, i' \in \mathcal{I}_{k'}} a_i b_{i'} \langle \boldsymbol{x}_i, \boldsymbol{x}_{i'} \rangle \leq -\mu_d \sum_{i \in \mathcal{I}_k, i' \in \mathcal{I}_{k'}} a_i b_{i'} \|\boldsymbol{x}_i\| \|\boldsymbol{x}_{i'}\| \\
&\leq -\mu_d \left\| \sum_{i \in \mathcal{I}_k} a_i \boldsymbol{x}_i \right\| \left\| \sum_{i' \in \mathcal{I}_{k'}} b_{i'} \boldsymbol{x}_{i'} \right\| \,.
\end{aligned}$$
$$\square$$

**Lemma 8.** *Given $\{z_i, i \in \mathcal{I}\}$ with $\langle z_i, z_j \rangle \leq 0, \forall i, j \in \mathcal{I}, i \neq j$, then $\|\sum_{i \in \mathcal{I}} z_i\| \leq \sqrt{\sum_{i \in \mathcal{I}} \|z_i\|^2}$.*

*Proof.* $\|\sum_{i \in \mathcal{I}} z_i\| = \sqrt{\sum_{i \in \mathcal{I}} \|z_i\|^2 + \sum_{i \neq j} \langle z_i, z_j \rangle} \leq \sqrt{\sum_{i \in \mathcal{I}} \|z_i\|^2}$. $\qquad\square$

**Lemma 9.** *Given a dataset that satisfies Assumption 1, $\exists 0 < \zeta < 1$ such that $\forall k \in [K]$ and $\forall w \in \{z : 0 < |\mathcal{I}_k^z| < |\mathcal{I}_k|\}$, we have $\left\|\sum_{i \in \mathcal{I}_k : \alpha_i > 0} x_i\right\|^2 - \mathcal{A}_k^2 \geq \mu_s X_{\min}^2 \zeta$.*

*Proof.* Notice that

$$\left\|\sum_{i \in \mathcal{I}_k : \alpha_i > 0} x_i\right\|^2 - \mathcal{A}_k^2 = \left\|\sum_{i \in \mathcal{I}_k : \alpha_i > 0} x_i\right\|^2 \left(1 - \left\langle \frac{w}{\|w\|}, \frac{\sum_{i \in \mathcal{I}_k : \alpha_i > 0} x_i}{\|\sum_{i \in \mathcal{I}_k : \alpha_i > 0} x_i\|} \right\rangle^2\right)$$

$$\overset{\text{(Lemma 7)}}{\geq} \mu_s X_{\min}^2 \left(1 - \left\langle \frac{w}{\|w\|}, \frac{\sum_{i \in \mathcal{I}_k : \alpha_i > 0} x_i}{\|\sum_{i \in \mathcal{I}_k : \alpha_i > 0} x_i\|} \right\rangle^2\right). \tag{43}$$

However, the nonnegative quantity $\left(1 - \left\langle \frac{w}{\|w\|}, \frac{\sum_{i \in \mathcal{I}_k : \alpha_i > 0} x_i}{\|\sum_{i \in \mathcal{I}_k : \alpha_i > 0} x_i\|} \right\rangle^2\right)$ can not be zero: Suppose it is zero, then $w \propto \pm \sum_{i \in \mathcal{I}_k : \alpha_i > 0} x_i$, which corresponds to either $|\mathcal{I}_k^z| = 0$ or $|\mathcal{I}_k^z| = |\mathcal{I}_k|$, a contradiction. We let its lowest value be $\zeta > 0$. This finishes the proof. $\qquad\square$

**The proof**. Now we are ready to prove Lemma 4.

*Proof of Lemma 4.* We define the following:

$$\tau_1 = \inf\left\{t \geq 0 : \mathcal{B}_k < 1 - \frac{1}{2(K-1)}\right\},$$

$$\tau_2 = \inf\left\{t \geq 0 : \mathcal{A}_k - 2\sum_{k' \neq k} \mathcal{A}_{k'} < \mathcal{A}_k(0) - 2\sum_{k' \neq k} \mathcal{A}_{k'}(0)\right\},$$

$$\tau_3 = \inf\left\{t \geq 0 : |\mathcal{I}_k^w| < |\mathcal{I}_k^{w(0)}| \text{ or } |\mathcal{I}_{k'}^w| > |\mathcal{I}_{k'}^{w(0)}| \text{ for some } k'\right\}.$$

Then it suffices to show that $\min\{\tau_1, \tau_2, \tau_3\} \geq \min\{T_{j,k}^*, T\}$, for which we prove them by contradiction. **Note: In the proof we will use "$\overset{(*)}{\geq}$" to represent an inequality that holds when $\epsilon$ is sufficiently small.**

**Case 1:** $\min\{\tau_1, \tau_2, \tau_3\} = \tau_1$.

At $\tau_1$, by the continuity of $\mathcal{B}_k$, we must have $\mathcal{B}_k(\tau_1) = 1 - \frac{1}{2(K-1)}$. Suppose $\tau_1 \leq \min\{T_{j,k}^*, T\}$, then we have the following derivation

$$\frac{\mathrm{d}}{\mathrm{d}t} \mathcal{B}_k \bigg|_{t=\tau_1}$$

$$= \sqrt{\frac{K-1}{K}} \left(\mathcal{A}_k(1 - \mathcal{B}_k^2) + \sum_{k' \neq k} \mathcal{A}_{k'} \left(-\frac{1}{K-1} - \mathcal{B}_k \mathcal{B}_{k'}\right)\right) + \mathcal{O}(\epsilon)$$

$$\overset{\text{(Lemma 6)}}{\geq} \sqrt{\frac{K-1}{K}} \left(\mathcal{A}_k(1 - \mathcal{B}_k^2) + \sum_{k' \neq k} \mathcal{A}_{k'} \left(-\frac{1}{K-1} - \mathcal{B}_k \left(2(1 - \mathcal{B}_k) - \frac{1}{K-1}\right)\right)\right) + \mathcal{O}(\epsilon)$$

$$= \sqrt{\frac{K-1}{K}} \left(\mathcal{A}_k(1 - \mathcal{B}_k^2) - 2\sum_{k' \neq k} \mathcal{A}_{k'} \left(\frac{1 - \mathcal{B}_k}{2(K-1)} + \mathcal{B}_k(1 - \mathcal{B}_k)\right)\right) + \mathcal{O}(\epsilon)$$

$$= \sqrt{\frac{K-1}{K}} \left(\left(\mathcal{A}_k - 2\sum_{k' \neq k} \mathcal{A}_{k'}\right)(1 - \mathcal{B}_k^2) + 2\sum_{k' \neq k} \mathcal{A}_{k'} \left(1 - \mathcal{B}_k^2 - \frac{1 - \mathcal{B}_k}{2(K-1)} - \mathcal{B}_k(1 - \mathcal{B}_k)\right)\right) + \mathcal{O}(\epsilon)$$

$$\overset{(t=\tau_1)}{\geq} \sqrt{\frac{K-1}{K}} \left(\mathcal{A}_k - 2\sum_{k' \neq k} \mathcal{A}_{k'}\right) \frac{1}{K-1} + 2\sum_{k' \neq k} \mathcal{A}_{k'} \left(\frac{1}{2(K-1)} \left(1 - \frac{1}{2(K-1)}\right)\right) + \mathcal{O}(\epsilon)$$

$$\overset{(\tau_2 \geq \tau_1)}{\geq} \sqrt{\frac{K-1}{K}} \left(\mathcal{A}_k(0) - 2\sum_{k' \neq k} \mathcal{A}_{k'}(0)\right) \frac{1}{K-1} + \mathcal{O}(\epsilon)$$

$$\overset{(40), \tau_1 \leq T}{\geq} \sqrt{\frac{K-1}{K}} \left(\mathcal{A}_k(0) - 2\sum_{k' \neq k} \mathcal{A}_{k'}(0)\right) \frac{1}{K-1} - \frac{16}{\sqrt{K}} \epsilon n X_{\max}^2 \sqrt{h} \overset{(*)}{\geq} 0. \tag{44}$$

The definition of $\tau_1$ suggests that $\mathcal{B}_k$ must drop below $1 - \frac{1}{2(K-1)}$ right after $t = \tau_1$, which contradicts that $\frac{\mathrm{d}}{\mathrm{d}t}\mathcal{B}_k\big|_{t=\tau_1} \geq 0$. Therefore $\min\{\tau_1, \tau_2, \tau_3\} > \min\{T^*_{j,k}, T\}$ can not be true under the case when $\min\{\tau_1, \tau_2, \tau_3\} = \tau_1$.

**Case 2:** $\min\{\tau_1, \tau_2, \tau_3\} = \tau_2$.

Again, we derive a contradiction by supposing $\tau_2 \leq \min\{T^*, T\}$. Since $\min\{\tau_1, \tau_2, \tau_3\} = \tau_2$, at $\tau_2$ we still have $\mathcal{B}_k \geq 1 - \frac{1}{2(K-1)} > 0$, and by Lemma 6, we also have $\mathcal{B}_{k'} \leq 2(1 - \mathcal{B}_k) - \frac{1}{K-1} \leq 0$.

Starting from (39) restricted to $t = \tau_2$, we have for the target class,

$$
\frac{\mathrm{d}}{\mathrm{d}t}\mathcal{A}_k\bigg|_{t=\tau_2}
$$

$$
= \sqrt{\tfrac{K-1}{K}} \sum_{1 \leq k' \leq K} \mathcal{B}_{k'} \left( \left\langle \sum_{i \in \mathcal{I}_k : \alpha_i > 0} \boldsymbol{x}_i, \sum_{i' \in \mathcal{I}_{k'} : \alpha_{i'} \geq 0} \xi_{i'} \boldsymbol{x}_{i'} \right\rangle - \mathcal{A}_k \mathcal{A}_{k'} \right) + \mathcal{O}(\epsilon)
$$

$$
= \sqrt{\tfrac{K-1}{K}} \underbrace{\mathcal{B}_k}_{\geq 0} \left( \left\| \sum_{i \in \mathcal{I}_k : \alpha_i > 0} \boldsymbol{x}_i \right\|^2 - \mathcal{A}_k^2 + \underbrace{\left\langle \sum_{i \in \mathcal{I}_k : \alpha_i > 0} \boldsymbol{x}_i, \sum_{i' \in \mathcal{I}_k : \alpha_{i'} = 0} \xi_{i'} \boldsymbol{x}_{i'} \right\rangle}_{\geq 0} \right)
$$

$$
+ \sqrt{\tfrac{K-1}{K}} \sum_{k' \neq k} \underbrace{\mathcal{B}_{k'}}_{\leq 0} \left( \underbrace{\left\langle \sum_{i \in \mathcal{I}_k : \alpha_i > 0} \boldsymbol{x}_i, \sum_{i' \in \mathcal{I}_{k'} : \alpha_{i'} \geq 0} \xi_{i'} \boldsymbol{x}_{i'} \right\rangle}_{\leq 0} - \underbrace{\mathcal{A}_k \mathcal{A}_{k'}}_{\geq 0} \right) + \mathcal{O}(\epsilon),
$$

$$
\geq \sqrt{\tfrac{K-1}{K}}\mathcal{B}_k \left( \left\| \sum_{i \in \mathcal{I}_k : \alpha_i > 0} \boldsymbol{x}_i \right\|^2 - \mathcal{A}_k^2 \right) + \mathcal{O}(\epsilon), \tag{45}
$$

and for non-target classes, we have

$$
\frac{\mathrm{d}}{\mathrm{d}t} \sum_{k' \neq k} \mathcal{A}_{k'} \bigg|_{t=\tau_2}
$$

$$
= \sqrt{\tfrac{K-1}{K}} \sum_{k' \neq k} \sum_{1 \leq k'' \leq K} \mathcal{B}_{k''} \left( \left\langle \sum_{i \in \mathcal{I}_{k'} : \alpha_i > 0} \boldsymbol{x}_i, \sum_{i' \in \mathcal{I}_{k''} : \alpha_{i'} \geq 0} \xi_{i'} \boldsymbol{x}_{i'} \right\rangle - \mathcal{A}_{k'} \mathcal{A}_{k''} \right) + \mathcal{O}(\epsilon)
$$

$$
= \sqrt{\tfrac{K-1}{K}} \sum_{k' \neq k} \mathcal{B}_k \left( \left\langle \sum_{i \in \mathcal{I}_k : \alpha_i > 0} \boldsymbol{x}_i, \sum_{i' \in \mathcal{I}_{k'} : \alpha_{i'} \geq 0} \xi_{i'} \boldsymbol{x}_{i'} \right\rangle - \mathcal{A}_{k'} \mathcal{A}_k \right)
$$

$$
+ \sqrt{\tfrac{K-1}{K}} \sum_{k' \neq k} \sum_{k'' \neq k} \mathcal{B}_{k''} \left( \left\langle \sum_{i \in \mathcal{I}_{k'} : \alpha_i > 0} \boldsymbol{x}_i, \sum_{i' \in \mathcal{I}_{k''} : \alpha_{i'} \geq 0} \xi_{i'} \boldsymbol{x}_{i'} \right\rangle - \mathcal{A}_{k'} \mathcal{A}_{k''} \right) + \mathcal{O}(\epsilon)
$$

$$
= \sqrt{\tfrac{K-1}{K}} \sum_{k' \neq k} \mathcal{B}_k \left\langle \sum_{i \in \mathcal{I}_k : \alpha_i > 0} \boldsymbol{x}_i, \sum_{i' \in \mathcal{I}_{k'} : \alpha_{i'} \geq 0} \xi_{i'} \boldsymbol{x}_{i'} \right\rangle
$$

$$
+ \sqrt{\tfrac{K-1}{K}} \left\langle \sum_{k' \neq k} \sum_{i \in \mathcal{I}_{k'} : \alpha_i > 0} \boldsymbol{x}_i, \sum_{k'' \neq k} \mathcal{B}_{k''} \sum_{i' \in \mathcal{I}_{k''} : \alpha_{i'} \geq 0} \xi_{i'} \boldsymbol{x}_{i'} \right\rangle
$$

$$
+ \sqrt{\tfrac{K-1}{K}} \sum_{k' \neq k} \left( -\mathcal{B}_k \mathcal{A}_{k'} \mathcal{A}_k - \sum_{k'' \neq k} \mathcal{B}_{k''} \mathcal{A}_{k'} \mathcal{A}_{k''} \right) + \mathcal{O}(\epsilon) \tag{46}
$$

$$
\leq -\sqrt{\tfrac{K-1}{K}} \left( \mu_d \mu_s X_{\min}^2 \mathcal{B}_k \sum_{k'=k} |\mathcal{I}_{k'}^{\boldsymbol{w}}|^2 + \tfrac{2}{K-1} X_{\max}^2 \sum_{k'=k} |\mathcal{I}_{k'}^{\boldsymbol{w}}|^2 \right) + \mathcal{O}(\epsilon), \tag{47}
$$

The last step to get (47) is to upper bound the three terms in (46) separately, which we defer to the end of this proof. Combining (45)(47), and recalling the upper bound on the norm of the $\mathcal{O}(\epsilon)$ terms, we have

$$
\frac{\mathrm{d}}{\mathrm{d}t} \left( \mathcal{A}_k - 2 \sum_{k' \neq k} \mathcal{A}_{k'} \right) \bigg|_{t=\tau_2}
$$

$$
\geq \sqrt{\tfrac{K-1}{K}}\mathcal{B}_k \left( \left\| \sum_{i \in \mathcal{I}_k : \alpha_i > 0} \boldsymbol{x}_i \right\|^2 - \mathcal{A}_k^2 \right)
$$

$$
2 \left( \mu_d \mu_s X_{\min}^2 \mathcal{B}_k - \tfrac{2}{K-1} X_{\max}^2 \right) \sum_{k'=k} |\mathcal{I}_{k'}^{\boldsymbol{w}}|^2 - 32\sqrt{K}\epsilon n^2 X_{\max}^3 \sqrt{h}
$$

$$
\geq \sqrt{\tfrac{K-1}{K}}\mathcal{B}_k \left( \left\| \sum_{i \in \mathcal{I}_k : \alpha_i > 0} \boldsymbol{x}_i \right\|^2 - \mathcal{A}_k^2 \right)
$$

$$
2 \underbrace{\left( \mu_d \mu_s X_{\min}^2 \left( 1 - \tfrac{1}{2(K-1)} \right) - \tfrac{2}{K-1} X_{\max}^2 \right)}_{\geq 0} \sum_{k'=k} |\mathcal{I}_{k'}^{\boldsymbol{w}}|^2 - 32\sqrt{K}\epsilon n^2 X_{\max}^3 \sqrt{h}
$$

$$\geq \sqrt{\tfrac{K-1}{K}}\left(1 - \tfrac{1}{2(K-1)}\right)\mu_s X_{\min}^2 \zeta - \tfrac{32}{\sqrt{K}}\epsilon n^2 X_{\max}^3 \sqrt{h} \overset{(*)}{\geq} 0. \tag{48}$$

The definition of $\tau_2$ suggests that $\mathcal{A}_k - 2\sum_{k'\neq k}\mathcal{A}_{k'}$ must drop below $\mathcal{A}_k(0) - 2\sum_{k'\neq k}\mathcal{A}_{k'}(0)$ right after $t = \tau_2$, which contradicts that $\frac{\mathrm{d}}{\mathrm{d}t}\left(\mathcal{A}_k - 2\sum_{k'\neq k}\mathcal{A}_{k'}\right)\Big|_{t=\tau_2} \geq 0$. Therefore $\min\{\tau_1, \tau_2, \tau_3\} > \min\{T_{j,k}^*, T\}$ can not be true under the case when $\min\{\tau_1, \tau_2, \tau_3\} = \tau_2$.

**Case 3:** $\min\{\tau_1, \tau_2, \tau_3\} = \tau_3$. Finally, it remains to exclude the case when $\min\{\tau_1, \tau_2, \tau_3\} = \tau_3$ and $\tau_3 \leq \min\{T_{j,k}^*, T\}$. At $t = \tau_3$, either of the following must happen:

1. $\exists i \in \mathcal{I}_k$ such that $\alpha_i = 0$ and $\frac{\mathrm{d}}{\mathrm{d}t}\alpha_i < 0$;

2. $\exists i \in \mathcal{I}_{k'}$ for some $k' \neq k$ such that $\alpha_i = 0$ and $\frac{\mathrm{d}}{\mathrm{d}t}\alpha_i > 0$;

However, at $t = \tau_3$, $\forall i \in \mathcal{I}_k$, we have

$$
\begin{aligned}
&\frac{\mathrm{d}}{\mathrm{d}t}\alpha_i\Big|_{\alpha_i=0} \\
&= \left\langle \boldsymbol{x}_i, \frac{\mathrm{d}}{\mathrm{d}t}\frac{\boldsymbol{w}}{\|\boldsymbol{w}\|}\right\rangle \\
&\overset{\text{(Lemma 3)}}{=} \left\langle \boldsymbol{x}_i, \Pi_{\boldsymbol{w}}^\perp\left(\sqrt{\tfrac{K-1}{K}}\sum_{i'=1}^n \xi_{i'}\left\langle \tilde{\boldsymbol{E}}\boldsymbol{y}_{i'}, \frac{\boldsymbol{v}}{\|\boldsymbol{v}\|}\right\rangle\boldsymbol{x}_{i'}\right)\right\rangle + \mathcal{O}(\epsilon) \\
&\overset{(\alpha_i=0)}{=} \sqrt{\tfrac{K-1}{K}}\left\langle \boldsymbol{x}_i, \left(\sum_{i'=1}^n \xi_{i'}\beta_{i'}\boldsymbol{x}_{i'}\right)\right\rangle + \mathcal{O}(\epsilon) \\
&= \sqrt{\tfrac{K-1}{K}}\left(\mathcal{B}_k\left\langle \boldsymbol{x}_i, \sum_{i\in\mathcal{I}_k,\alpha_i\geq 0}\xi_i\boldsymbol{x}_i\right\rangle + \sum_{k'\neq k}\underbrace{\mathcal{B}_{k'}}_{\leq 0}\underbrace{\left\langle \boldsymbol{x}_i, \sum_{i'\in\mathcal{I}_{k'},\alpha_{i'}\geq 0}\xi_{i'}\boldsymbol{x}_{i'}\right\rangle}_{\leq 0}\right) + \mathcal{O}(\epsilon) \\
&\geq \sqrt{\tfrac{K-1}{K}}\mathcal{B}_k\left\langle \boldsymbol{x}_i, \sum_{i\in\mathcal{I}_k,\alpha_i\geq 0}\xi_i\boldsymbol{x}_i\right\rangle + \mathcal{O}(\epsilon) \\
&\overset{\text{(Lemma 7)}}{\geq} \sqrt{\tfrac{K-1}{K}}\left(1 - \tfrac{1}{2(K-1)}\right)|\mathcal{I}_k(0)|\mu_s X_{\min}^2 - \tfrac{16}{\sqrt{K}}\epsilon n X_{\max}^3\sqrt{h} \overset{(*)}{\geq} 0, \tag{49}
\end{aligned}
$$

therefore it can not be that $\exists i \in \mathcal{I}_k$ such that $\alpha_i = 0$ and $\frac{\mathrm{d}}{\mathrm{d}t}\alpha_i < 0$. Next, $\forall i \in \mathcal{I}_{k'}, k' \neq k$, we have

$$
\begin{aligned}
&\frac{\mathrm{d}}{\mathrm{d}t}\alpha_i\Big|_{\alpha_i=0} \\
&= \sqrt{\tfrac{K-1}{K}}\left\langle \boldsymbol{x}_i, \left(\sum_{i'=1}^n \xi_{i'}\beta_{i'}\boldsymbol{x}_{i'}\right)\right\rangle + \mathcal{O}(\epsilon) \\
&= \sqrt{\tfrac{K-1}{K}}\left(\mathcal{B}_k\left\langle \boldsymbol{x}_i, \sum_{i\in\mathcal{I}_k,\alpha_i\geq 0}\xi_i\boldsymbol{x}_i\right\rangle + \sum_{k'\neq k}\mathcal{B}_{k'}\left\langle \boldsymbol{x}_i, \sum_{i'\in\mathcal{I}_{k'},\alpha_{i'}\geq 0}\xi_{i'}\boldsymbol{x}_{i'}\right\rangle\right) + \mathcal{O}(\epsilon) \\
&= \sqrt{\tfrac{K-1}{K}}\left(\mathcal{B}_k\left\langle \boldsymbol{x}_i, \sum_{i\in\mathcal{I}_k,\alpha_i\geq 0}\xi_i\boldsymbol{x}_i\right\rangle + \left\langle \boldsymbol{x}_i, \sum_{k'\neq k}\sum_{i'\in\mathcal{I}_{k'},\alpha_{i'}\geq 0}\mathcal{B}_{k'}\xi_{i'}\boldsymbol{x}_{i'}\right\rangle\right) + \mathcal{O}(\epsilon) \\
&\overset{\text{(Lemma 7)}}{\leq} \sqrt{\tfrac{K-1}{K}}\left(-\mu_d\mathcal{B}_k\|\boldsymbol{x}_i\|\|\sum_{i\in\mathcal{I}_k,\alpha_i\geq 0}\xi_i\boldsymbol{x}_i\| + \|\boldsymbol{x}_i\|\|\sum_{k'\neq k}\sum_{i'\in\mathcal{I}_{k'},\alpha_{i'}\geq 0}\mathcal{B}_{k'}\xi_{i'}\boldsymbol{x}_{i'}\|\right) + \mathcal{O}(\epsilon) \\
&\overset{\text{(Lemma 7)}}{\leq} \sqrt{\tfrac{K-1}{K}}\left(-\mu_d\mu_s\mathcal{B}_k X_{\min}^2|\mathcal{I}_k| + \|\boldsymbol{x}_i\|\|\sum_{k'\neq k}\sum_{i'\in\mathcal{I}_{k'},\alpha_{i'}\geq 0}\mathcal{B}_{k'}\xi_{i'}\boldsymbol{x}_{i'}\|\right) + \mathcal{O}(\epsilon) \\
&\overset{\text{(Lemma 8)}}{\leq} \sqrt{\tfrac{K-1}{K}}\left(-\mu_d\mu_s\mathcal{B}_k X_{\min}^2|\mathcal{I}_k| + \|\boldsymbol{x}_i\|\sqrt{\sum_{k'\neq k}\|\sum_{i'\in\mathcal{I}_{k'},\alpha_{i'}\geq 0}\mathcal{B}_{k'}\xi_{i'}\boldsymbol{x}_{i'}\|^2}\right) + \mathcal{O}(\epsilon) \\
&\leq \sqrt{\tfrac{K-1}{K}}\left(-\mu_d\mu_s\mathcal{B}_k X_{\min}^2|\mathcal{I}_k| + X_{\max}^2\sqrt{\sum_{k'\neq k}|\mathcal{B}_{k'}|^2|\mathcal{I}_{k'}|^2}\right) + \mathcal{O}(\epsilon) \\
&\overset{(\tau_3\leq\tau_1)}{\leq} \sqrt{\tfrac{K-1}{K}}\left(-\mu_d\mu_s\left(1 - \tfrac{1}{2(K-1)}\right)X_{\min}^2|\mathcal{I}_k| + \tfrac{2}{K-1}X_{\max}^2\sqrt{\sum_{k'\neq k}|\mathcal{I}_{k'}|^2}\right) + \mathcal{O}(\epsilon) \\
&\overset{(t=\tau_3)}{\leq} \sqrt{\tfrac{K-1}{K}}|\mathcal{I}_k(0)|\left(-\mu_d\mu_s\left(1 - \tfrac{1}{2(K-1)}\right)X_{\min}^2 + \tfrac{2}{K-1}X_{\max}^2\right) - \tfrac{16}{\sqrt{K}}\epsilon n X_{\max}^3\sqrt{h} \overset{(*)}{\leq} 0
\end{aligned}
$$
$$\tag{50}$$

therefore it can not be that $\exists i \in \mathcal{I}_{k'}, k' \neq k$ such that $\alpha_i = 0$ and $\frac{\mathrm{d}}{\mathrm{d}t}\alpha_i > 0$. By excluding both scenarios, $\min\{\tau_1, \tau_2, \tau_3\} > \min\{T_{j,k}^*, T\}$ can not be true under the case when $\min\{\tau_1, \tau_2, \tau_3\} = \tau_3$. The proof is complete once we add the derivations for 47.

**Complete the proof**. Lastly, it remains to prove (47), which comes from the following derivations: For the first term,

$$\sum_{k'=k} \mathcal{B}_k \left\langle \sum_{i\in\mathcal{I}_k:\alpha_i>0} \boldsymbol{x}_i, \sum_{i'\in\mathcal{I}_{k'}:\alpha_{i'}\geq 0} \xi_{i'}\boldsymbol{x}_{i'} \right\rangle$$

$$\leq \sum_{k'=k} \underbrace{\mathcal{B}_k}_{\geq 0} \left( \left\langle \sum_{i\in\mathcal{I}_k:\alpha_i>0} \boldsymbol{x}_i, \sum_{i'\in\mathcal{I}_{k'}:\alpha_{i'}>0} \boldsymbol{x}_{i'} \right\rangle + \underbrace{\left\langle \sum_{i\in\mathcal{I}_k:\alpha_i>0} \boldsymbol{x}_i, \sum_{i'\in\mathcal{I}_{k'}:\alpha_{i'}=0} \xi_{i'}\boldsymbol{x}_{i'} \right\rangle}_{\leq 0} \right)$$

$$\leq \sum_{k'=k} \mathcal{B}_k \left\langle \sum_{i\in\mathcal{I}_k:\alpha_i>0} \boldsymbol{x}_i, \sum_{i'\in\mathcal{I}_{k'}:\alpha_{i'}>0} \boldsymbol{x}_{i'} \right\rangle$$

$$\overset{\text{(Lemma 7)}}{\leq} -\mu_d \sum_{k'=k} \mathcal{B}_k \left\| \sum_{i\in\mathcal{I}_k:\alpha_i>0} \boldsymbol{x}_i \right\| \left\| \sum_{i'\in\mathcal{I}_{k'}:\alpha_{i'}>0} \boldsymbol{x}_{i'} \right\|$$

$$\overset{\text{(Lemma 7)}}{\leq} -\mu_d \mu_s X_{\min}^2 \sum_{k'=k} \mathcal{B}_k |\mathcal{I}_k^{\boldsymbol{w}}||\mathcal{I}_{k'}^{\boldsymbol{w}}|$$

$$\overset{(\tau_2 \leq \tau_3)}{\leq} -\mu_d \mu_s X_{\min}^2 \mathcal{B}_k \sum_{k'=k} |\mathcal{I}_{k'}^{\boldsymbol{w}}|^2 \,.$$

For the second term,

$$\left\langle \sum_{k'\neq k} \sum_{i\in\mathcal{I}_{k'}:\alpha_i>0} \boldsymbol{x}_i, \sum_{k''\neq k} \mathcal{B}_{k''} \sum_{i'\in\mathcal{I}_{k''}:\alpha_{i'}\geq 0} \xi_{i'}\boldsymbol{x}_{i'} \right\rangle$$

$$= -\left\langle \sum_{k'\neq k} \sum_{i\in\mathcal{I}_{k'}:\alpha_i>0} \boldsymbol{x}_i, \sum_{k''\neq k} \sum_{i'\in\mathcal{I}_{k''}:\alpha_{i'}\geq 0}(-\mathcal{B}_{k''})\xi_{i'}\boldsymbol{x}_{i'} \right\rangle$$

$$\leq \left\| \sum_{k'\neq k} \sum_{i\in\mathcal{I}_{k'}:\alpha_i>0} \boldsymbol{x}_i \right\| \left\| \sum_{k''\neq k} \sum_{i'\in\mathcal{I}_{k''}:\alpha_{i'}\geq 0}(-\mathcal{B}_{k''})\xi_{i'}\boldsymbol{x}_{i'} \right\|$$

$$\overset{\text{(Lemma 8)}}{\leq} \sqrt{\sum_{k'\neq k} \left\| \sum_{i\in\mathcal{I}_{k'}:\alpha_i>0} \boldsymbol{x}_i \right\|^2} \sqrt{\sum_{k''\neq k}(-\mathcal{B}_{k''})^2 \left\| \sum_{i'\in\mathcal{I}_{k''}:\alpha_{i'}\geq 0} \xi_{i'}\boldsymbol{x}_{i'} \right\|^2}$$

$$\overset{\text{(Lemma 6)}}{\leq} \sqrt{\sum_{k'\neq k} |\mathcal{I}_{k'}^{\boldsymbol{w}}|^2 X_{\max}^2} \sqrt{\left(\tfrac{2}{K-1}\right)^2 \sum_{k''\neq k} |\mathcal{I}_{k''}^{\boldsymbol{w}}|^2 X_{\max}^2} \tag{51}$$

$$\leq \tfrac{2}{K-1} X_{\max}^2 \sum_{k'=k} |\mathcal{I}_{k'}^{\boldsymbol{w}}|^2 \,,$$

where (51) uses that $-\frac{2}{K-1} \leq \mathcal{B}_{k'} \leq 0, \forall k' \neq k$ by Lemma 6. And for the last term,

$$\sum_{k'\neq k} \left( -\mathcal{B}_k \mathcal{A}_{k'} \mathcal{A}_k - \sum_{k''\neq k} \mathcal{B}_{k''} \mathcal{A}_{k'} \mathcal{A}_{k''} \right)$$

$$\overset{(t=\tau_2)}{\leq} \sum_{k'\neq k} \mathcal{A}_{k'} \left( -2\mathcal{B}_k \sum_{k''\neq k} \mathcal{A}_{k''} - \sum_{k''\neq k} \mathcal{B}_{k''} \mathcal{A}_{k''} \right)$$

$$= \sum_{k'\neq k} \mathcal{A}_{k'} \sum_{k''\neq k} \mathcal{A}_{k''} (-2\mathcal{B}_k - \mathcal{B}_{k''})$$

$$\overset{(\tau_2\leq\tau_1)}{\leq} \sum_{k'\neq k} \mathcal{A}_{k'} \sum_{k''\neq k} \mathcal{A}_{k''} \left( -2 + \tfrac{3}{K-1} \right) \leq 0 \,.$$

$$\square$$

### C.3.2 Proof of Lemma 5

We will use the following lemma:

**Lemma 10.** *For any $\bar{\boldsymbol{v}} \in \mathbb{S}^{K-1}$ such that $\langle \bar{\boldsymbol{v}}, \tilde{\boldsymbol{e}}_1 \rangle = \beta \in [0, 1]$, then $\forall \boldsymbol{p}$ such that $\boldsymbol{p} \geq \boldsymbol{0}$, $[\boldsymbol{p}]_1 = 0$, and $\langle \boldsymbol{p}, \mathbb{1} \rangle = 1$, we have*

$$\left| \tfrac{1}{K-1} \sum_{k>1} [\bar{\boldsymbol{v}}]_k - \langle \boldsymbol{p}, \bar{\boldsymbol{v}} \rangle \right| \leq \sqrt{1-\beta^2} \,, \tag{52}$$

*Proof.* First of all, since $\min_{k>1}[\bar{\boldsymbol{v}}]_k \leq \langle \boldsymbol{p}, \bar{\boldsymbol{v}} \rangle \leq \max_{k>1}[\bar{\boldsymbol{v}}]_k$, we know that

$$\left| \tfrac{1}{K-1} \sum_{k>1} [\bar{\boldsymbol{v}}]_k - \langle \boldsymbol{p}, \bar{\boldsymbol{v}} \rangle \right| \leq \max \left\{ \left| \tfrac{\sum_{k>1}[\bar{\boldsymbol{v}}]_k}{K-1} - \min_{k>1}[\bar{\boldsymbol{v}}]_k \right|, \left| \tfrac{\sum_{k>1}[\bar{\boldsymbol{v}}]_k}{K-1} - \max_{k>1}[\bar{\boldsymbol{v}}]_k \right| \right\}$$

$$\leq \left\| \tfrac{\sum_{k>1}[\bar{\boldsymbol{v}}]_k}{K-1} \mathbb{1} - [\bar{\boldsymbol{v}}]_{2:K} \right\|_\infty \leq \left\| \tfrac{\sum_{k>1}[\bar{\boldsymbol{v}}]_k}{K-1} \mathbb{1} - [\bar{\boldsymbol{v}}]_{2:K} \right\| \,.$$

Now given that $\langle \bar{\boldsymbol{v}}, \tilde{\boldsymbol{e}}_1 \rangle = \beta$, we can write $\bar{\boldsymbol{v}} = \beta \tilde{\boldsymbol{e}}_1 + \sqrt{1 - \beta^2} \boldsymbol{y}^\perp$, where $\boldsymbol{y}^\perp \in \mathbb{S}^{K-1}$ and $\boldsymbol{y} \perp \tilde{\boldsymbol{e}}_1$. Therefore,

$$
\begin{aligned}
\left\| \tfrac{\sum_{k>1}[\bar{\boldsymbol{v}}]_k}{K-1} \mathbb{1} - [\bar{\boldsymbol{v}}]_{2:K} \right\| &= \left\| \beta \left( \tfrac{\sum_{k>1}[\tilde{\boldsymbol{e}}_1]_k}{K-1} \mathbb{1} - [\tilde{\boldsymbol{e}}_1]_{2:K} \right) + \sqrt{1 - \beta^2} \left( \tfrac{\sum_{k>1}[\boldsymbol{y}^\perp]_k}{K-1} \mathbb{1} - [\boldsymbol{y}^\perp]_{2:K} \right) \right\| \\
&= \sqrt{1 - \beta^2} \left\| \tfrac{\sum_{k>1}[\boldsymbol{y}^\perp]_k}{K-1} \mathbb{1} - [\boldsymbol{y}^\perp]_{2:K} \right\| \\
&= \sqrt{1 - \beta^2} \left\| \left( I - \tfrac{1}{K-1} \mathbb{1}\mathbb{1}^\top \right) [\boldsymbol{y}^\perp]_{2:K} \right\| \le \sqrt{1 - \beta^2} \| [\boldsymbol{y}^\perp]_{2:K} \| \le \sqrt{1 - \beta^2} \,.
\end{aligned}
$$

$\square$

*Proof of Lemma 5.* Without loss of generality, we prove this lemma for $k = 1$. We define

$$
T_1 = \inf\{t > T^*_{j,k} : \mathcal{B}_k < \tfrac{\sqrt{2}}{2}\},
$$
$$
T_2 = \inf\{t > T^*_{j,k} : |\mathcal{I}^{\boldsymbol{w}_j}_k| \ne |\mathcal{I}_k| \text{ or } |\mathcal{I}^{\boldsymbol{w}_j}_{k'}| \ne 0\}\,.
$$

We need to show that $\min\{T_1, T_2\} = \infty$. We derive a contradiction by assuming it is finite.

**Case one:** $\min\{T_1, T_2\} = T_1$ **is finite**

Assuming $\min\{T_1, T_2\} = T_1$ is finite, our primary focus is the angular dynamics of $\frac{\boldsymbol{v}_j}{\|\boldsymbol{v}_j\|}, \forall j \in \mathcal{N}_1$,

$$
\tfrac{\mathrm{d}}{\mathrm{d}t} \tfrac{\boldsymbol{v}_j}{\|\boldsymbol{v}_j\|} = \Pi^\perp_{\boldsymbol{v}_j} \sum_{i \in \mathcal{I}_1} \left\langle \boldsymbol{x}_i, \tfrac{\boldsymbol{w}_j}{\|\boldsymbol{w}_j\|} \right\rangle (\boldsymbol{e}_1 - \hat{\boldsymbol{y}}_i)\,,
$$

and in particular those of its alignment with pseudo-label $\tilde{\boldsymbol{e}}_1$,

$$
\begin{aligned}
\tfrac{\mathrm{d}}{\mathrm{d}t} \mathcal{B}^{\boldsymbol{v}_j}_j &= \left\langle \tilde{\boldsymbol{e}}_1, \tfrac{\mathrm{d}}{\mathrm{d}t} \tfrac{\boldsymbol{v}_j}{\|\boldsymbol{v}_j\|} \right\rangle \\
&= \left\langle \tilde{\boldsymbol{e}}_1, \Pi^\perp_{\boldsymbol{v}_j} \sum_{i \in \mathcal{I}_1} \left\langle \boldsymbol{x}_i, \tfrac{\boldsymbol{w}_j}{\|\boldsymbol{w}_j\|} \right\rangle (\boldsymbol{e}_1 - \hat{\boldsymbol{y}}_i) \right\rangle \\
&= \sum_{i \in \mathcal{I}_1} \left\langle \boldsymbol{x}_i, \tfrac{\boldsymbol{w}_j}{\|\boldsymbol{w}_j\|} \right\rangle \left\langle \tilde{\boldsymbol{e}}_1, \Pi^\perp_{\boldsymbol{v}_j} (\boldsymbol{e}_1 - \hat{\boldsymbol{y}}_i) \right\rangle\,.
\end{aligned}
\tag{53}
$$

We shall focus on the term $\left\langle \tilde{\boldsymbol{e}}_1, \Pi^\perp_{\boldsymbol{v}_j} (\boldsymbol{e}_1 - \hat{\boldsymbol{y}}_i) \right\rangle$. For each $i \in \mathcal{I}_1$, we let $z_{ik} = [\boldsymbol{V}\boldsymbol{W}\boldsymbol{x}_i]_k = \left[ \sum_{j \in \mathcal{N}_1} \boldsymbol{v}_j \boldsymbol{w}_j^\top \boldsymbol{x}_i \right]_k$, then

$$
\begin{aligned}
\boldsymbol{e}_1 - \hat{\boldsymbol{y}}_i &= \begin{bmatrix} 1 \\ 0 \\ \vdots \\ 0 \end{bmatrix} - \begin{bmatrix} \frac{\exp(z_{i1})}{\sum_{k=1}^K \exp(z_{ik})} \\ \frac{\exp(z_{i2})}{\sum_{k=1}^K \exp(z_{ik})} \\ \vdots \\ \frac{\exp(z_{iK})}{\sum_{k=1}^K \exp(z_{ik})} \end{bmatrix} = \begin{bmatrix} \frac{\sum_{k>1} \exp(z_{ik})}{\sum_{k=1}^K \exp(z_{ik})} \\ -\frac{\exp(z_{i2})}{\sum_{k=1}^K \exp(z_{ik})} \\ \vdots \\ -\frac{\exp(z_{iK})}{\sum_{k=1}^K \exp(z_{ik})} \end{bmatrix} \\[2mm]
&\overset{(z_{ik} - z_{i1} := \tilde{z}_{ik})}{=} \begin{bmatrix} \frac{\sum_{k>1} \exp(\tilde{z}_{ik})}{1 + \sum_{k>1} \exp(\tilde{z}_{ik})} \\ -\frac{\exp(\tilde{z}_{i2})}{1 + \sum_{k>1} \exp(\tilde{z}_{ik})} \\ \vdots \\ -\frac{\exp(\tilde{z}_{iK})}{\sum_{k>1} \exp(\tilde{z}_{ik})} \end{bmatrix} \\[2mm]
&= \begin{bmatrix} \frac{\sum_{k>1} \exp(\tilde{z}_{ik})}{1 + \sum_{k>1} \exp(\tilde{z}_{ik})} \\ -\frac{1}{K-1} \frac{\sum_{k>1} \exp(\tilde{z}_{ik})}{1 + \sum_{k>1} \exp(\tilde{z}_{ik})} \\ \vdots \\ -\frac{1}{K-1} \frac{\sum_{k>1} \exp(\tilde{z}_{ik})}{\sum_{k>1} \exp(\tilde{z}_{ik})} \end{bmatrix} + \begin{bmatrix} 0 \\ \frac{-\exp(\tilde{z}_{i2}) + \frac{1}{K-1} \sum_{k>1} \exp(\tilde{z}_{ik})}{1 + \sum_{k>1} \exp(\tilde{z}_{ik})} \\ \vdots \\ \frac{-\exp(\tilde{z}_{iK}) + \frac{1}{K-1} \sum_{k>1} \exp(\tilde{z}_{ik})}{1 + \sum_{k>1} \exp(\tilde{z}_{ik})} \end{bmatrix} \\[2mm]
&= \frac{\sum_{k>1} \exp(\tilde{z}_{ik})}{1 + \sum_{k>1} \exp(\tilde{z}_{ik})} \left( \sqrt{\tfrac{K}{K-1}} \tilde{\boldsymbol{e}}_1 + \begin{bmatrix} 0 \\ \frac{-\exp(\tilde{z}_{i2})}{\sum_{k>1} \exp(\tilde{z}_{ik})} + \frac{1}{K-1} \\ \vdots \\ \frac{-\exp(\tilde{z}_{iK})}{\sum_{k>1} \exp(\tilde{z}_{ik})} + \frac{1}{K-1} \end{bmatrix} \right),
\end{aligned}
\tag{54}
$$

thus we have

$$\left\langle \tilde{\boldsymbol{e}}_1, \Pi^{\perp}_{\boldsymbol{v}_j}(\boldsymbol{e}_1 - \hat{\boldsymbol{y}}_i) \right\rangle$$

$$= \frac{\sum_{k>1}\exp(\tilde{z}_{ik})}{1+\sum_{k>1}\exp(\tilde{z}_{ik})} \left\langle \tilde{\boldsymbol{e}}_1, \left(I - \frac{\boldsymbol{v}_j\boldsymbol{v}_j^\top}{\|\boldsymbol{v}_j\|^2}\right) \left( \sqrt{\frac{K}{K-1}}\tilde{\boldsymbol{e}}_1 + \begin{bmatrix} 0 \\ \frac{-\exp(\tilde{z}_{i2})}{\sum_{k>1}\exp(\tilde{z}_{ik})} + \frac{1}{K-1} \\ \vdots \\ \frac{-\exp(\tilde{z}_{iK})}{\sum_{k>1}\exp(\tilde{z}_{ik})} + \frac{1}{K-1} \end{bmatrix} \right) \right\rangle$$

$$= \frac{\sum_{k>1}\exp(\tilde{z}_{ik})}{1+\sum_{k>1}\exp(\tilde{z}_{ik})} \left( \sqrt{\frac{K}{K-1}}(1 - (\mathcal{B}_j^{\boldsymbol{v}_j})^2) + \left\langle \tilde{\boldsymbol{e}}_1, \left(I - \frac{\boldsymbol{v}_j\boldsymbol{v}_j^\top}{\|\boldsymbol{v}_j\|^2}\right) \begin{bmatrix} 0 \\ \frac{-\exp(\tilde{z}_{i2})}{\sum_{k>1}\exp(\tilde{z}_{ik})} + \frac{1}{K-1} \\ \vdots \\ \frac{-\exp(\tilde{z}_{iK})}{\sum_{k>1}\exp(\tilde{z}_{ik})} + \frac{1}{K-1} \end{bmatrix} \right\rangle \right)$$

$$= \frac{\sum_{k>1}\exp(\tilde{z}_{ik})}{1+\sum_{k>1}\exp(\tilde{z}_{ik})} \left( \sqrt{\frac{K}{K-1}}(1 - (\mathcal{B}_j^{\boldsymbol{v}_j})^2) - \mathcal{B}_j^{\boldsymbol{v}_j} \left( \frac{1}{K-1}\sum_{k>1}\left[\frac{\boldsymbol{v}_j}{\|\boldsymbol{v}_j\|}\right]_k - \sum_{k>1}\frac{\exp(\tilde{z}_{ik})\left[\frac{\boldsymbol{v}_j}{\|\boldsymbol{v}_j\|}\right]_k}{\sum_{k>1}\exp(\tilde{z}_{ik})} \right) \right)$$

$$\overset{\text{(Lemma 10)}}{\geq} \frac{\sum_{k>1}\exp(\tilde{z}_{ik})}{1+\sum_{k>1}\exp(\tilde{z}_{ik})} \left( \sqrt{\frac{K}{K-1}}(1 - (\mathcal{B}_j^{\boldsymbol{v}_j})^2) - \mathcal{B}_j^{\boldsymbol{v}_j}\sqrt{1 - (\mathcal{B}_j^{\boldsymbol{v}_j})^2} \right)$$

$$= \frac{\sum_{k>1}\exp(\tilde{z}_{ik})}{1+\sum_{k>1}\exp(\tilde{z}_{ik})} \sqrt{1 - (\mathcal{B}_j^{\boldsymbol{v}_j})^2} \left( \sqrt{\frac{K}{K-1}}\sqrt{1 - (\mathcal{B}_j^{\boldsymbol{v}_j})^2} - \mathcal{B}_j^{\boldsymbol{v}_j} \right), \tag{55}$$

from which we see that at $t = T_1$, we have

$$\frac{\mathrm{d}}{\mathrm{d}t}\mathcal{B}_j^{\boldsymbol{v}_j}\Big|_{\mathcal{B}_j^{\boldsymbol{v}_j}=\frac{\sqrt{2}}{2}} \geq \sum_{i=\in\mathcal{I}_1} \underbrace{\left\langle \boldsymbol{x}_i, \frac{\boldsymbol{w}_j}{\|\boldsymbol{w}_j\|} \right\rangle}_{\geq 0} \frac{\sum_{k>1}\exp(\tilde{z}_{ik})}{1+\sum_{k>1}\exp(\tilde{z})}\sqrt{\frac{1}{2}}\left(\sqrt{\frac{K}{K-1}}\sqrt{\frac{1}{2}} - \frac{\sqrt{2}}{2}\right) > 0, \tag{56}$$

contradicting the definition of $T_1$.

**Case two: $\min\{T_1, T_2\} = T_2$ is finite**

Assuming $\min\{T_1, T_2\} = T_2$ is finite, we shall focus on the time interval $[T_{j,k}^*, T_2]$, when we have

$$\frac{\mathrm{d}}{\mathrm{d}t}\boldsymbol{w}_j = \sum_{i\in\mathcal{I}_1}\left\langle \boldsymbol{e}_1 - \hat{\boldsymbol{y}}_i, \boldsymbol{v}_j \right\rangle \boldsymbol{x}_i = \sum_{i\in\mathcal{I}_1}\left\langle \boldsymbol{e}_1 - \hat{\boldsymbol{y}}_i, \frac{\boldsymbol{v}_j}{\|\boldsymbol{v}_j\|} \right\rangle \|\boldsymbol{v}_j\|\boldsymbol{x}_i \tag{57}$$

From (54), we have $\forall t \leq T_2$

$$\left\langle \boldsymbol{e}_1 - \hat{\boldsymbol{y}}_i, \frac{\boldsymbol{v}_j}{\|\boldsymbol{v}_j\|} \right\rangle = \frac{\sum_{k>1}\exp(\tilde{z}_{ik})}{1+\sum_{k>1}\exp(\tilde{z})}\left(\sqrt{\frac{K}{K-1}}\mathcal{B}_j^{\boldsymbol{v}_j} + \left(\frac{1}{K-1}\sum_{k>1}\left[\frac{\boldsymbol{v}_j}{\|\boldsymbol{v}_j\|}\right]_k - \sum_{k>1}\frac{\exp(\tilde{z}_{ik})\left[\frac{\boldsymbol{v}_j}{\|\boldsymbol{v}_j\|}\right]_k}{\sum_{k>1}\exp(\tilde{z}_{ik})}\right)\right)$$

$$\overset{\text{(Lemma 10)}}{\geq} \frac{\sum_{k>1}\exp(\tilde{z}_{ik})}{1+\sum_{k>1}\exp(\tilde{z})}\left(\sqrt{\frac{K}{K-1}}\mathcal{B}_j^{\boldsymbol{v}_j} - \sqrt{1 - (\mathcal{B}_j^{\boldsymbol{v}_j})^2}\right)$$

$$\overset{(T_1 \geq T_2)}{\geq} \frac{\sum_{k>1}\exp(\tilde{z}_{ik})}{1+\sum_{k>1}\exp(\tilde{z})}\left(\sqrt{\frac{K}{K-1}}\frac{\sqrt{2}}{2} - \frac{\sqrt{2}}{2}\right) \geq 0. \tag{58}$$

Therefore, by the Fundamental Theorem of Calculus, we have

$$\boldsymbol{w}_j(T_2) = \boldsymbol{w}_j(T_{j,k}^*) + \sum_{i\in\mathcal{I}_1}\underbrace{\left(\int_{T_{j,k}^*}^{T_2}\left\langle \boldsymbol{e}_1 - \hat{\boldsymbol{y}}_i, \frac{\boldsymbol{v}_j}{\|\boldsymbol{v}_j\|} \right\rangle \|\boldsymbol{v}_j\|\right)}_{\geq 0}\boldsymbol{x}_i, \tag{59}$$

which ensures that $|\mathcal{I}_k^{\boldsymbol{w}_j(T_2)}| = |\mathcal{I}_k|$ and $|\mathcal{I}_{k'}^{\boldsymbol{w}_j(T_2)}| = 0$, contradicting to the definition of $T_2$.

Therefore, the proof is finished by the fact that $\min\{T_1, T_2\}$ cannot be finite. $\qquad\square$

### C.4 Proof of Proposition 1

As we have discussed in Appendix C.3, it suffices to prove Proposition 3.

*Proof of Proposition 3.* We have shown that before $\min\{T_{j,k}^*, T\}$, the properties of the weights in Lemma 4 hold. We consider a sufficiently small $\epsilon$ such that

$$\frac{16}{\sqrt{K}}\epsilon n^2 X_{\max}^3 \sqrt{h} \leq \frac{1}{2}\sqrt{\frac{K-1}{K}}\left(1 - \frac{1}{2(K-1)}\right)\mu_s X_{\min}^2 \zeta\,, \tag{60}$$

and

$$\frac{2X_{\max}|\mathcal{I}_k|}{\sqrt{\frac{K-1}{K}}\left(1 - \frac{1}{2(K-1)}\right)\mu_s X_{\min}^2 \zeta} \leq \frac{1}{4nX_{\max}}\log\frac{1}{\sqrt{h}\epsilon} \tag{61}$$

Then we show that $\min\{T_{j,k}^*, T\} = T_{j,k}^*$ by contradiction: Suppose that $T \leq T_{j,k}^*$, then during $[0, T]$, we have, from (45),

$$\frac{\mathrm{d}}{\mathrm{d}t}\mathcal{A}_k \geq \sqrt{\frac{K-1}{K}}\mathcal{B}_k\left(\left\|\sum_{i\in\mathcal{I}_k:\alpha_i>0} x_i\right\|^2 - \mathcal{A}_k^2\right) + \mathcal{O}(\epsilon)\,,$$

$$\geq \sqrt{\frac{K-1}{K}}\left(1 - \frac{1}{2(K-1)}\right)\mu_s X_{\min}^2 \zeta - \frac{16}{\sqrt{K}}\epsilon n^2 X_{\max}^3 \sqrt{h}$$

$$\overset{(60)}{\geq} \frac{1}{2}\sqrt{\frac{K-1}{K}}\left(1 - \frac{1}{2(K-1)}\right)\mu_s X_{\min}^2 \zeta. \tag{62}$$

Then by the Fundamental Theorem of Calculus, we have

$$\mathcal{A}_k(T) \geq \mathcal{A}(0) + \frac{T}{2}\sqrt{\frac{K-1}{K}}\left(1 - \frac{1}{2(K-1)}\right)\mu_s X_{\min}^2 \zeta \overset{(61)}{\geq} \mathcal{A}(0) + X_{\max}|\mathcal{I}_k|\,, \tag{63}$$

which is a contradiction, knowing that $\mathcal{A}_k$ cannot exceed $X_{\max}|\mathcal{I}_k|$. Therefore, we must have $\min\{T_{j,k}^*, T\} = T_{j,k}^*$ and $T_{j,k}^* \leq T = \frac{1}{4nX_{\max}}\log\frac{1}{\sqrt{h}\epsilon}$ is finite. Then the rest of the Proposition 3 follows Lemma 5. $\qquad\square$

# D Asymptotic Convergence Analysis under Multi-class Orthogonally Separable Data

## D.1 Basic results upon inter-class separation

With the loss decomposition upon inter-class separation, for which we have shown to persist after $T^* = \max_{j,k} T^*_{j,k}$,

$$\mathcal{L}(\boldsymbol{\theta}) = \sum_{k=1}^K \mathcal{L}_{\text{CE}}\left(\boldsymbol{Y}_k, \boldsymbol{V}_k \boldsymbol{W}_k^\top \boldsymbol{X}_k\right) = \sum_{k=1}^K \sum_{i=1}^{n_k} \ell_{\text{CE}}\left(\boldsymbol{y}_{k,i}, \boldsymbol{V}_k \boldsymbol{W}_k^\top \boldsymbol{x}_{k,i}\right), \tag{64}$$

It suffices to study the following GF on $\sum_{i=1}^{n_k} \ell_{\text{CE}}\left(\boldsymbol{y}_{k,i}, \boldsymbol{V}_k \boldsymbol{W}_k^\top \boldsymbol{x}_{k,i}\right)$:

$$\dot{\boldsymbol{W}}_k = \boldsymbol{X}_k (\boldsymbol{Y}_k - \hat{\boldsymbol{Y}}_k)^\top \boldsymbol{V}_k$$

$$\dot{\boldsymbol{V}}_k = (\boldsymbol{Y}_k - \hat{\boldsymbol{Y}}_k) \boldsymbol{X}_k^\top \boldsymbol{W}_k$$

$$\text{where } \hat{\boldsymbol{Y}}_k = \text{SoftmaxCol}(\boldsymbol{V}_k \boldsymbol{W}_k^\top \boldsymbol{X}_k) \tag{65}$$

The following basic results can be obtained from [21, 27]

1. $\|\boldsymbol{W}_k\|_F, \|\boldsymbol{V}_k\| \to \infty$; $\bar{\boldsymbol{W}}_k, \bar{\boldsymbol{V}}_k$ exist;
2. $\bar{\boldsymbol{V}}_k^\top \bar{\boldsymbol{V}}_k - \bar{\boldsymbol{W}}_k^\top \bar{\boldsymbol{W}}_k = 0$;
3. $\bar{\boldsymbol{W}}_k, \bar{\boldsymbol{V}}_k$ is a KKT point of

$$\min_{\boldsymbol{W}_k, \boldsymbol{V}_k} \|\boldsymbol{W}_k\|_F^2 + \|\boldsymbol{V}_k\|_F^2, \quad s.t. \ [\boldsymbol{V}_k \boldsymbol{W}_k^\top \boldsymbol{x}_i]_k - [\boldsymbol{V}_k \boldsymbol{W}_k^\top \boldsymbol{x}_i]_l \geq 1, \forall i \in \mathcal{I}_k, \forall l \neq k \tag{66}$$

## D.2 Proof of Proposition 2

Our proof of Proposition 2 follows the same strategy as those in [17, 21], with the major difference being that we are handling cross-entropy loss, in which we provide an extension of Lemma 2.11 in [21], stated as Lemma 13. Lemma 13 is central to our proof.

**Lemma 11.** *Let* $\gamma := \min_{1 \leq k \leq K} \gamma_k$, *where* $\gamma_k := \min_{i \in \mathcal{I}_k} \langle \bar{\boldsymbol{u}}_{\infty,k}, \boldsymbol{x}_i \rangle$, *then* $\gamma \geq \mu_s X_{\min}$.

*Proof.* For any $1 \leq k \leq K$, $\bar{\boldsymbol{u}}_{\infty,k} = \sum_{i \in \mathcal{I}_k} a_i \boldsymbol{x}_i$, for some $a_i \geq 0$, then immediately we have, $\forall i \in \mathcal{I}_k$

$$\langle \bar{\boldsymbol{u}}_{\infty,k}, \boldsymbol{x}_i \rangle = \langle \sum_{i' \in \mathcal{I}_k} a_{i'} \boldsymbol{x}_{i'}, \boldsymbol{x}_i \rangle \geq \mu_s \| \sum_{i' \in \mathcal{I}_k} a_{i'} \boldsymbol{x}_{i'} \| \|\boldsymbol{x}_i\| \geq \mu_s X_{\min}. \tag{67}$$

$\square$

**Lemma 12.** $\gamma^\perp := \min_{k \in [K]} \min_{\|\xi\|=1, \xi \perp \bar{\boldsymbol{u}}_k} \max_{i \in \mathcal{I}_k} \langle \xi, \boldsymbol{x}_i \rangle > 0$

*Proof.* This result is from Lemma 2.10 in [21]. Note that the referenced Lemma requires an additional assumption that the support vectors of $\boldsymbol{x}_i, i \in \mathcal{I}_k$ span the ambient space, but the authors of [21] have commented that this condition can be relaxed to the case that the span of support vectors is the span of $\boldsymbol{x}_i, i \in \mathcal{I}_k$, which is true here given the positive correlations between $\boldsymbol{x}_i, i \in \mathcal{I}_k$. $\square$

**Lemma 13.** *Given some* $\boldsymbol{\Theta} = [\boldsymbol{\theta}_1, \cdots, \boldsymbol{\theta}_K] \in \mathbb{R}^{D \times K}$ *and some* $1 \leq k \leq K$. *If it holds that* $\exists k' \neq k$, $(\boldsymbol{\theta}_k - \boldsymbol{\theta}_{k'})^\top \bar{\boldsymbol{u}}_{\infty,k} > 0$ *and* $\|\Pi^\perp_{\bar{\boldsymbol{u}}_\infty} (\boldsymbol{\theta}_k - \boldsymbol{\theta}_{k'})\|$ *sufficiently large, then* $\text{tr}\left((\boldsymbol{e}_k \mathbb{1}_n^T - \hat{\boldsymbol{Y}})^\top \boldsymbol{\Theta}^\top \Pi^\perp_{\bar{\boldsymbol{u}}_\infty} \boldsymbol{X}\right) \leq 0$, *where* $\hat{\boldsymbol{Y}} = \text{SoftmaxCol}(\boldsymbol{\Theta}^\top \boldsymbol{X})$.

*Proof.* It suffices to prove the case when $k = 1$ (We discuss the others at the end of the proof). We start by the following derivations:

$$\text{tr}\left((\boldsymbol{e}_1 \mathbb{1}_n^T - \hat{\boldsymbol{Y}})^\top \boldsymbol{\Theta}^\top \Pi^\perp_{\bar{\boldsymbol{u}}_\infty} \boldsymbol{X}\right)$$

$$= \sum_{i=1}^n \left\langle [\boldsymbol{e}_1 \mathbb{1}_n^T - \hat{\boldsymbol{Y}}]_{:,i}, \left[\boldsymbol{\Theta}^\top \Pi^\perp_{\bar{\boldsymbol{u}}_\infty} \boldsymbol{X}\right]_{:,i} \right\rangle$$

$$= \sum_{i=1}^n \left\langle e_1 - \hat{y}_i, \Theta^\top \Pi^\perp_{\bar{u}_\infty} x_i \right\rangle$$

$$= \sum_{i=1}^n \left( (1 - \hat{y}_{i1}) \theta_1^\top \Pi^\perp_{\bar{u}_\infty} x_i + \sum_{k \neq 1} (-\hat{y}_{ik}) \theta_k^\top \Pi^\perp_{\bar{u}_\infty} x_i \right)$$

$$= \sum_{i=1}^n \frac{((\sum_{k=1}^K \exp(\theta_k^\top x_i)) - \exp(\theta_1^\top x_i)) \theta_1^\top \Pi^\perp_{\bar{u}_\infty} x_i + \sum_{k \neq 1} (-\exp(\theta_k^\top x_i)) \theta_k^\top \Pi^\perp_{\bar{u}_\infty} x_i}{\sum_{k=1}^K \exp(\theta_k^\top x_i)}$$

$$= \sum_{i=1}^n \frac{\sum_{k \neq 1} \exp(\theta_k^\top x_i)(\theta_1 - \theta_k)^\top \Pi^\perp_{\bar{u}_\infty} x_i}{\sum_{k=1}^K \exp(\theta_k^\top x_i)}$$

$$= \sum_{i=1}^n \frac{\sum_{k \neq 1} \exp(-(\theta_1 - \theta_k)^\top x_i)(\theta_1 - \theta_k)^\top \Pi^\perp_{\bar{u}_\infty} x_i}{1 + \sum_{k \neq 1} \exp(-(\theta_1 - \theta_k)^\top x_i)}$$

$$= \sum_{k \neq 1} \sum_{i=1}^n \frac{\exp(-(\theta_1 - \theta_k)^\top x_i)(\theta_1 - \theta_k)^\top \Pi^\perp_{\bar{u}_\infty} x_i}{1 + \sum_{k \neq 1} \exp(-(\theta_1 - \theta_k)^\top x_i)} . \tag{68}$$

For the $k$-th summand, let

$$i_k^* = \arg\max_i (\theta_1 - \theta_k)^\top \Pi^\perp_{\bar{u}_\infty} x_i , \tag{69}$$

we have

$$\sum_{i=1}^n \frac{\exp(-(\theta_1 - \theta_k)^\top x_i)(\theta_1 - \theta_k)^\top \Pi^\perp_{\bar{u}_\infty} x_i}{1 + \sum_{k \neq 1} \exp(-(\theta_1 - \theta_k)^\top x_i)}$$

$$\leq -\frac{\exp(-(\theta_1 - \theta_k)^\top x_{i_k^*})(-(\theta_1 - \theta_k)^\top \Pi^\perp_{\bar{u}_\infty} x_{i_k^*})}{1 + \sum_{k \neq 1} \exp(-(\theta_1 - \theta_k)^\top x_{i_k^*})}$$

$$+ \sum_{i:(\theta_1 - \theta_k)^\top \Pi^\perp_{\bar{u}_\infty} x_i \geq 0} \frac{\exp(-(\theta_1 - \theta_k)^\top x_i)(\theta_1 - \theta_k)^\top \Pi^\perp_{\bar{u}_\infty} x_i}{1 + \sum_{k \neq 1} \exp(-(\theta_1 - \theta_k)^\top x_i)} .$$

One can upper bound these two terms separately as follows:

$$- \frac{\exp(-(\theta_1 - \theta_k)^\top x_{i_k^*})(-(\theta_1 - \theta_k)^\top \Pi^\perp_{\bar{u}_\infty} x_{i_k^*})}{1 + \sum_{k \neq 1} \exp(-(\theta_1 - \theta_k)^\top x_{i_k^*})}$$

$$\leq -\frac{1}{K} \exp(-(\theta_1 - \theta_k)^\top x_{i_k^*})(-(\theta_1 - \theta_k)^\top \Pi^\perp_{\bar{u}_\infty} x_{i_k^*})$$

$$= -\frac{1}{K} \exp(-(\theta_1 - \theta_k)^\top \Pi_{\bar{u}_\infty} x_{i_k^*}) \exp(-(\theta_1 - \theta_k)^\top \Pi^\perp_{\bar{u}_\infty} x_{i_k^*})(-(\theta_1 - \theta_k)^\top \Pi^\perp_{\bar{u}_\infty} x_{i_k^*})$$

$$\leq -\frac{1}{K} \exp(-\gamma(\theta_1 - \theta_k)^\top \bar{u}_\infty) \exp(-(\theta_1 - \theta_k)^\top \Pi^\perp_{\bar{u}_\infty} x_{i_k^*})(-(\theta_1 - \theta_k)^\top \Pi^\perp_{\bar{u}_\infty} x_{i_k^*})$$

$$\leq -\frac{1}{K} \exp\left(-\gamma(\theta_1 - \theta_k)^\top \bar{u}_\infty\right) \exp\left(\gamma^\perp \|\Pi^\perp_{\bar{u}_\infty}(\theta_1 - \theta_k)\|\right) \gamma^\perp \|\Pi^\perp_{\bar{u}_\infty}(\theta_1 - \theta_k)\| , \tag{70}$$

and for the second term,

$$\sum_{i:(\theta_1 - \theta_k)^\top \Pi^\perp_{\bar{u}_\infty} x_i \geq 0} \frac{\exp(-(\theta_1 - \theta_k)^\top x_i)(\theta_1 - \theta_k)^\top \Pi^\perp_{\bar{u}_\infty} x_i}{1 + \sum_{k \neq 1} \exp(-(\theta_1 - \theta_k)^\top x_i)}$$

$$\leq \sum_{i:(\theta_1 - \theta_k)^\top \Pi^\perp_{\bar{u}_\infty} x_i \geq 0} \exp(-(\theta_1 - \theta_k)^\top x_i)(\theta_1 - \theta_k)^\top \Pi^\perp_{\bar{u}_\infty} x_i$$

$$\leq \sum_{i:(\theta_1 - \theta_k)^\top \Pi^\perp_{\bar{u}_\infty} x_i \geq 0} \exp(-(\theta_1 - \theta_k)^\top \Pi_{\bar{u}_\infty} x_i) \exp(-(\theta_1 - \theta_k)^\top \Pi^\perp_{\bar{u}_\infty} x_i)(\theta_1 - \theta_k)^\top \Pi^\perp_{\bar{u}_\infty} x_i$$

$$\leq \sum_{i:(\theta_1 - \theta_k)^\top \Pi^\perp_{\bar{u}_\infty} x_i \geq 0} \exp(-\gamma(\theta_1 - \theta_k)^\top \bar{u}_\infty) \exp(-(\theta_1 - \theta_k)^\top \Pi^\perp_{\bar{u}_\infty} x_i)(\theta_1 - \theta_k)^\top \Pi^\perp_{\bar{u}_\infty} x_i$$

$$\leq \sum_{i:(\theta_1 - \theta_k)^\top \Pi^\perp_{\bar{u}_\infty} x_i \geq 0} \exp(-\gamma(\theta_1 - \theta_k)^\top \bar{u}_\infty) \frac{1}{e}$$

$$\leq \exp(-\gamma(\theta_1 - \theta_k)^\top \bar{u}_\infty) \frac{n}{e} \tag{71}$$

Therefore, putting (68)(70)(71) together, we have

$$\text{tr}\left((e_1 \mathbb{1}_n^T - \hat{Y})^\top \Theta^\top \Pi^\perp_{\bar{u}_\infty} X\right)$$

$$\leq \sum_k \exp(-\gamma(\theta_1 - \theta_k)^\top \bar{u}_\infty) \left(\frac{n}{e} - \frac{1}{K} \exp\left(\gamma^\perp \|\Pi^\perp_{\bar{u}_\infty}(\theta_1 - \theta_k)\|\right) \gamma^\perp \|\Pi^\perp_{\bar{u}_\infty}(\theta_1 - \theta_k)\|\right)$$

$$= \exp(-\gamma(\theta_1 - \theta_{k'})^\top \bar{u}_\infty) \left(\frac{n}{e} - \frac{1}{K} \exp\left(\gamma^\perp \|\Pi^\perp_{\bar{u}_\infty}(\theta_1 - \theta_{k'})\|\right) \gamma^\perp \|\Pi^\perp_{\bar{u}_\infty}(\theta_1 - \theta_{k'})\|\right)$$

$$+ \sum_{k \neq k'} \exp(-\gamma(\theta_1 - \theta_k)^\top \bar{u}_\infty) \frac{n}{e}$$

$$= \frac{\exp(-\gamma(\boldsymbol{\theta}_1 - \boldsymbol{\theta}_{k'})^\top \bar{\boldsymbol{u}}_\infty)}{\sum_{k \neq k'} \exp(-\gamma(\boldsymbol{\theta}_1 - \boldsymbol{\theta}_k)^\top \bar{\boldsymbol{u}}_\infty)} \left( \frac{n}{e} - \frac{1}{K} \exp\left( \gamma^\perp \|\Pi^\perp_{\bar{\boldsymbol{u}}_\infty}(\boldsymbol{\theta}_1 - \boldsymbol{\theta}_{k'})\| \right) \gamma^\perp \|\Pi^\perp_{\bar{\boldsymbol{u}}_\infty}(\boldsymbol{\theta}_1 - \boldsymbol{\theta}_{k'})\| \right) + \frac{n}{e}$$
$$\leq 0\,,$$

when $\|\Pi^\perp_{\bar{\boldsymbol{u}}_\infty}(\boldsymbol{\theta}_1 - \boldsymbol{\theta}_{k'})\|$ is sufficiently large.

If $k \neq 1$, consider the permutation matrix $\boldsymbol{P}_{1 \leftrightarrow k}$ that swap the 1-st and $k$-th rows/columns of a matrix, then

$$\operatorname{tr}\left( (\boldsymbol{e}_k \mathbb{1}_n^T - \hat{\boldsymbol{Y}}_k)^\top \boldsymbol{\Theta}^\top \Pi^\perp_{\bar{\boldsymbol{u}}_\infty} \boldsymbol{X} \right)$$
$$= \operatorname{tr}\left( (\boldsymbol{e}_k \mathbb{1}_n^T - \hat{\boldsymbol{Y}}_k)^\top \boldsymbol{P}_{1 \leftrightarrow k} \boldsymbol{\Theta}^\top \Pi^\perp_{\bar{\boldsymbol{u}}_\infty} \boldsymbol{X} \right)$$
$$= \operatorname{tr}\left( (\boldsymbol{P}_{1 \leftrightarrow k} \boldsymbol{e}_k \mathbb{1}_n^T - \boldsymbol{P}_{1 \leftrightarrow k} \hat{\boldsymbol{Y}}_k)^\top \boldsymbol{P}_{1 \leftrightarrow k} \boldsymbol{\Theta}^\top \Pi^\perp_{\bar{\boldsymbol{u}}_\infty} \boldsymbol{X} \right)$$
$$= \operatorname{tr}\left( (\boldsymbol{e}_1 \mathbb{1}_n^T - \operatorname{SoftmaxCol}(\boldsymbol{P}_{1 \leftrightarrow k} \boldsymbol{\Theta}^\top \boldsymbol{X}))^\top \boldsymbol{P}_{1 \leftrightarrow k} \boldsymbol{\Theta}^\top \Pi^\perp_{\bar{\boldsymbol{u}}_\infty} \boldsymbol{X} \right)\,.$$

Following the derivations for the case $k = 1$ gives the desired result. $\square$

*Proof of Proposition 2.* Without loss of generality, we prove the case of $k = 1$. The existence $\{\bar{\boldsymbol{W}}_1, \bar{\boldsymbol{V}}_1\}$ is by [27]. We first show that $\bar{\boldsymbol{W}}_1 \propto \boldsymbol{u}_1 \boldsymbol{g}_1^\top$, which is equivalent to the statement that $\frac{\|\Pi^\perp_{\boldsymbol{u}_1} \bar{\boldsymbol{W}}_1\|_F}{\|\bar{\boldsymbol{W}}_1\|_F} = 0$, and we prove by contradiction. Suppose $\frac{\|\Pi^\perp_{\boldsymbol{u}_1} \bar{\boldsymbol{W}}_1\|_F}{\|\bar{\boldsymbol{W}}_1\|_F} > 0$, which necessarily implies that $\exists \rho > 0$ such that $\frac{\|\Pi^\perp_{\boldsymbol{u}_1} \bar{\boldsymbol{W}}_1 \bar{\boldsymbol{W}}_1^\top\|_F}{\|\bar{\boldsymbol{W}}_1 \bar{\boldsymbol{W}}_1^\top\|_F} = \rho$, then for any $\epsilon > 0, M > 0$ exists $T_{\epsilon, M} > 0$ such that $\forall t \geq T_{\epsilon, M}$, we have $\|\frac{\bar{\boldsymbol{W}}_1 \bar{\boldsymbol{V}}_1^\top}{\|\bar{\boldsymbol{W}}_1 \bar{\boldsymbol{V}}_1^\top\|_F} - \frac{\boldsymbol{W}_1 \boldsymbol{V}_1^\top}{\|\boldsymbol{W}_1 \boldsymbol{V}_1^\top\|_F}\| \leq \epsilon$ and $\|\boldsymbol{W}_1 \boldsymbol{V}_1^\top\|_F \geq M$. We will make clear the choice of $\epsilon$ and $M$ later.

Consider the time derivative of $\|\Pi^\perp_{\bar{\boldsymbol{u}}_\infty} \boldsymbol{W}\|_F^2$:

$$\frac{\mathrm{d}}{\mathrm{d}t} \|\Pi^\perp_{\bar{\boldsymbol{u}}_\infty} \boldsymbol{W}_1\|_F^2 = 2\operatorname{tr}\left( \boldsymbol{W}_1^\top \Pi^\perp_{\bar{\boldsymbol{u}}_\infty} \frac{\mathrm{d}}{\mathrm{d}t} \boldsymbol{W}_1 \right)$$
$$= 2\operatorname{tr}\left( \boldsymbol{W}_1^\top \Pi^\perp_{\bar{\boldsymbol{u}}_\infty} \boldsymbol{X} (\boldsymbol{e}_1 \mathbb{1}_n^T - \hat{\boldsymbol{Y}})^\top \boldsymbol{V} \right)$$
$$= 2\operatorname{tr}\left( (\boldsymbol{e}_1 \mathbb{1}_n^T - \hat{\boldsymbol{Y}})^\top \boldsymbol{V}_1 \boldsymbol{W}_1^\top \Pi^\perp_{\bar{\boldsymbol{u}}_\infty} \boldsymbol{X} \right)$$

We would like to use the result in Lemma 13 so we should examine:

$$\|\Pi^\perp_{\bar{\boldsymbol{u}}_\infty} \bar{\boldsymbol{W}}_1 \bar{\boldsymbol{V}}_1^T\|_F^2 = \operatorname{tr}(\Pi^\perp_{\bar{\boldsymbol{u}}_\infty} \bar{\boldsymbol{W}}_1 \bar{\boldsymbol{V}}_1^T \bar{\boldsymbol{V}}_1 \bar{\boldsymbol{W}}_1^\top)$$
$$= \operatorname{tr}(\Pi^\perp_{\bar{\boldsymbol{u}}_\infty} \bar{\boldsymbol{W}}_1 \bar{\boldsymbol{W}}_1^\top \bar{\boldsymbol{W}}_1 \bar{\boldsymbol{W}}_1^\top) = \rho \|\bar{\boldsymbol{W}}_1 \bar{\boldsymbol{W}}_1^\top\|_F > 0\,. \tag{72}$$

Therefore, $\exists \delta > 0, k \neq 1$ such that $\|\Pi^\perp_{\bar{\boldsymbol{u}}_\infty} \bar{\boldsymbol{W}}_1 \bar{\boldsymbol{V}}_1^T (\boldsymbol{e}_1 - \boldsymbol{e}_k)\|^2 = \delta$, otherwise, $\|\Pi^\perp_{\bar{\boldsymbol{u}}_\infty} \bar{\boldsymbol{W}}_1 \bar{\boldsymbol{V}}_1^T (\boldsymbol{e}_1 - \boldsymbol{e}_k)\| = 0, \forall k \neq 1$, which can not happen. Since $\boldsymbol{e}_1 - \boldsymbol{e}_k, k \neq 1$ spans a $k-1$-dimensional subspace orthogonal to $\frac{\mathbb{1}}{\sqrt{K}}$, the projection of $\Pi^\perp_{\bar{\boldsymbol{u}}_\infty} \bar{\boldsymbol{W}}_1 \bar{\boldsymbol{V}}_1^T$ onto this subspace is zero suggests $\Pi^\perp_{\bar{\boldsymbol{u}}_\infty} \bar{\boldsymbol{W}}_1 \bar{\boldsymbol{V}}_1^T$ is rank-1 and all columns of $\boldsymbol{V}_k$ are aligned with $\frac{\mathbb{1}}{\sqrt{K}}$, which contradicts our alignment result in Lemma 4 (these columns must have at least $\frac{\sqrt{2}}{2}$ cosine alignment with $\tilde{\boldsymbol{e}}_1$).

Then $\forall t \geq T_{\epsilon, M}$, and for the $k$ such that $\|\Pi^\perp_{\bar{\boldsymbol{u}}_\infty} \bar{\boldsymbol{W}}_1 \bar{\boldsymbol{V}}_1^T (\boldsymbol{e}_1 - \boldsymbol{e}_k)\|^2 = \delta$, we have

$$\|\Pi^\perp_{\bar{\boldsymbol{u}}_\infty} \boldsymbol{W}_1 \boldsymbol{V}_1^T (\boldsymbol{e}_1 - \boldsymbol{e}_k)\|^2$$
$$= \|\Pi^\perp_{\bar{\boldsymbol{u}}_\infty} \frac{\boldsymbol{W}_1 \boldsymbol{V}_1^T}{\|\boldsymbol{W}_1 \boldsymbol{V}_1^T\|_F} (\boldsymbol{e}_1 - \boldsymbol{e}_k)\|^2 \|\boldsymbol{W}_1 \boldsymbol{V}_1^T\|_F^2$$
$$\geq \left( \|\Pi^\perp_{\bar{\boldsymbol{u}}_\infty} \frac{\bar{\boldsymbol{W}}_1 \bar{\boldsymbol{V}}_1^T}{\|\bar{\boldsymbol{W}}_1 \bar{\boldsymbol{V}}_1^T\|_F} (\boldsymbol{e}_1 - \boldsymbol{e}_k)\| - \sqrt{2}\epsilon \right)^2 \|\boldsymbol{W}_1 \boldsymbol{V}_1^T\|_F^2$$
$$\geq \left( \|\Pi^\perp_{\bar{\boldsymbol{u}}_\infty} \frac{\bar{\boldsymbol{W}}_1 \bar{\boldsymbol{V}}_1^T}{\|\bar{\boldsymbol{W}}_1 \bar{\boldsymbol{V}}_1^T\|_F} (\boldsymbol{e}_1 - \boldsymbol{e}_k)\| - \sqrt{2}\epsilon \right)^2 M^2 \tag{73}$$

Choose sufficiently small $\epsilon$ and sufficiently large $M$, we ensure that $\|\Pi^\perp_{\bar{\boldsymbol{u}}_\infty} \boldsymbol{W}_1 \boldsymbol{V}_1^T (\boldsymbol{e}_1 - \boldsymbol{e}_k)\|$ is sufficiently large to apply Lemma 13 so that $\frac{\mathrm{d}}{\mathrm{d}t} \|\Pi^\perp_{\bar{\boldsymbol{u}}_\infty} \boldsymbol{W}_1\|_F^2 = 2\operatorname{tr}\left( (\boldsymbol{e}_1 \mathbb{1}_n^T - \hat{\boldsymbol{Y}})^\top \boldsymbol{V}_1 \boldsymbol{W}_1^\top \Pi^\perp_{\bar{\boldsymbol{u}}_\infty} \boldsymbol{X} \right) \leq 0.$

On the other hand, $\|\boldsymbol{W}_k\|_F^2 \to \infty$, contradicting our assumption that $\frac{\|\Pi_{\boldsymbol{u}_1}^{\perp} \bar{\boldsymbol{W}}_1\|_F}{\|\bar{\boldsymbol{W}}_1\|_F} > 0$. This proves that $\bar{\boldsymbol{W}}_1 \propto \boldsymbol{u}_1 \boldsymbol{g}_1^{\top}$.

By balancedness $\bar{\boldsymbol{V}}_1^{\top} \bar{\boldsymbol{V}}_1 - \bar{\boldsymbol{W}}_1^{\top} \bar{\boldsymbol{W}}_1 = 0$, we know that $\bar{\boldsymbol{V}}_1 \propto \bar{\boldsymbol{v}}_1 \boldsymbol{g}_1^{\top}$ for some $\bar{\boldsymbol{v}}_1 \in \mathbb{S}^{K-1}$. It remains to show that $\bar{\boldsymbol{v}}_1 = \tilde{\boldsymbol{e}}_1$, which is proved by again contradiction.

Suppose $\bar{\boldsymbol{v}}_1 \neq \tilde{\boldsymbol{e}}_1$, then $\exists k^*$ such that $[\bar{\boldsymbol{v}}_1]_{k^*} \geq [\bar{\boldsymbol{v}}_1]_k, \forall k \neq k^*, k \neq 1$, and not all equalities can be obtained. As a results, consider $[0 \quad -\exp(\tilde{z}_{i2}) \quad \cdots \quad -\exp(\tilde{z}_{iK})]/\sum_{k>1}\exp(\tilde{z}_{ik})$ that appeared in (55), it converges to $\boldsymbol{e}_{k^*}, \forall i \in [n]$. Based on this, for any $\epsilon_1, \epsilon_2, \exists T_{\epsilon_1, \epsilon_2}$ such that $\forall t > T_{\epsilon_1, \epsilon_2}$, we have $\max_{j \in \mathcal{N}_1} \|\frac{\boldsymbol{v}_j}{\|\boldsymbol{v}_j\|} - \bar{\boldsymbol{v}}_1\| \leq \epsilon_1$ and $\max_{i \in [n]} \|[0 \quad -\exp(\tilde{z}_{i2}) \quad \cdots \quad -\exp(\tilde{z}_{iK})]/\sum_{k>1}\exp(\tilde{z}_{ik}) - \boldsymbol{e}_{k^*}\| \leq \epsilon_2$. Therefore, for some $j \in [h]$ and $t > T_{\epsilon_1, \epsilon_2}$, we have, from (55)

$$\frac{\mathrm{d}}{\mathrm{d}t} \mathcal{B}_j^{\boldsymbol{v}_j}$$

$$= \left\langle \tilde{\boldsymbol{e}}_1, \frac{\mathrm{d}}{\mathrm{d}t} \frac{\boldsymbol{v}_j}{\|\boldsymbol{v}_j\|} \right\rangle$$

$$= \left\langle \tilde{\boldsymbol{e}}_1, \Pi_{\boldsymbol{v}_j}^{\perp} \sum_{i \in \mathcal{I}_1} \left\langle \boldsymbol{x}_i, \frac{\boldsymbol{w}_j}{\|\boldsymbol{w}_j\|} \right\rangle (\boldsymbol{e}_1 - \hat{\boldsymbol{y}}_i) \right\rangle$$

$$= \sum_{i \in \mathcal{I}_1} \left\langle \boldsymbol{x}_i, \frac{\boldsymbol{w}_j}{\|\boldsymbol{w}_j\|} \right\rangle \left\langle \tilde{\boldsymbol{e}}_1, \Pi_{\boldsymbol{v}_j}^{\perp} (\boldsymbol{e}_1 - \hat{\boldsymbol{y}}_i) \right\rangle$$

$$= \sum_{i \in \mathcal{I}_1} \frac{\left\langle \boldsymbol{x}_i, \frac{\boldsymbol{w}_j}{\|\boldsymbol{w}_j\|} \right\rangle \sum_{k>1} \exp(\tilde{z}_{ik})}{1 + \sum_{k>1} \exp(\tilde{z}_{ik})} \left( \sqrt{\frac{K}{K-1}} (1 - (\mathcal{B}_j^{\boldsymbol{v}_j})^2) \right.$$

$$\left. - \mathcal{B}_j^{\boldsymbol{v}_j} \left( \frac{1}{K-1} \sum_{k>1} \left[ \frac{\boldsymbol{v}_j}{\|\boldsymbol{v}_j\|} \right]_k - \sum_{k>1} \frac{\exp(\tilde{z}_{ik}) \left[ \frac{\boldsymbol{v}_j}{\|\boldsymbol{v}_j\|} \right]_k}{\sum_{k>1} \exp(\tilde{z}_{ik})} \right) \right)$$

$$\geq \sum_{i \in \mathcal{I}_1} \frac{\left\langle \boldsymbol{x}_i, \frac{\boldsymbol{w}_j}{\|\boldsymbol{w}_j\|} \right\rangle \sum_{k>1} \exp(\tilde{z}_{ik})}{1 + \sum_{k>1} \exp(\tilde{z}_{ik})} \left( \sqrt{\frac{K}{K-1}} (1 - (\mathcal{B}_j^{\boldsymbol{v}_j})^2) - \mathcal{B}_j^{\boldsymbol{v}_j} \left( [\bar{\boldsymbol{v}}_1]_{k^*} - \frac{\sum_{k>1} [\bar{\boldsymbol{v}}_1]_k}{K-1} - \epsilon_2 - 2\epsilon_1 \right) \right)$$

$(\epsilon_1, \epsilon_2$ sufficiently small$)$

$$\geq \sum_{i \in \mathcal{I}_1} \frac{\left\langle \boldsymbol{x}_i, \frac{\boldsymbol{w}_j}{\|\boldsymbol{w}_j\|} \right\rangle \sum_{k>1} \exp(\tilde{z}_{ik})}{1 + \sum_{k>1} \exp(\tilde{z}_{ik})} \left( \sqrt{\frac{K}{K-1}} (1 - (\mathcal{B}_j^{\boldsymbol{v}_j})^2) \right) . \tag{74}$$

The right-hand side of (74) is positive and $\Theta(\frac{1}{t})$, by the fact that weight $\|\boldsymbol{W}_1\|, \|\boldsymbol{V}_1\|$ grow at a rate $\Theta(\log(t))$. Therefore (74) suggests the divergence of $\mathcal{B}_j^{\boldsymbol{v}_j}$, a contradiction.

Finally, we have shown $\bar{\boldsymbol{W}}_1 \propto \boldsymbol{u}_1 \boldsymbol{g}_1^{\top}$ and $\bar{\boldsymbol{V}}_1 \propto \tilde{\boldsymbol{e}}_1 \boldsymbol{g}_1^{\top}$, and the same for other $k$. The choices of $s_k$ are determined by the fact that $\bar{\boldsymbol{W}}_k, \bar{\boldsymbol{V}}_k$ must be a KKT point of (66). $\qquad \square$

