# OpenReview forum: "Neural Collapse under Gradient Flow on Shallow ReLU Networks for Orthogonally Separable Data"
_NeurIPS.cc/2025/Conference — NeurIPS 2025 poster_

### Official Review · Reviewer_kUvC · 2025-06-25

**Clarity:** 3
**Significance:** 3
**Originality:** 3
**Rating:** 4
**Confidence:** 3

**Summary:**

In this article, the authors analyze Neural Collapse (NC) for two-layer ReLU neural networks, optimized using gradient flow, in the context of a classification problem. Their analysis is conducted under certain assumptions on the weight initialization and the data.

Assumption 1, referred to as orthogonal separability, pertains solely to the data and represents a clear limitation of the results presented in the article.

Assumption 2 concerns only the initialization and appears to be relatively weak.

Assumption 3 applies to binary classification and also seems weak when the data is spread out. However, due to the constraints imposed by Assumption 1, it becomes more restrictive. In this case, it requires a good initialization.

Assumption 4 applies to multi-class problems, involves both the data and the initialization, and appears to be much stronger. If understood correctly, it requires that, at initialization, the activations are already essentially separated. Ideally, at initialization, each class $k$ should correspond to a group of neurons in the hidden layer that are activated only by examples from class $k$.

Under these assumptions, the authors show that gradient flow converges, and its limit satisfies Neural Collapse (NC).

The authors also present experimental results demonstrating that NC occurs:
- on MNIST, for two-layer neural networks as well as ResNet-18 and ResNet-50, and
- on CIFAR-10, for ResNet-18 and ResNet-50.

**Questions:**

- Footnote 1: I do not understand this remark. A simple dataset where $(x_i, y_i) \in \\{(1, 1), (-1, -1)\\}$ for all $i \in [n]$ satisfies equation (1) with $\mu_s = \mu_d = 1$.
Does your footnote imply that such a dataset cannot exist? If so, could you clarify why? Is there a reason to exclude such a dataset from your analysis?

- Line 84: I believe that $V \in \mathbb{R}^{d_y \times h}$. Could you confirm if this is correct?

- Theorem 1: Theorem 1 would benefit from including the "additional case-dependent assumptions on the data and initialization shape." Providing them explicitly would make the statement more formal and self-contained.

- Equation (6): Please define the identity matrix $I$ in the notation section for clarity.

- Footnote 2: Please define the acronym ETF. In the formula for $\tilde{E}$, I believe it should be $K$, not $k$.

- Equations below Line 221: There appears to be a typo. In the second equality, it should read: $\forall i \in I_-$ instead of $\forall i \in I_+$.

- Claims in Section 4.1: Could you provide a reference to a specific proposition in the appendix where each claim is stated and proved?

- Line 233: I may have missed it, but could you clarify what $u_+$ and $u_-$ refer to? Please define them or provide a pointer to their definitions. Similarly, could you provide references for $u_k$ and $\tilde{e}_k$ in Proposition 2?

- Line 329: Typo: "pesudo" should be corrected to "pseudo".

**Ethical Concerns:**

["NO or VERY MINOR ethics concerns only"]

**Final Justification:**

The assumptions made in this article considerably limit its scope. For this reason, my recommendation is a borderline accept.

**Limitations:**

As already mentioned, the assumptions significantly limit the scope and practical relevance of the article. To better assess the extent of this limitation, it would be valuable to conduct an experiment testing whether, in ResNet-18 and ResNet-50, the data once mapped in the penultimate layer are orthogonally separated. For instance, the authors could report the empirical values of $\mu_s$ and $\mu_d$. The limitations could be better explained (see suggestions above).

**Paper Formatting Concerns:**

There are no formatting issues.

**Quality:**

3

**Strengths And Weaknesses:**

The article studies an intriguing phenomenon known to occur in deep learning and believed to contribute to the success of neural networks in classification tasks. It is, therefore, an important topic of investigation. The conclusions of the theorems clearly explain Neural Collapse (NC). While I have not verified all the details, the parts I did check were correct.

The main drawback is that the hypotheses of the theorems are very strong and quite far from the settings in which NC has been observed empirically. Although this limitation is common in many theoretical works on deep learning, it is worth emphasizing that the article does not provide a clear path for extending its theoretical results to settings that would explain NC as it appears in practice.

Another drawback is related to readability: the assumptions would benefit from being illustrated with diagrams, examples, and explanatory comments.

In Proposition 1, the assumption $\frac{X_{max}^2}{X_{min}^2} < (2 K -3) \mu_d\mu_s^2$
appears to be strong. In particular, it becomes vacuous if $(2K - 3) \mu_d \mu_s^2 < 1$. This represents a significant limitation on the data, given that $\mu_s < 1$ and $\mu_d \leq \frac{1}{\sqrt{K - 1}}$. For instance, when $K = 3$ and $\mu_d = \frac{1}{\sqrt{K - 1}}$, one must have $\mu_s \geq \sqrt{\frac{\sqrt{2}}{3}} \simeq 0.69$. This implies that data from the same class must be strongly correlated. Such limitations should be better explained in the discussion of the assumptions.

The experiments confirm that NC occurs on standard datasets. However, they do not seem directly related to the theoretical statements. These could be enhanced—or even replaced—by experiments designed to test whether, in ResNet-18 and ResNet-50, the data in the penultimate layer are orthogonally separated. For example, one could report the values of the constants $\mu_s$ and $\mu_d$.

---

> ### Author Rebuttal · Authors · 2025-07-31
>
> We thank the reviewer for the constructive feedback. We respond to your questions/concerns below:
>
> **Extension to deep nets** We have acknowledged the limitations of our results in the paper through multiple remarks. However, as the reviewer pointed out, the discussion on extensions to deep nets was not sufficiently detailed.. We will revise our conclusion section to elaborate more. Specifically, it is our belief that the key technical ideas (alignment dynamics in transient phase, and max-margin bias in asymptotic phase), whose connection to NC was not well explored in prior work, will be useful to study the convergence to NC in deep nets as both ideas appear in deep nets with positively homogenous activations (Kumar and Haupt, 2025; Ji and Telgarsky, 2020; Lyu and Li, 2019). Therefore, following the same high-level proof and utilizing these existing results on alignment and max-margin for deep nets might be the way to extend the current work to a more practical setting.
>
> **Assumption in Proposition 1** We will add a remark to explain the limitation of this assumption on the data correlations. Specifically, this assumption is required to show that the subsets of the parameter space defined in Assumption 4 are invariant under GF. We believe such an assumption is not needed in practice, but our limited understanding of the ROAs and their invariant subsets, as we have discussed from line 279 to line 289, has led to this additional technical condition to ensure directional convergence. Future research on better characterizations of the ROAs and their invariant sets will naturally relax or even potentially remove this requirement.
>
> **Features in ResNet** We clarify that the main purpose of our ResNet experiments is mostly to examine the role of the normalization layer in determining the NC level in deep networks. It is unlikely the features in ResNet would satisfy our separability assumption (in fact, our MNIST experiment already shows that it is only satisfied approximately for MNIST). Nonetheless, the fact that our results align with some of the empirical observations (RMSNorm improves NC) shows that results with the separability assumption are not uninformative for practice, and relaxing this assumption should lead to more practically relevant results.
>
> **Clarifying Footnote 1** Footnote 1 formally says that there exists no dataset satisfying equation 1 with some $0<\mu_s\leq 1$ and $\mu_d>\frac{1}{\sqrt{K-1}}$, we will add its proof to the appendix. Note that the simple dataset the reviewer mentioned does not invalidate this statement, as this dataset satisfies equation 1 with $\mu_d=1\leq \frac{1}{\sqrt{K-1}}$ (note that $K=2$ here)
>
> **Claims in Section 4.1** The Claims in Section 4.1 are directly implied by existing results, which we have referenced at the beginning of the claims. We understand that this makes the paper not self-contained (which is also pointed out by Reviewer ti2o). We will, in Section 4.1, add high-level explanations to the proofs with illustrative figures, and in the appendix, explain how the claims are derived from previous results.
>
> **Definitions of u vectors** $u_+$ and $u_-$ vectors are max-margin vectors for positive-class and negative-class data, respectively, defined in Theorem 1, right after equation (4). In Theorem 1, we have stated two cases, and the "case one" is when $\mathcal{K}=\lbrace +,-\rbrace$, and $u_k, \mathcal{K}=\lbrace +,-\rbrace$ defined for this case gives rise to $u_+, u_-$
>
> Lastly, we will fix the typos the reviewer mentioned (in particular, V is in $\mathbb{R}^{d_y\times h}$, as the reviewer pointed out), and add the definitions for identity matrix and acronym ETF (Equiangular Tight Frame).
>
> We hope our rebuttal addresses your questions. If you have additional questions/concerns, please feel free to post them during the discussion phase.
>
> References:
>
> Kumar and Haupt, Early Directional Convergence in Deep Homogeneous Neural Networks for Small Initializations, TMLR, 2025
>
> Ji and Telgarsky, Directional convergence and alignment in deep learning, NeurIPS, 2020
>
> Lyu and Li, Gradient descent maximizes the margin of homogeneous neural networks, ICLR, 2019

---

> > ### Comment · Reviewer_kUvC · 2025-08-03
> >
> > Thank you for your responses.
> >
> > I am leaving my rating unchanged due to the restrictive assumptions.

---

### Official Review · Reviewer_ti2o · 2025-06-30

**Clarity:** 2
**Significance:** 2
**Originality:** 3
**Rating:** 5
**Confidence:** 3

**Summary:**

The paper describes the phenomenon of neural collapse (NC) in the classification setting, according to which the feature vectors associated to each data point converge to the same direction in the last layer of the network. Going beyond previous works on the topic, the paper rigorously proves that NC happens, under some conditions on the data, even in the setting of two-layer neural networks with ReLU activation function without explicit regularization.
The paper fully characterizes the NC solution in the binary and multi-class classification scenarios and explain how it emerges from the implicit bias of gradient descent.
Some simple numerical experiments validate the analytical results.

**Questions:**

- Can you give some intuition on why it is required that the neurons initializations are exactly balanced? The output weights do not change sign also if $||\boldsymbol{w_{j0}}||^2-|\boldsymbol{v_{j0}}|^2<0$ when the output is a single neuron.
- How much does the result depend on the neurons' balance property? What if the network has a non-homogeneous activation function, with no rescaling symmetry and, therefore, no balance under GF?
- How does the inclusion of biases change the picture? In these cases, they are usually implemented by adding an extra dimension to the input vector that always takes value 1. Does this affect the assumptions of the theorems, like orthogonal separability?
- Does the analysis provide estimates on the time scale $T^*$ required to have NC, and possibly its scaling depending on the network's hyperparameters?

**Ethical Concerns:**

["NO or VERY MINOR ethics concerns only"]

**Final Justification:**

My main concern about the paper in its current form is the lack of clarity in the exposition of the proofs of the main results. The authors addressed this point in the rebuttal by outlining a plan on how to improve this aspect. If this is implemented as promised in the revised version, I think the paper would greatly benefit in terms of clarity and accessibility, thus deserving to be accepted.

**Limitations:**

yes

**Paper Formatting Concerns:**

No formatting concerns.

**Quality:**

3

**Strengths And Weaknesses:**

**Strenghts**
- The setting of validity of the results is limited, but overall the paper provides an interesting step in the direction of understanding the NC phenomenon. In particular, the paper is able to rigorously prove the emergence of neural collapse for the cross-entropy loss without the need of regularization, a result which was previously observed empirically but not treated analytically.
- The paper gives novel technical contributions, like the multi-class analysis, which may be useful for other works in this direction.
- The paper is clear about its contributions and how they connect to known results in the literature. Up to page 4, it is easy to follow and provides a very clear exposition to the NC phenomenon.

**Weaknesses**
- The scope of the theoretical analysis is quite limited, as it requires the strong assumption of orthogonal separability and is concerned with two-layer networks trained with gradient flow.  It should be noted, however, that these are common assumptions in the literature for this kind of analysis.
- The main weakness of the paper in its current form is its dense and technical writing, which is very hard to follow. After page 4, the paper focuses on the proofs of the main theorems. Due to the space constraints, however, a lot of information, notation and details are presented quickly, without taking the required space needed for a reader to process them. For these reasons the proofs, which occupy almost half of the paper, are not particularly instructive and to me mostly feel like they should be in the supplementary material. Moreover, the key steps in the proof of neural collapse in the binary classification case i.e. inter-class separation (Eq. 10) and the claim of line 231 where the implicit bias is involved, are proven in other works. While this is not an issue per se, it means that reading the proof doesn't help a lot in understanding how the result follows from the implicit bias, and thus provides little added value. Overall, I think that section 4 should be much more streamlined, focusing on intuition and leaving the most technical aspects for the supplementary material.

**Minor issues / suggestions**
- The notion of "implicit bias" may not be immediately familiar to every reader, and so it should be explained as early as possible. At the moment is taken for granted and so it is practically undefined. I would add a few sentences in the introduction after line 41 to give some intuition.
- It would be useful to give very specific pointers (equation number) for the parts of the proofs which come from other works, so that it is easier to look them up. For instance in Eq. 7 and in the Claim of line 212.

---

> ### Author Rebuttal · Authors · 2025-07-31
>
> We thank the reviewer for the constructive feedback. We respond to your questions/concerns below:
>
> **Section 4 is too technical** We thank the reviewer for the suggestions. We will, in Section 4.1, add high-level explanations to the proofs with illustrative figures, and also improve the appendix to better explain the technical aspects of the proofs.
>
> **Requirement on weight balancedness** Indeed, the analysis can be extended to the case when $|v_{j0}|>\lVert w_{j0}\rVert$. We will check how much editing is required in the proof to determine whether to modify the main statement or to add a remark.
>
> **Requirement on homogenous activations** We believe it is possible to extend the current analysis to networks with approximately homogenous activations. Specifically, we studied the dynamical behaviors of GF in different phases: alignment in the transient phase and max-margin bias in the asymptotic phase. For the alignment, it is mainly derived from the linearization of the network around the origin; thus, similar analysis can be done with non-homogeneous activations, albeit losing weight balancedness, potentially making the alignment dynamics harder to work with. For the max-margin, it requires positively homogeneous activations. Nonetheless, we note that recently there has been work on extension to approximately homogeneous activations (Cai et al., 2025). We are happy to add the discussion here to the appendix if needed.
>
> **Adding bias** We will add a discussion on the network bias term in the appendix. Specifically, since $\sigma(\langle w, x\rangle +b)=\sigma(\langle [w;b],[x;1]\rangle)$, adding a bias term effectively adds one homogenous coordinate to the entire dataset. Notably, homogenous coordinate increases data correlation: $\langle [x_i;1],[x_j;1]\rangle=\langle x_i,x_j\rangle +1$, thus it mainly affects the negative correlation between data points with different labels. In the case when $\min_i \|x_i\|\gg 1$, the orthogonal separability (among $[x_i;1]$) still holds with the bias term.
>
> **Time scale of $T^\*$** We first note that $T^\*$ is the time to achieve inter-class separation, but not necessarily NC. For binary problems, its scale is characterized in (Min et al., 2024), which mainly depends on the data separability $\mu_s,\mu_d$ and dataset size. For multi-class problems, our current Proposition 1 did not provide a scale for $T^*$, but we believe it should have a similar scale as the binary case. We will work on adding the scale to the revision.
>
> We hope our rebuttal addresses your questions. If you have additional questions/concerns, please feel free to post them during the discussion phase.
>
> References:
>
> Y. Cai, et al., Implicit Bias of Gradient Descent for Non-Homogeneous Deep Networks, ICML 2025
>
> H. Min, et al., Early Neuron Alignment in Two-layer ReLU Networks with Small Initialization, ICLR 2024

---

> > ### Comment · Reviewer_ti2o · 2025-08-05
> >
> > I thank the authors for the responses.
> >
> > - **On section 4:** could the authors be a little more precise on how they plan to improve the clarity of section 4?
> > - **On biases:** I agree with the authors that a discussion about biases should be added to the appendix. I also think that a quick mention to this could be added to the limitations section.

---

> > > ### Author Response · Authors · 2025-08-05
> > >
> > > We thank the reviewer for the follow up questions.
> > >
> > > 1. We believe that the reviewer has two main concerns on the writing of the Section 4: (a) the introduction of many notations without detailed explanation; (b) Proofs that heavily references prior work without elaborating how they lead to NC results. We plan to address them as follows:
> > >  - In Section 4.1: For (a), we will add illustrative figures that shows simple 2-d example of orthogonal separable data, the extremal vectors $x_+,x_-$, together with neuron weights. There will be multiple subfigures: one shows how the neurons' directional dynamics in (7) are defined from input data points, similar to Figure 1 in (Min et al., 2024); another shows how neurons' directions move towards extremal vectors by following this directional dynamics, similar to Figure 1 in (Wang and Ma, 2022). (We mention these figures just to give a rough idea how the illustrative figure might look like). By doing so, we hope to illustrate the neuron directional dynamics and explain why they lead the neurons to align with extremal vectors, which also naturally get the readers familiar with all the notations; And we believe that the first claim about intra-class separation should also be clear as one will be able to observe directly from the figure that if the neurons are align with extremal vectors $x_+,x_-$, then (10) follows immediately.
> > >  - In Section 4.1: For (b), we will expand line 226 to line 230 with a brief introduction to max-margin bias in positive homogeneous networks (Ji and Telgarsky, 2020; Lyu and Li, 2019), with an detailed discussion on such bias in linear networks (Ji and Telgarsky, 2019), which is the basis of our second claim, then explain how such bias lead to the direction collapse as stated in the claim.
> > >  - In Section 4.2: We will add another illustrative figure for multi-class problem, highlight the additional directional dynamics introduced for output weights. The figure will show the extremal vectors in multi-class problems, together with an example of semi-local initialization defined in Assumption 4, which will explain the rational behind Assumption 4: if the initial neuron directions are somewhat close to the extremal vectors, then directional convergence is guaranteed (this also gives an illustration on the limitation of this Assumption). Accordingly, we plan to move most of the discussions on attractors and their regions of attractions from line 271 to line 289 to appendix, and to leave only high-level explanations of the neurons' direction convergence in the multi-class problems, as they should be more or less clear from the illustration.
> > >
> > > This plan requires addition of several illustrative figures and editing on some paragraphs, which should be possible within less than a week. We invite all the reviewers and AC to check this plan and provide comments and suggestions.
> > >
> > > 2. We will include the discussion on adding bias to the network in the appendix, and a quick mention in the limitation section.
> > >
> > > References:
> > >
> > > Min, et al., Early Neuron Alignment in Two-layer ReLU Networks with Small Initialization, ICLR 2024
> > >
> > > Wang and Ma, Early Stage Convergence and Global Convergence of Training Mildly Parameterized Neural Networks, NeurIPS 2022
> > >
> > > Ji and Telgarsky, Directional convergence and alignment in deep learning, NeurIPS, 2020
> > >
> > > Lyu and Li, Gradient descent maximizes the margin of homogeneous neural networks, ICLR, 2019
> > >
> > > Ji and Telgarsky, Gradient descent aligns the layers of deep linear networks, ICLR 2019

---

> > > > ### Comment · Reviewer_ti2o · 2025-08-06
> > > >
> > > > I thank the authors for the detailed response and appreciate their willingness to edit that section. I agree with the plan and believe that the proposed additions will improve the paper significantly.

---

### Official Review · Reviewer_YaRJ · 2025-07-02

**Clarity:** 3
**Significance:** 4
**Originality:** 3
**Rating:** 5
**Confidence:** 3

**Summary:**

This paper provides a rigorous theoretical analysis of the emergence of Neural Collapse (NC) under gradient flow in two-layer ReLU networks trained on orthogonally separable data. Unlike prior works that relied on surrogate optimization landscapes (e.g., unconstrained feature models) or required strong regularization (e.g., weight decay, infinite width), this paper directly analyzes the gradient flow dynamics of a finite-width shallow ReLU network with cross-entropy loss and small initialization. It proves that under orthogonally separable data and mild assumptions on the initialization, the network exhibits all three NC properties: intra-class directional collapse, orthogonal class means, and projected self-duality between classifier weights and class means. The analysis is carried out carefully for both binary and multi-class classification, and supported by proof sketches, detailed appendices, and qualitative experiments.

**Questions:**

1) Empirical results on CIFAR-10 underperform in the appendix: Test accuracy of ~80% with ResNet-50 is well below SoTA and not in the zero training loss regime, where NC is most clearly observed. This weakens the NC interpretation in this setting.
2) Could you briefly explain (even informally) how the transition from approximate early-phase dynamics to exact max-margin behavior remains valid? Does any residual error affect final alignment?
3) Does Assumption 4 hold with high probability under standard Gaussian initialization? If so, please clarify. If not, how robust are your results to neurons initialized near decision boundaries?
4) You mention extension to deeper networks as future work. Do you expect the two-phase picture (alignment then collapse) to hold in three-layer ReLU networks as well?
5) Please clarify lemma 3: my current understanding is the authors bound the deviation between the true directional dynamics and their approximation at each moment in time. If my understanding is correct, then it says nothing yet about how the trajectory evolves due to that mismatch. It remains unclear whether these small local errors can accumulate into large global deviations, which could affect convergence to the correct attractor.

**Ethical Concerns:**

["NO or VERY MINOR ethics concerns only"]

**Final Justification:**

This work is interesting and the author's responses are thorough

**Limitations:**

YES

**Quality:**

3

**Strengths And Weaknesses:**

Strength:
1) Technically sound and rigorous: The main theorems for both binary and multi-class NC are carefully stated, well-motivated, and the proofs build logically on prior implicit bias and dynamical systems literature.
2) Novel theoretical contribution: This is, to my knowledge, the first work proving that NC emerges without explicit regularization in a shallow ReLU network trained with gradient flow. This advances understanding of the implicit bias of optimization.
3) Clear comparison to unconstrained feature models (UFM): The paper carefully contrasts the geometry of NC in ReLU networks with that in UFM (e.g., direction collapse vs. point collapse, orthogonal frame vs. simplex ETF).
4) Unification of implicit bias and NC theory: The authors show how early-time alignment and late-phase max-margin bias together explain NC emergence, offering a conceptually clean two-phase view of the dynamics.

Weakness:
1) The main weakness lies in the use of decoupled neuron dynamics approximations under cross-entropy loss. The paper assumes that neurons evolve independently during the alignment phase (Equations (7), (16), and Lemma 3), despite the fact that the softmax function globally couples the outputs of all neurons. This approximation underlies the entire alignment-phase analysis but is not rigorously justified. In particular: 1) No Grönwall or Lyapunov-type bound is provided to control the accumulation of $O(\epsilon)$ errors over the alignment time scale $T = O(\log(1/\epsilon))$ 2) The transition from approximate dynamics to max-margin behavior is assumed to be seamless, but no stability guarantee is given. 3) In the multi-class case, the approximation of neuron dynamics as independent is especially problematic due to softmax-induced coupling and feature competition across classes.

These points do not invalidate the proofs, but they significantly obscure where the approximation is safe and where it may break down. Clarifying these issues would help broaden the audience and ensure robustness of the claims.

2) Assumption 4 (semi-local initialization) plays a critical role in ensuring neurons align to all class attractors, but the paper provides no quantitative analysis of the attraction basins.

---

> ### Author Rebuttal · Authors · 2025-07-31
>
> We thank the reviewer for the constructive feedback. We respond to your questions/concerns below:
>
> **Why neuron dynamics decouple** During the alignment phase (any time t prior to $T=O(\log(1/\epsilon))$), the neuron dynamics decouple as (16)(17) even with the presence of softmax: The neuron coupling is always through $\hat{y}_i$, and Lemma 1 suggests that during alignment phase, $\hat{y}_i\simeq \mathbf{1}/K$, making the dependence on other neurons negligible up to some $\epsilon$ error. Moreover, this approximation error is well controlled throughout the alignment phase. Specifically, at initialization, the norm of the weights satisfies $\lVert w_j(0)\rVert^2=O(\epsilon^2)$, then one can show that the norm only grows linearly. Then a Gronwall argument shows that for any t before $T=O(\log(1/\epsilon))$, the norm can only grow up to $\lVert w_j(t)\rVert^2=O(\epsilon)$, which gives rise to Lemma 2. The Gronwall argument is used in line 874 in its proof. Finally, $\lVert w_j(t)\rVert^2$ is used to control the error as shown in Lemma 3, and all the subsequent results (Lemmas 4 and 5) on the dynamical behaviors of neurons have taken this error into account.
>
> **What happens in the transition from decouple dynamics to max-margin behavior?** We note that our results adopt two different analyses for two phases. For the alignment phase prior to $T^\*$, the approximation of decoupled neuron dynamics remains valid; for the dynamic phase after $T^\*$, there is no neuron decoupling, nor does the convergence analysis rely on an approximate decoupled dynamics. The asymptotic stability of GF, i.e., the limit point $\theta/\lVert\theta\rVert$ exists, is shown through prior work (Lyu and Li, 2019) with the residual errors fully taken into account, and our results in Proposition show its connection to NC. We will clarify the different analyses for the two phases in the revision.
>
> **Quantitative analysis of the attraction basins** As we have discussed in Section 4.2, ROAs of multiple stationary points of the decoupled neuron dynamics pose unique challenges (to our best knowledge, our paper discussed them first) in understanding the directional convergence of neurons in the alignment phase in multi-class problems, and our current analysis can only provide invariant subsets of these ROAs, as defined by Assumption 4. Admittedly, the invariant subsets are relatively small when compared with the actual ROAs, because random initialization does not satisfy Assumption 4, yet empirically, the MNIST experiments show that the alignment with the attractors from random initialization. The quantitative analysis of these ROAs, as well as the relaxation of Assumption 4, has broader importance in understanding the implicit bias of GF on shallow networks in multi-class problems and is left to future research.
>
> **Accuracy gap between SOTA** We thank the reviewer for the observation. Similar gap between SOTA exists for ResNet18 in the empirical results in our main paper, for which we believe the main source of the gap is the lack of data augumentation: we rerun the ResNet-CIFAR10 experiments with data agumentation and report the NC and Acc. of the trained network after 50 epochs in the following table, from which we see the Acc. improvement from agumentation, but less prominent NC (it is harder to collapse data under augmentation).
>
> | Setting | NC1 | NC2 | NC3 | Loss | Train Acc. | Test Acc. |
> |---|---|---|---|---|---|---|
> |w.o. Data aug, w. RMSNorm | 0.2118 | 0.4848 | 0.2991 | 0.0000 | 99.99% | 81.95% |
> |w. Data aug, w. RMSNorm |0.3564 |0.8906 |0.3769 |0.1111 |96.12% | 89.30% |
>
> For ResNet50-CIFAR10, there was an additional issue of training instability: we used a modified ResNet50, for which the SGD with an initial step size of 0.1 sometimes diverges. This led us to use an initial step size of 0.01. Given that empirically a larger step size leads to better generalization, the small step size in our experiments with ResNet50 further enlarges the gap between SOTA. We will refine the experiments to better align with practice.
>
> **When does Assumption 4 hold** Assumption 4 does not hold under random initialization. We will add this to the remark on the limitations of Assumption 4. We believe the analysis can be extended to all cases when the weight initialization does not exactly land on the boundary between two regions of attraction, as mentioned in Section 4.2, but showing the inter-class separation time $T^*$ exists requires additional technical efforts, and thus is left to future research.
>
> **How to extend to deep nets** We believe a similar two-phase analysis is possible for three-layer networks, given certain initialization and data assumptions. Existing work (Kumar and Haupt, 2025) has shown that weight alignment also happens at the early phase of GF when training deep networks with positive homogeneous activations, and likewise, max-margin bias also appears in the asymptotic phase (Ji and Telgarsky, 2020; Lyu and Li, 2019). We believe our analysis suggests a recipe for connecting these results to NC in deep networks in future works.
>
> **Does Lemma 3 account for error** As we have discussed in previous point, the error accumulates but well controlled until $T=O(\log(1/\epsilon))$, after which we no longer need to control the error as the analysis in the second phase does not require decoupled neuron dynamics.
>
> We hope our rebuttal addresses your questions. If you have additional questions/concerns, please feel free to post them during the discussion phase.
>
> References:
>
> Kumar and Haupt, Early Directional Convergence in Deep Homogeneous Neural Networks for Small Initializations, TMLR, 2025
>
> Ji and Telgarsky, Directional convergence and alignment in deep learning, NeurIPS, 2020
>
> Lyu and Li, Gradient descent maximizes the margin of homogeneous neural networks, ICLR, 2019

---

> > ### Comment · Reviewer_YaRJ · 2025-08-06
> > **Thank you for the detailed response**
> >
> > I thank the authors for the comprehensive answers to my questions. I will maintain my score and recommend acceptance.

---

### Official Review · Reviewer_4Gmk · 2025-07-03

**Clarity:** 3
**Significance:** 3
**Originality:** 3
**Rating:** 4
**Confidence:** 4

**Summary:**

This paper provides a theoretical analysis of the Neural Collapse (NC) phenomenon under gradient flow (GF) dynamics for two-layer ReLU neural networks trained on orthogonally separable data. Unlike prior work that studies NC using unconstrained feature models, this work investigates how NC emerges during training due to the implicit bias of gradient flow, even in the presence of nonlinearities and input structure.

The authors show an emergence of NC from the training dynamics. The paper proves that gradient flow with small initialisation on a two-layer ReLU network trained on orthogonally separable data converges to NC solutions. They also provide a more-refined characterisation of NC, especially intra-class directional collapse, where features of the same class converge to a 1D subspace, not a singleton; most likely due to limited representation power of a two-layer ReLU. Finally, they provide an explaination for the role of implicit bias of gradient flow, in terms of NC, in binary and multi-class classification, and have conducted experiments supporting their case.

**Questions:**

In your theoretical analysis, intra-class features converge to a one-dimensional subspace (directional collapse), rather than collapsing to a single point as in the classical Neural Collapse framework. However, given the universal approximation capabilities of two-layer ReLU networks, one might expect that full singleton collapse is still achievable in principle, even without an explicit normalisation. Do you view directional collapse as merely an emergent structure induced by gradient flow (with small initialisation), or as a reflection of a true global optimum under ReLU constraints? Additionally, your experiments suggest that applying RMSNorm can recover full singleton collapse—does this imply that normalisation layers are effectively necessary to realise the full Neural Collapse geometry in shallow ReLU networks?

**Ethical Concerns:**

["NO or VERY MINOR ethics concerns only"]

**Final Justification:**

This is an interesting work connecting neural collapse with implicit bias. I particularly liked the comparison that authors provided between the DUFM and their approach. I suggest that a brief discussion about this to be added in the final revision of the paper. I recommend acceptance.

**Limitations:**

Yes

**Paper Formatting Concerns:**

No issues

**Quality:**

3

**Strengths And Weaknesses:**

Strengths:
- The paper departs from the more "classical" analysis of Neural Collapse (NC) under the Unconstrained Feature Model (UFM) by characterising NC directly in terms of the input data distribution, without relying on free optimisation variables. While this direction has been explored in some recent works, the authors contribute to this growing line of analysis by incorporating the effect of ReLU activations on the emergence of NC. That said, similar considerations have already appeared in deep UFM settings, where the role of nonlinearity is also studied.
- The paper rigorously demonstrates how gradient flow, even in the absence of explicit regularisation (e.g. weight decay), naturally drives the network towards Neural Collapse structures. By analysing the training dynamics in both the early and late phases, the authors show that inter-class orthogonality arises from early directional alignment of neurons (due to small initialisation), and that intra-class directional collapse and self-duality emerge from the asymptotic max-margin bias of gradient flow. This provides a compelling theoretical explanation for the empirical observation that NC often arises in trained networks without any explicit architectural or regularisation enforcement.
- The paper generalises its results from the binary setting to the more complex multi-class case, where both input and output neuron weights evolve jointly under gradient flow with cross-entropy loss. This introduces non-trivial coupling between dimensions and breaks the relative simplicity of scalar output dynamics. To manage this, the authors identify structured attractors (pairs of class-average inputs and pseudo-labels) and introduce a “semi-local” initialisation condition that ensures convergence to these attractors. Their analysis captures how neuron pairs align with specific class directions and ultimately lead to inter-class feature orthogonality and intra-class directional collapse, mirroring the structure observed in NC. This is a significant step beyond prior analyses, which often restrict attention to binary settings or unconstrained models.

Weaknesses:
- While the analysis of two-layer ReLU networks is mathematically tractable and offers insights into how input data structure and implicit bias give rise to NC, it remains far removed from the dynamics of deep networks used in practice. In contrast, the Deep Unconstrained Feature Model (DUFM) arguably captures the emergent behaviour of deep networks more faithfully, particularly due to the implicit feature propagation and layer-wise alignment phenomena. As such, while this work contributes valuable theoretical insight, its explanatory power for real-world deep learning behaviour is limited.
- The theoretical results rely on two key assumptions that, while enabling tractable analysis, limit general applicability. First, the orthogonal separability assumption on the data—that intra-class inputs are positively correlated and inter-class inputs are negatively correlated—is highly restrictive and rarely satisfied exactly in practice. Second, the initialisation conditions, particularly the semi-local initialisation required in the multi-class setting, demand that each neuron be meaningfully aligned with a specific class direction and pseudo-label from the outset, which goes beyond typical random initialisation schemes. While these assumptions narrow the scope of the results, the authors are explicit and upfront about these limitations, providing clear justification for their modelling choices and discussing how future work could relax these constraints to better reflect practical deep learning settings.

Overall, this is a thoughtful and rigorous contribution to the theoretical understanding of Neural Collapse for shallow ReLU neural networks, and thus I recommend acceptance.

---

> ### Author Rebuttal · Authors · 2025-07-31
>
> We thank the reviewer for the constructive feedback. We respond to your questions/concerns below:
>
> **Comparison with DUFM** We agree with the reviewer that DUFM studies multi-layer networks, but we do not think the DUFM captures practical NC more faithfully due to its two main limitations: (1) it does not incorporate any input data, and (2) as a consequence, it exhibits exact collapse even at the shallowest layer, which is inconsistent with practical networks, where the shallowest-layer features typically show little to no collapse, and collapse becomes progressively more pronounced from shallow to deep layers. Furthermore, the analysis of DUFM lacks convergence guarantees beyond a local result. Our contribution is precisely to (1) incorporate input data into the analysis and (2) bridge the gap between landscape analysis (neural collapse under UFM/DUFM) and convergence analysis (implicit bias in shallow networks), establishing convergence to neural collapse.
>
> As such, although DUFM studies a more practical model than ours, our analysis offers advantages over (D)UFM in several aspects (data dependence, convergence, etc.) Moreover, since the dynamic behaviors (alignment, and max-margin bias) also appear in GF on deep networks with positive homogenous activations(Kumar and Haupt, 2025; Ji and Telgarsky, 2020; Lyu and Li, 2019), we believe our work, although on shallow networks, opens up oppotunities to tackle the problem of convergence to NC in deep networks.
>
> **Can two-layer ReLU achieve singleton collapse** Based on our remark "Intra-class directional collapse" from line 144 to line 154, we believe directional collapse is the global optimal, and normalization along the collapsed direction is one way to achieve singleton collapse as in prior works. First of all, we note that shallow ReLU does not have a universal approximation capability in terms of mapping from input to hidden layer. The universal approximation generally refers to the ability to approximate any input-to-output function mapping $f^*: \mathbb{R}^{d_x}\rightarrow\mathbb{R}^{d_y}$. Here, to achieve NC, the first layer mapping $\sigma(W^\top x)$ must approximate a function $F: \mathbb{R}^{d_x}\rightarrow\mathbb{R}^{h}$ that maps training data to ETF features at hidden layer, but no universal approximation theorem exists in this case. Therefore, we believe the directional collapse is due to the limited representation power of $\sigma(W^\top x)$ for mapping orthogonally separable dataset to collapsed features. Then, our discussion on RMSNorm only suggests that there exists a shallow network with RMSNorm achieving full NC on orthogonally separable dataset, whether RMSNorm is necessary does not immediately follow from this statement. In fact, the understanding of the role of normalization on NC is mainly on an empirical level. Besides our experiments on ResNet with RMSNorm, (Pan and Cao, 2023) demonstrated that batch normalization has a critical influence on the emergence of NC, and their experiments show that models exhibit a stronger tendency toward NC when batch normalization is used. Theoretically proving such a relation warrants a separate paper in future research.
>
> We hope our rebuttal addresses your questions. If you have additional questions/concerns, please feel free to post them during the discussion phase.
>
> References:
>
> Kumar and Haupt, Early Directional Convergence in Deep Homogeneous Neural Networks for Small Initializations, TMLR, 2025
>
> Ji and Telgarsky, Directional convergence and alignment in deep learning, NeurIPS, 2020
>
> Lyu and Li, Gradient descent maximizes the margin of homogeneous neural networks, ICLR, 2019
>
> Pan and Cao. Towards understanding neural collapse: The effects of batch normalization and weight decay." arXiv 2023.

---

> > ### Comment · Reviewer_4Gmk · 2025-08-05
> >
> > I thank the authors for answering my questions and cleared all my concerns. I would like to maintain my score and recommend acceptance of the paper.

---

### Decision · Program_Chairs · 2025-09-17

**Decision:**

Accept (poster)

**Comment:**

This paper proves the emergence of Neural Collapse in shallow networks trained with gradient flow, under the assumption of orthogonally separable data. This improves upon previous results that assumed unconstrained last-hidden layer features, or required specific architectures. This paper brings tools from the convergence analysis of shallow networks to the Neural Collapse anlysis, which is relatively new, and might have other applications.

In agreement with the opinion of the reviewers, we are happy to accept this paper as a poster at NeurIPS.